# Sustainability Assessment of the National Museum of Egyptian Civilization (NMEC): Environmental, Social, Economic, and Cultural Analysis

**Mohsen Aboulnaga** [1,*] , **Paola Puma** [2] , **Dalia Eletrby** [1] , **Mai Bayomi** [1] **and Mohamed Farid** [3,*]

1   Department of Architecture, Faculty of Engineering, Cairo University, Giza City 3725121, Egypt
2   Department of Architecture, University of Florence, 50122 Firenze, Italy
3   Faculty of International Business & Humanities, Egypt-Japan University of Science and Technology (E-JUST), New Borg El Arab 5221241, Egypt
*   Correspondence: maboulnaga@eng.cu.edu.eg (M.A.); mohamed.farid@ejust.edu.eg (M.F.)

**Abstract:** This article presents an assessment of sustainability conducted post the opening of the National Museum of Egyptian Civilization (NMEC), which underwent vast development that had significant impacts, not only on the global level but also on the international attention towards Egypt's great civilization. The study investigates the impact of the NMEC's environmental, social, and economic sustainability and cultural value. Both qualitative and quantitative approaches were adopted. The qualitative includes a preliminary study followed by site visits for collecting data and mapping the four sustainability pillars: environmental, social, economic, and cultural. The quantitative approach has been conducted by exploiting 33 indicators to measure five sustainability dimensions in addition to the UNESCO 15 Thematic Indicators for Culture in the 2030 Agenda; the impact of NMEC on social media using the data scraping technique exploiting GitHub. Energy audit results illustrate that the total annual energy consumption is 491,376.00 kWh (79% in the ground fl. & 21% in the Mummies fl.), as well as 19.98 kWh/m$^2$ (Gr. fl.) and 144 kWh/m$^2$ (Mummies fl.); the first matches RIBA's benchmark for museums, well below the ranking 'Good' (50 kWh/m$^2$). Social sustainability impacts indicate that the word count's effect on social media is 27%, 31%, and 42% on Facebook, Instagram, and Twitter, respectively, while the number of followers is 92%, 7%, and 1%. On Google, it is 1275 and ranks 4.7, whereas the number of posts is 231, 350, and 258. Economic sustainability assessment has been addressed by calculating the revenues throughout one year since the grand opening, and the total revenues amount to USD 2,794,047. The cultural sustainability assessment showed a positive response to the evaluation recorded for 9 out of 15 indicators. The sustainability assessment of the NMEC plays a key role in assuring livable and regenerative cities.

**Keywords:** sustainability; cultural sustainability; environmental sustainability; energy audit; social sustainability; the national museum of egyptian civilization NMEC; Egypt

## 1. Introduction

Being a valuable source of information and cultural vitality promotion along with their significant contribution to heritage preservation, museums can play a vital role in sustaining culture and raising public consciousness [1,2], as well as in affecting the surrounding urban context and social sustainability and contributing to sustainable development and the UN Agenda 2030 and the Sustainable Development Goals (SDGs). This could be through engaging with the surrounding community; fostering tourists' attention; boosting the local economy and its growth; upgrading urban spaces; and, most of all, reflecting culture, history, and identity. In addition, the function of the museum is different from others, as museums can educate, enhance learning and knowledge, connect communities, and create a sense of belonging, as well as promote sustainability in societies by raising citizens' awareness.

Museums and cultural-heritage buildings have a significant global impact, not only on social and economic sustainability but also on environmental and cultural sustainability. In its broad definition, sustainable developments are defined by the World Commission on Environment and Development (WCED) as "the developments that have the ability to meet the present needs without compromising the ability of future generations to meet their own needs" [3]. Becoming a growing priority, cultural sustainability was the main focus within the sustainable agendas of the United Nations post-2015 sustainability goals [4]. Nonetheless, cultural sustainability's impact on sustainable development was significantly acknowledged, and researchers started to focus more on cultural sustainability and to consider it a fourth pillar among the other social, economic, and environmental sustainability pillars [5]. Christer and Akram highlighted the vital role played by museums within both the environmental sustainability and the socio-cultural contexts in the revitalization process of cities. Moreover, they conducted a review of a number of case studies that analyzed the environmental and social impacts of museums on regional developments. Their study ended by listing the different ways in which museums can contribute to societies by increasing both social and environmental sustainability [1]. While Izabela Luiza, et al. conducted an analysis of literature and a survey of 86 museums to study the different impacts of cultural, social, economic, and environmental pillars on reaching cultural stability in Romanian museums. This is where the results have indicated that the social and economic performance of museums does affect their ability to achieve cultural sustainability [6].

With the continuous change occurring in urban environments, it sometimes becomes a hard job to maintain sustainable developments, rectifying the different challenges, such as urban decay, social degradation, loss of identity, and the lack of belonging [7].

In the current perspective, museums are considered a key element in urban-renewal development projects through their role in serving societies and engaging with the surrounding communities, as well as the possibility of partnering with other organizations for the purpose of addressing specific social and economic issues. Beatriz and Silke shed light on a specific type of global museums whose main aim is to reactivate the economy of their cities and analyze the factors that can be addressed in order to determine the capability of these museums in fulfilling their role as urban reactivations [8]. While Robin Foster has provided an insight into the significance of the relationship between social relevance and museums' sustainability by analyzing three museums in Massachusetts to explore their impact on the economic and social revitalization of urban areas, he ended the study by providing suggestions for future urban revitalization practical applications [9]. Along with social and cultural sustainability, museums are regarded as playing a vital role in shaping cities' sustainable future. In recent decades, economic, social, and environmental aspects became the main pillars that architects started to focus on when designing sustainable cultural development. In this context, Elena Lucchi presented a simplified method for the evaluation and assessment of museums' energy performance and environmental quality. The method has been applied to 50 European museums to compare and identify the main problems and vulnerabilities. Finally, the method suggested a strategic approach for the enhancement of museums' energy efficiency [10]. Izabela Luiza Pop and Anca Borza also proposed a set of 33 indicators that can be applied in the assessment and evaluation of museums' sustainability. These indicators resulted from the qualitative research of the conducted interviews and literature reviews that were pursued for the sake of identifying the most significant factors affecting the measure of sustainability in museums [11]. On the other hand, Izabela and Anca have discussed in different research on the topic of museums' sustainability from a different perspective, at which point their research was divided into two parts. The first one discussed sustainability in museums, while the second part elaborated on the significance of museum quality and its effect on sustainability by discussing its influencing factors and testing on practical models [12]. In terms of social sustainability, Yung et al. emphasized the social factors' benefit for heritage conservation in five major facets. They also developed an analytical framework for the related social aspects [13].

In addition, museums play a vital role in making a significant contribution to creating memorable experiences once tourists interact with tourism products or events. Ekinil and Kazmina identify the innovative activities of museums in the Rostov region as a factor in increasing the tourist attractiveness of the region. This study reveals the concept and essence of museums and substantiates the relevance and feasibility of developing innovative activities of museums as a cultural and educational institution and a component of the city's tourism infrastructure [14]. Finally, the sustainability assessment was carried out on cultural heritage and the restoration of a historical building in terms of environmental sustainability, social sustainability, economic sustainability, and cultural sustainability [15,16].

Based on the previous research studies, it is definite that museums contribute to societies, not only through their crucial role in preserving history and local culture, but also by promoting unity and a sense of belonging, as well as contributing to economic and environmental sustainability, which elaborates the vital role it plays in the revitalization process of cities. However, throughout the literature review, it was found that the majority of the studies focus on the impact of museums on one or more of the four sustainability pillars. When addressing the topic of museums, most of the studies discussed the issue from the economic point of view by addressing specific case studies of museums and defining their role in economic revitalization. Therefore, more attention is needed to be given to the impact of museums on the surrounding urban fabric. This work focuses on applying sustainability and cultural assessment to one of the most significant museums that have recently opened in Cairo, Egypt, in terms of examining the four mentioned pillars of sustainability.

Hence, it is imperative to define the term "museum". The International Council of Museums (ICOM) has approved the new museum definition in the framework of the 26th ICOM General Conference held in Prague, the Czech Republic, on 24 August 2022 as "*A museum is a not-for-profit, permanent institution in the service of society that researches, collects, conserves, interprets and exhibits tangible and intangible heritage. Open to the public, accessible and inclusive, museums foster diversity and sustainability. They operate and communicate ethically, professionally and with the participation of communities, offering varied experiences for education, enjoyment, reflection and knowledge sharing*". The ICOM's new definition is aligned with some of the major changes in the role of museums, mainly recognizing the importance of inclusivity, community participation, and sustainability [17]. The report "Culture and local development: maximizing the impact: Guide for local governments, communities and museums," which was published by ICOM and OECD in 2018, highlights the models and guides of museums [18]. According to the International Council of Museums (ICOM) conference in Venice 2018, the contribution of museums to sustainable development is now an essential element to its agenda. "Sustainability is a dynamic process of museums based on the recognition and preservation of tangible and intangible heritage with the museums responding to the needs of the community". Indeed, many economic impact-assessment studies demonstrate that museums contribute to job creation, generate GDP, and bring substantial tax revenues to their communities. In the United States, for example, museums contributed USD 50 billion to the GDP, supported 726,200 jobs, and generated USD 12 billion in fiscal contributions during 2016 [19]. ICOM has also developed a guide for governments and museums that are willing to maximize the impact of museums. These guides provide a self-assessment framework for museums. The guide is organized around the following five themes, as shown in Figure 1 [18,20].

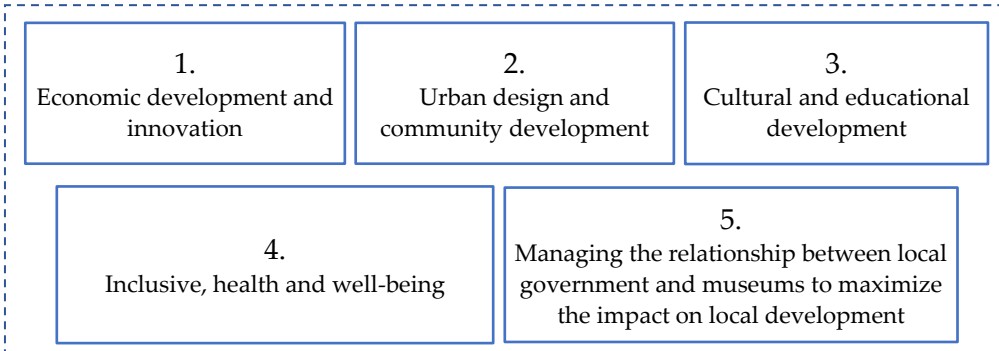

**Figure 1.** The themes of the ICOM Guide for museums (developed by the authors after ICOM).

In addition, ICOM developed a series of actions for each guide that should be discussed and addressed to both museums and local governments. In the study of the NMEC, three out of the five guides will be selected and addressed as a guide and action on the NMEC sustainability assessment. Table 1 highlights the selected guide and action to inform this research.

**Table 1.** ICOM and OECD guide for museums [1].

| Guide | Policy Options for Local Governments' | Action Options for Museums |
|---|---|---|
| **Economic development and innovation** [21] | - Coordinate local cultural institutions and tourism offices with the museum to offer an integrated cultural supply attractive to visitors<br>- Catalyst partnership between museums and economic actors (artisans, SMEs, etc.) for economic development | - Develop a relevant supply of cultural services inside and outside museums to attract tourists and local visitors<br>- Become facilitators of knowledge and creativity by creating opportunities for artists, entrepreneurs, designers, and craftsmen to display and access the collection |
| **Urban regeneration and community development** [18] | - Integrate museums and their environment in urban, ecological design, and planning policy<br>- Support the organization of activities for creating social capital<br>- Promote the development of creative and cultural enterprises into the museum environment | - Consider the museum's place in urban design and its surroundings as a part of the cultural fabric<br>- Develop activities contributing to social capital<br>- Become a center of a creative district<br>- Support eco-friendly initiatives |
| **Cultural development, education, and creativity** [22] | - Recognize the role of museums in cultural and educational development<br>- Take into consideration that cultural heritage can induce and promote reflection and creativity | - Contribute to cultural and educational development as a source of inductive and reflective knowledge<br>- Consider how the presentation and interpretation of collections can support the dissemination of creative skills |

[1] Source: ICOM 2022.

The objective of this study is to investigate the newly opened museum–the National Museum for Egyptian Civilization (NMEC), in Cairo, Egypt. It also assesses the environmental, economic, social, and cultural aspects of sustainability, since the new museum has an internationally recognized historical, cultural, and social value.

## 2. Materials and Methods

The method is based on qualitative and quantitative approaches. The first approach includes a theoretical study (literature review) and a preliminary assessment of the museum after its grand opening in April 2021. The quantitative approach includes a field study to collect data and information about the National Museum of Egyptian Civilization (NMEC) based on the ICOM's new definition of 'museum' in 2022. This was conducted through a successive number of visits to the NMEC main building in order to obtain an in-depth understanding of the four sustainability pillars: (a) environmental sustainability, in terms of an energy audit to evaluate the energy use in the museum's building and the heat island effect; (b) social sustainability, in terms of the impact of NMEC in terms of occupancy pattern (visitors and staff); community welfare and participation; education, including awareness, values, research, and learning tools; (c) economic sustainability, in terms of museum revenues from visitors, and direct jobs created in the NMEC. In this part, secondary data has been obtained and studied in collaboration with museum management in order to analyze the economic impact on a monthly basis and also to create segmentation for the main drivers behind the revenue streams; and (d) cultural sustainability through exploring the museum's crucial role in preserving local culture and its significant impact, not only within the limits of its regional context but also its role in attracting international attention towards Egypt as a country of great civilizations and glorified culture. In this part, the authors have discussed further the built heritage, cultural heritage, and adaptive regeneration, as well as the digital cultural heritage and its impact on enhancing the level of awareness in this regard. Also, how can the NMEC support the 2030 agenda of Egypt and the notion of sustainable cultural heritage in terms of cultural sustainability? This was pursued by applying the UNESCO indicators, as well as Izabela Luiza and Anca Borza's 33 indicators model, which was published by the journal *Sustainability* in 2016, measuring the sustainability of the NMEC. The purpose of the 33 indicators model is to identify the factors that determine the level of sustainability of museums based on a quantitative basis in a way that can be measured and compared with other locations. The model divides the 33 indicators into seven sustainability dimensions' segments: (1) cultural (collection, storage, conservation, and research); (2) social and cultural; (3) social (collection accessibility, community involvement, the museum's social impact); (4) social and economic (labor productivity, efficiency of financial resources used, attractiveness of the museum electronic resources); (5) social and economic (voluntary involvement, capital productivity, consumption of the financial resources); (6) natural environment (using the resources as efficiently as possible), and (7) economic (efficiency, economic impact on the community). This paper is considered to be the pioneer in conducting this quantitative study on a new museum (the NMEC) in Egypt, measuring the 33 indicators, which can create a stepping stone and open the way for further research to be conducted and compare the sustainability of different locations, as well as comparing the sustainability of a specific location over the years to measure the positive progress towards sustainability.

## 3. The Case Study and Scope of Work

The selected case study of this research work is a mega project, the National Museum of Egyptian Civilization (NMEC) in Cairo that bears significant impacts not only on the regional location, but also on the international attention towards Egypt as a country of a great civilization, which is ranked first on the country history index [23].

The NMEC went through many development stages, and its official opening culminated in the spectacular and historical Royal Golden Mummies Parade that majestically took place in the center of Cairo, Egypt, on 3 April 2021, during which mummies of ancient

kings and queens were moved from the Egyptian Museum in Tahrir Square to be finally relocated at the NMEC [24].

The number of mummies and coffins transported was 22 royal mummies and 17 royal coffins, dated back to the 17th, 18th, 19th, and 20th Dynasties. Among these, 18 of the mummies are for kings, while four are for queens [25]. This study mainly concentrates on the museum building after its development to assess the sustainability of NMEC by addressing the four sustainability pillars, as shown in Figure 2.

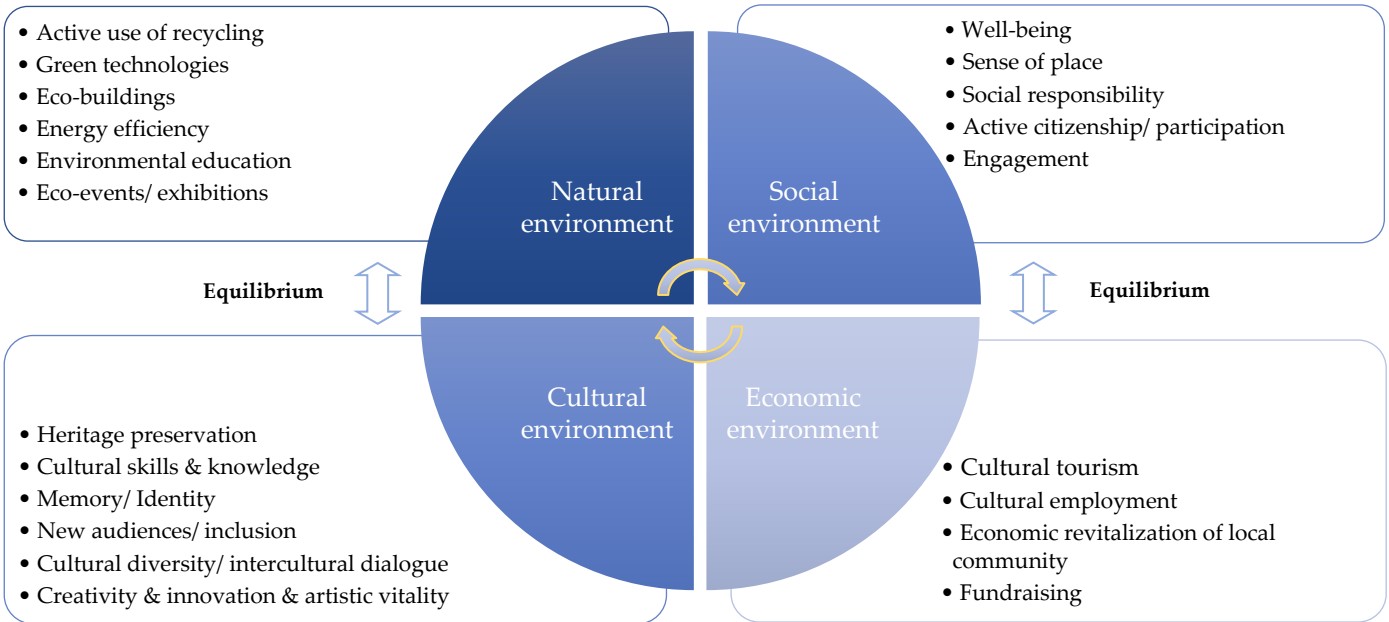

**Figure 2.** Main themes of museums' sustainability addressed in the study approach (developed by authors after [12]).

## 4. Museums and Sustainability

Beatriz Plaza and Silke Haarich pursued a study that aims at highlighting different factors and conditions of museums (Table 2). These factors indicate the degree of success or failure of museums used as urban economic and urban regeneration as part of urban regeneration strategies [8]. In terms of the importance of museums in the urban development plans of any city, Ravzvan-Andrei and Ruxandra Irina stated that museums have become a key element within urban competition due to their high potential for raising local income and developing the local urban economy.

**Table 2.** Conditions needed to be addressed in museum development as economic engines.

| Museum Development and Economic Engines | Conditions |
|---|---|
| Basic location and economic conditions | a. Location and accessibility<br>b. Coherence of cultural investment in a favorable framework |
| Public policy framework of actions | c. Diversified public policy<br>d. Continuous public funding of a museum as an investment in urban and regional development<br>e. Engaging the local |
| The museum project and its management | f. Visibility effect through iconic building<br>g. Branding power of museum or art foundation<br>h. ICT and media going global<br>i. Attractive exhibition and event management |

Museums are very important economically and socially. Furthermore, museums are considered cultural resources for cities [26]. For instance, the British Museum's success vividly manifested such an issue. The study shows that the main key to its popularity was the conditions mentioned in Table 2 and prominently following a mega marketing plan globally, which led to the museum being ranked second in the top-ten most-visited museums [26]. In this context, Pop and Borza proposed a set of indicators that could be used to measure museums' sustainability [11], as listed in Appendix A–Table A1. They also studied sustainable museums for sustainable development, and concluded that sustainable museum causes a sustainable development of the society as a whole [27].

### 4.1. History of the National Museum of Egyptian Civilization Development

The National Museum of Egyptian Civilization (NMEC) was first planned to be located in Ard al Ma'ared (National Exhibition complex) in Nasr City, Cairo (Figure 3a), but the project's design went through many development stages over the years. The location of the NMEC was then changed to beside the opera house in Cairo (Figure 3b). Finally, the NMEC was relocated to its current location, which was decided in 1999 in Al-Fustat (old Cairo capital) in Cairo (Figure 3c). The foundation stone of the NMEC was laid in 2002; the beginning of the first phase was in 2004, and the second phase took place in 2010. The NMEC was partially opened in 2017 [28], but it was officially opened on 4 April 2021, for the public and non-Egyptian (tourists) to both the main museum hall (ground floor) and the mummies hall (lower floor) [29].

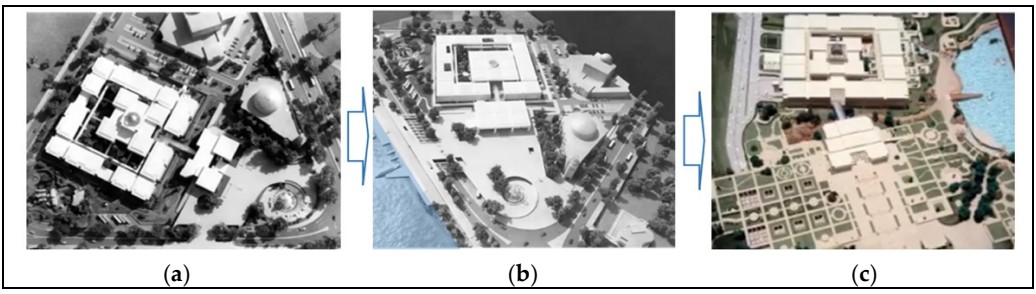

|  (a)  |  (b)  |  (c)  |

**Figure 3.** National Museum of Egyptian Civilization site options: (**a**) first location, Ard al Ma'ared; (**b**) second location, near the Cairo Opera House; (**c**) current location, Al Fustat, Old Cairo.

### 4.2. NMEC Information

Regarded as one of the world's oldest civilizations that accordingly delivered designative archaeological, architectural, and artistic evidence from a wide range of time periods and history, the NMEC acts as an integrated hub that narrates the achievement of the Egyptians across ages. The museum encompasses a large collection of artifacts dating back to the Prehistoric, ancient Egyptian, Greek–Roman, Coptic, and Islamic periods until the modern and contemporary era, beside the end resting-place of 22 mummies [30].

### 4.3. Representation Analysis of the Building and Site

The NMEC is strategically located in old historic Cairo, a "world heritage site" within the boundaries of the city of Al-Fustat, and is in proximity to the famous religious complex, a distinctive historic area. The NMEC overlooks the natural lake of Ain Al-Sira, known for hosting several sulfur springs [29]. The lake was abandoned for a long period of time, and then its surrounding was turned into an informal area (a slum) and a dump site for the excavation of Line 1 of the Cairo Metro [28,29]. Once the museum was planned to be located in this area, a renovation plan was developed to enhance the lake and its surroundings. After completing the renovation phase (Figure 4), Ain Al-Sira (64.9 ac) was turned into a tourist destination as an extension of the NMEC layout (135,000 m$^2$). It included four restaurants (an area of 1.54 ac), green areas, and parks facilitating all services' networks and infrastructure [28,31].

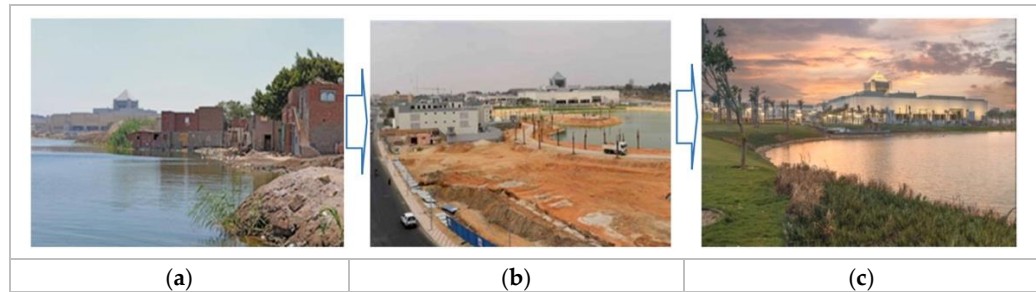

| (a) | (b) | (c) |

**Figure 4.** The renovation phases of the NMEC: (**a**) the site was turned into an informal area, (**b**) during construction, (**c**) the museum site after development in Al Fustat, Cairo.

This study focuses on the ground floor of NMEC and its main hall, as well as the mummies' floor, since it is the only part that is fully and officially open and accessible for the public for 365 days. Figure 5 shows the selected study area in the NMEC floor plans, while Table 3 lists the NMEC building's function and space area inside the museum.

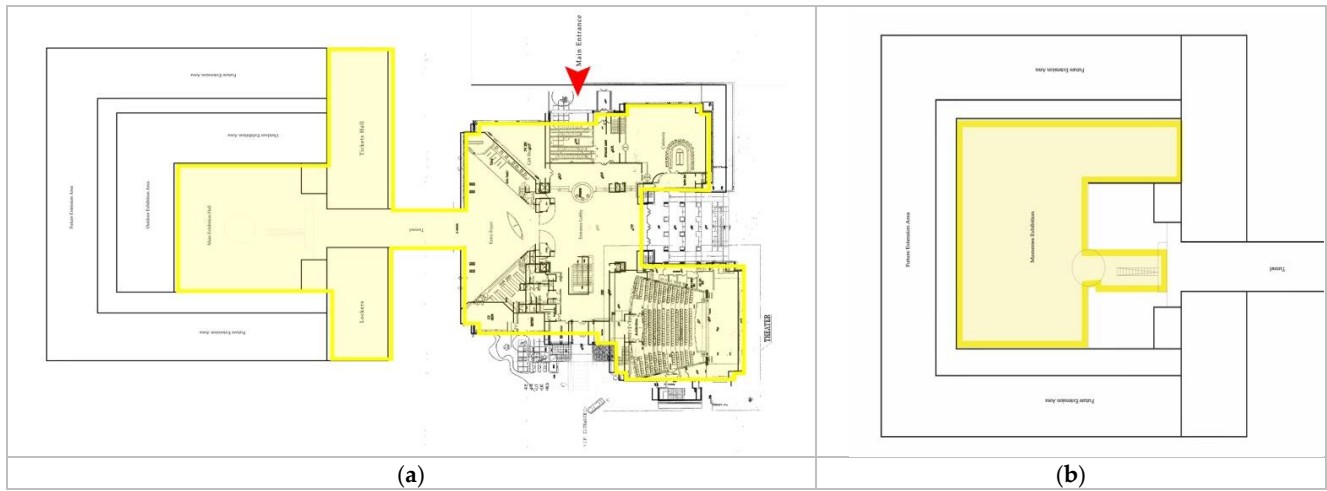

| (a) | (b) |

**Figure 5.** Floor plans of the selected study and area of the NMEC: (**a**) main hall opened area of 19,535 m$^2$ and (**b**) mummies' hall opened area of 700 m$^2$ (Source: developed by authors after NMEC).

**Table 3.** The NMEC building's functions and area of spaces.

| NMEC Elements [1] | Area |
|---|---|
| Total project and its lake area | 398,372.5 m$^2$ (98.44 Acre) |
| Total land area | 135,000 m$^2$ |
| Total ground floor area (opened musuem) | 34,291 m$^2$ |
| Total built-up area | 675,000 m$^2$ |
| Galleries of permanent and temporarily exhibitions | 23,235 m$^2$ |
| The number of scientific labs | 15 Labs. |
| Number of main exhibition halls | 9 Halls |
| Parking area | 11,700 m$^2$ |
| Number of parking slots | 450 cars and 55 buses |

[1] Data of the museum's elements/areas were collected via interviews with the NMEC officials.

## 5. Assessment of Sustainability

In the sustainability assessment, four pillars will be analyzed, including (a) environmental sustainability (energy audit to understand energy consumption and energy efficiency); (b) social sustainability (occupancy patterns, number of visitors and education, and livability through walkability around the NMEC main hall); (c) economic sustainability including revenues and job created by the NMEC within its premises (direct and indirect);

and (d) cultural sustainability by mapping the UNESCO 15 Themes indicators after the opening of the NMEC.

It is imperative to mention that, due to the fact that the NMEC is newly built and opened, there was no record of energy consumption to compare with to measure environmental sustainability; nevertheless, an energy audit will be carried out to understand the energy use pattern and performance coupled with the heat island effect.

In order to assess the sustainability dimensions, five site visits to the NMEC for energy audits and two interviews were conducted with its officials to collect data. These visits took place between December 2021 and February 2022. Figure 6 presents the NMEC floor plan and the new Ain Al-Sira Lake from the outside, while Figures 7–9 portray the NMEC building, including the main hall from the inside and outside landscape.

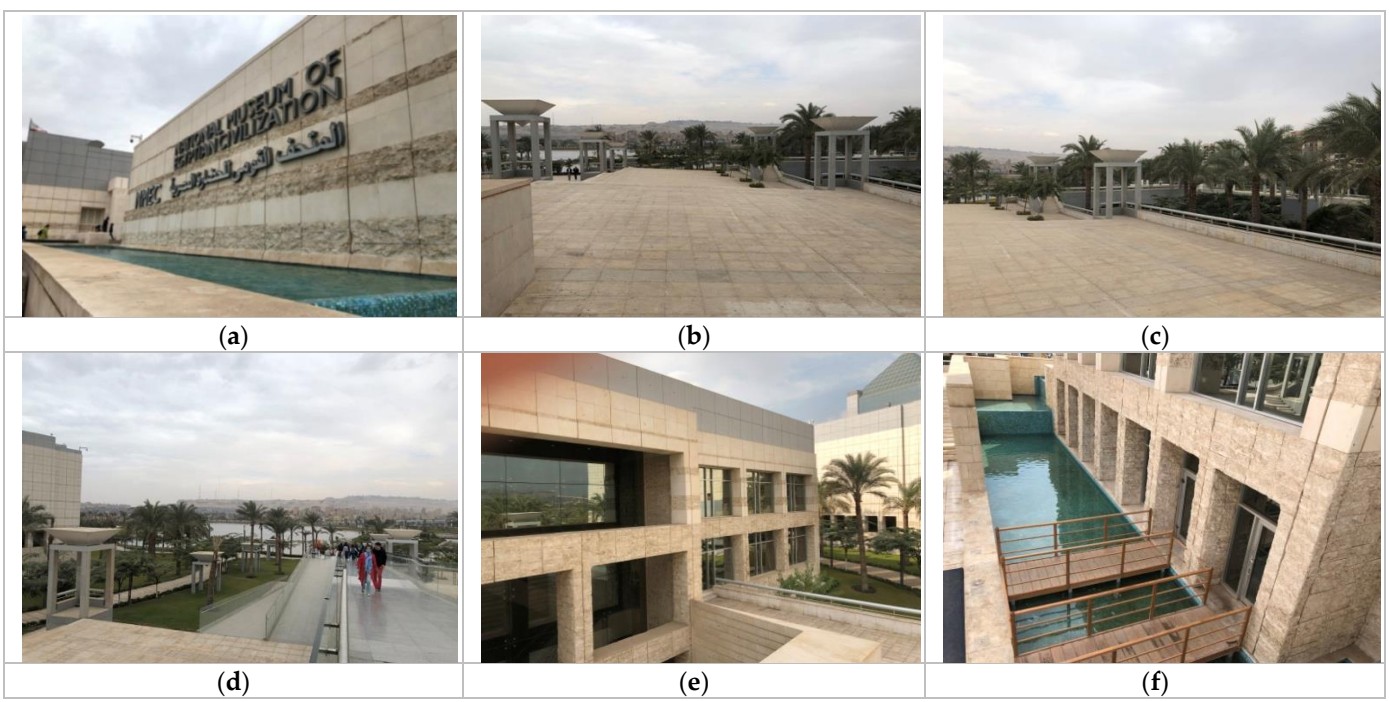

**Figure 6.** The main entrance and plaza of the National Museum of Egyptian Civilization (NMEC): (**a**) main entrance, (**b**) the front plaza, (**c**) landscape around the main building, (**d**) the ramp to the lake view plaza, (**e**) the stone and glass of the side façade, (**f**) a water pond surrounding the lower floor of the museum's main hall.

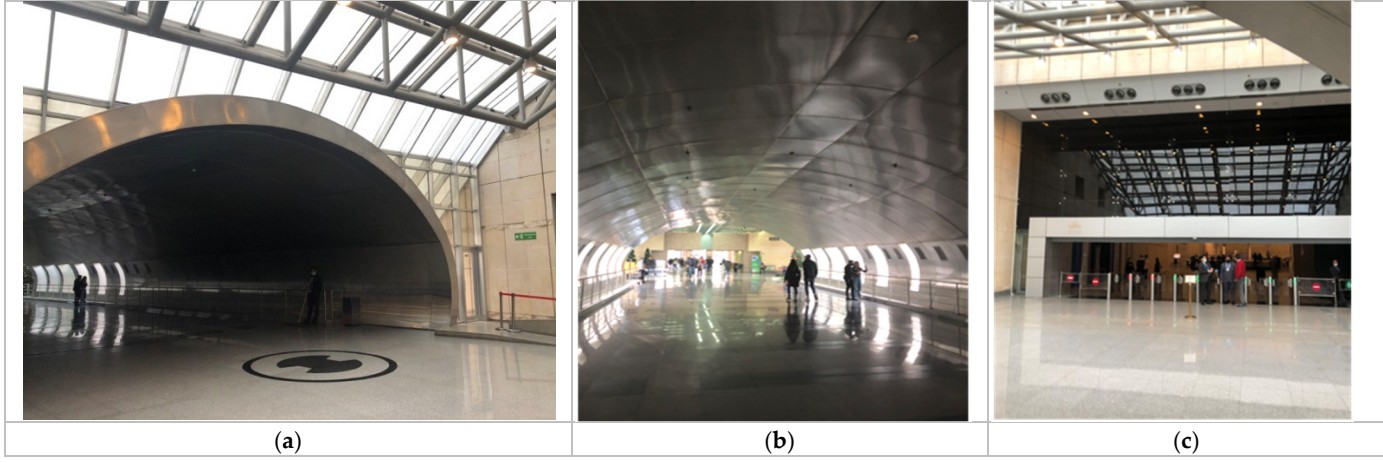

**Figure 7.** *Cont.*

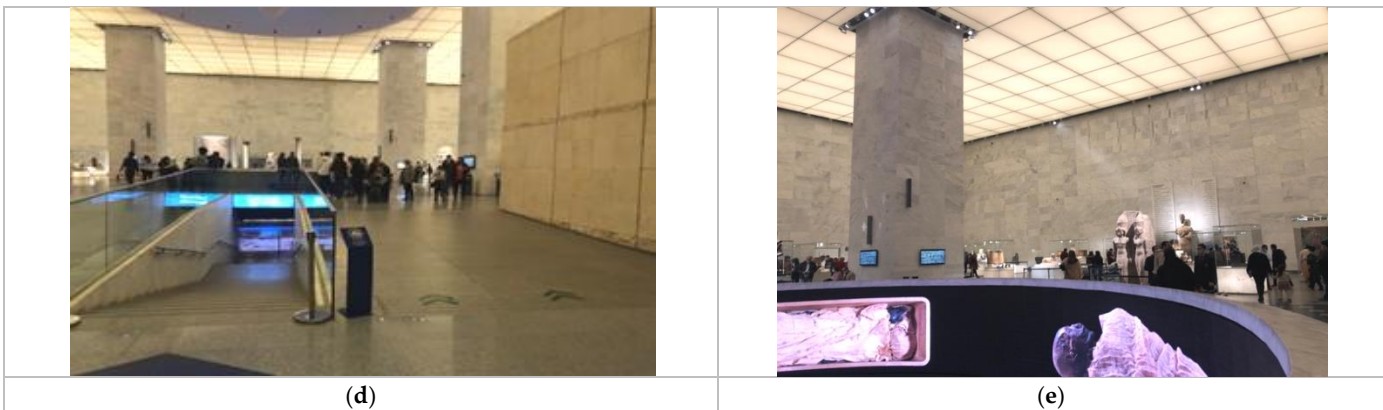

**Figure 7.** The main entrance and plaza of the National Museum of Egyptian Civilization: (**a**) main entrance to the arcade from the tickets desk to the main objects hall; (**b**) the stainless-steel tube connecting the tickets area to the NMEC main hall; (**c**) tickets' check gates for entry to the main hall; (**d**) main exhibition hall and the ramp to the mummies hall; (**e**) mummies' hologram display.

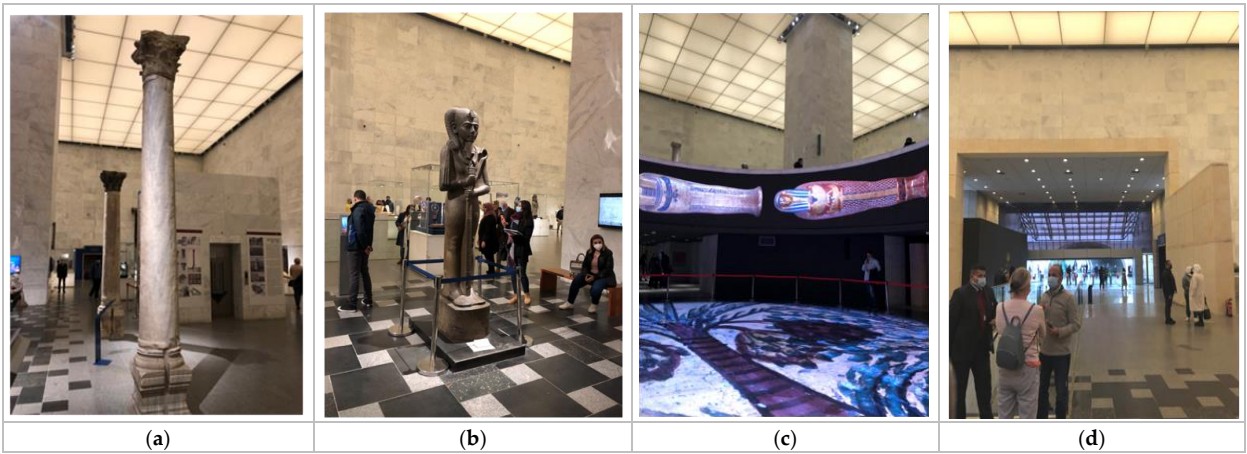

**Figure 8.** Main museum hall exhibiting the antiquities and artifacts of ancient Egypt and Roman civilization at NMEC: (**a**) Roman Corinthian columns; (**b**) ancient Egyptian statue of King Tutankhamun; (**c**) entrance of the mummies hall and lobby with the hologram display of the mummies on the circular floor and wall; (**d**) exit lobby of the museum's main exhibition hall.

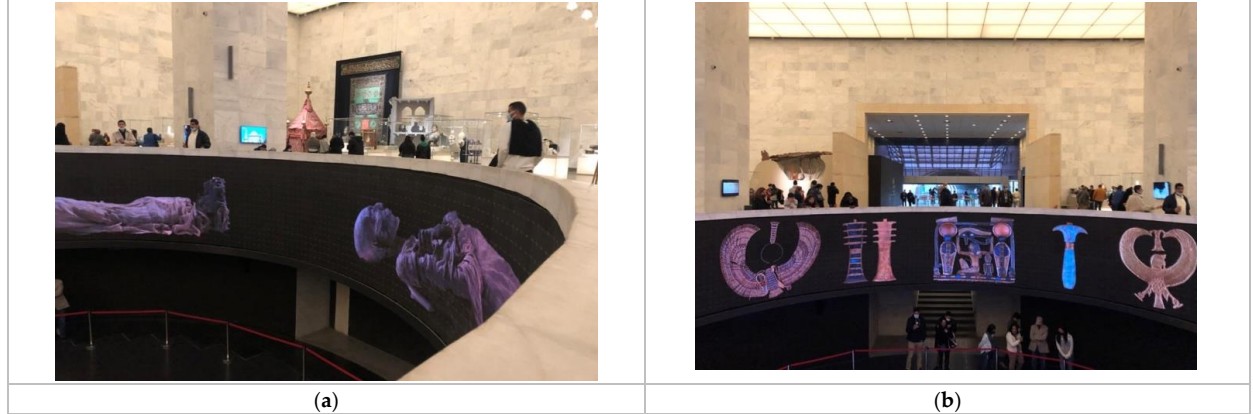

**Figure 9.** The grand hall of the NMEC shows various eras of ancient Egyptian civilizations: (**a**) the mummy of King Ramses II displayed by hologram technology on the circular wall; (**b**) view from inside the main grand hall with the hologram display exhibiting the antiquities (the mummies) and artifacts of Ancient Egypt.

### 5.1. Environmental Sustainability

Environmental sustainability concentrates on assessing energy performance and its efficiency in the main building of the NMEC. Moreover, it focuses on the museum's site in terms of pavement impact on the urban heat island effect (UHIE) of the NMEC site after renovating the surrounding urban area. In order to understand the UHIE of the NMEC and its surrounding landscape, a visit was conducted to the NMEC landscape. This is mainly to observe and map the distribution and type of surfaces and their solar reflective index (SRI), since UHIE can increase energy demand depending on the site location and surrounding landscape, such as pavements, buildings' roofs, vegetation and plants, and water or lakes [32].

Figure 10 illustrates the areas surrounding the NMEC buildings, including built areas, pavements, green areas, and water and vacant lands. Figure 10a–c illustrate land surface overview and annual surface UHIE intensity in degrees Celsius. Figure 11 presents the general surface categories of the region, which consists of 43% built areas with beige roofs; 13% roads and pavements; 13% green areas; 3% water (lakes), including the renovated landscape of the NMEC; and 28% desert-type vacant lands (calculated approximately by authors using AutoCAD software on Google Maps [33]).

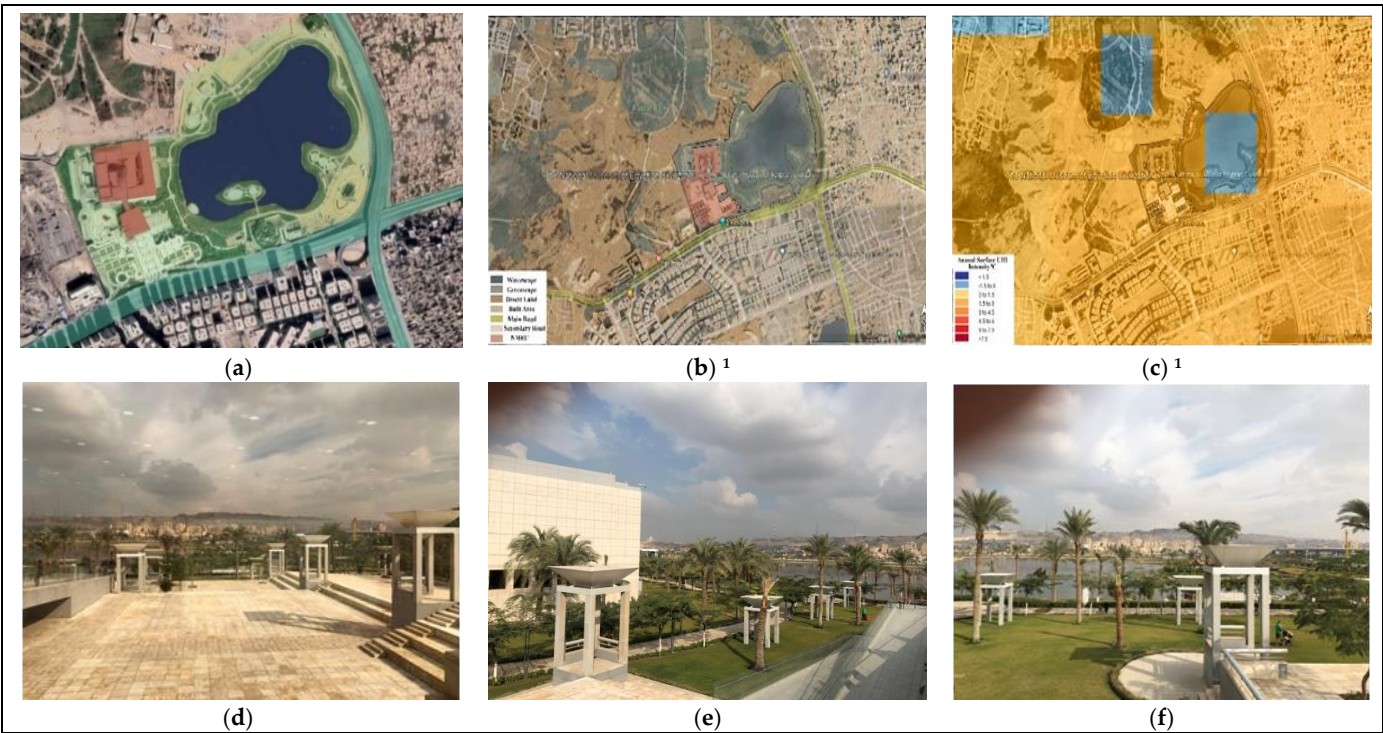

**Figure 10.** The urban heat island effect of the site and landscape of the NMEC after development: (**a**) urban location and renovated NMEC and surrounding site; (**b**) land surface overview; (**c**) UHI annual surface intensity in degrees Celsius; (**d**) bright-color stone tiles of the plaza outside the NMEC building; (**e**) the landscape surrounding the museum's building; (**f**) green areas and water surface surrounding the main NMEC building. [1] Source: Google Maps, edited by authors.

As per the UHI surface intensity (Figure 10c), the immediate surroundings of the NMEC have surface intensity ranging from 1.5 to 2 °C due to being in proximity to a high-density unplanned area with surface intensity ranging from 3 to 4.5 degrees Celsius. It is also indicated that the landscape and Ain Al-Sira lake have caused the surface intensity reduction to range from 0.0 to −1.5 °C

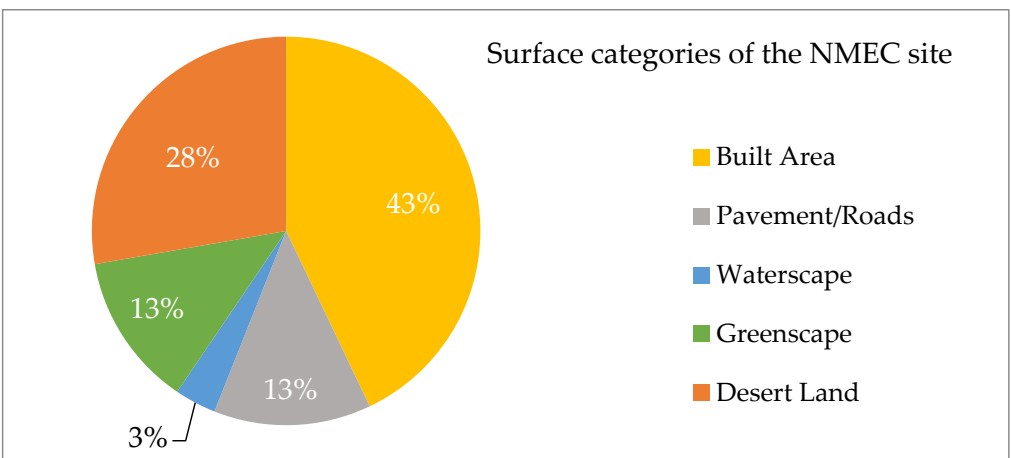

**Figure 11.** The distribution of surface categories for the NMEC HIE analysis.

Such reduction in UHIE is mainly due to the adjacent water area, added greenery, and the usage of light-colored material for the pavements with a high solar reflective index (SRI), landscape, and the repeated sheds along the entryway. All the above have contributed to lessening the UHIE immensely (Figure 10d–f).

Energy Audit

Museums usually present high demand for energy consumption, mainly for air conditioning, due to architectural aspects such as the presence of large exhibition halls [34]. The classes of control are set as specifications for indoor comfort temperature and relative humidity based on the American Society of Heating, Refrigeration, and Air Conditioning Engineers (ASHRAE) [35,36]. For assessing energy in museums, Elena Lucchi conducted a simplified assessment method for the environment and energy quality in museum buildings, which was applied to 50 European museums. This method concentrates on environmental and energy quality, which depends largely on public enjoyment, human comfort, communications, preventive conservation, energy consumption, and safety precautions [10].

In this study, we mainly focus on energy consumption. To understand the energy consumption of the NMEC building and assess its efficiency, an energy audit was conducted through several visits, walk-throughs in the site and spaces to record the lighting system (lamps and power), HVAC, and equipment in the museum, along with having the temperature and humidity controlled due its objects' sensitivity. In addition to meetings and interviews with the NMEC management authority, namely Ahmed Ghoneim, Managing Executive Director of the Authority of the NMEC, and Ahmed Elbayoumi, Director of Finance and Administration. Appendix A (Tables A2 and A3) presents the record of total energy consumption per floor, including lighting, power, HVAC, and equipment. Figure 12 shows the interior lighting and systems inside the NMEC, including the lobby, ticket area, and main objects hall.

Based on the energy audit, the annual energy consumption of the NMEC opened parts, the ground floor (Figure 5a) and mummies' floor (Figure 5b), is estimated to be 491,38 MWh (491,376.00 kWh), and the total average energy consumption/m$^2$ is 20.24 kWh/m$^2$. The annual energy consumption for the ground floor (main hall plan) is 390,346.46 kWh (390.35 MW), while it is 101,029.54 kWh (101.029 MW) for the mummies' floor. The annual total energy consumption per square meter for the ground floor and the mummies' floor is 19.98 kWh/m$^2$ and 144 kWh/m$^2$ respectively.

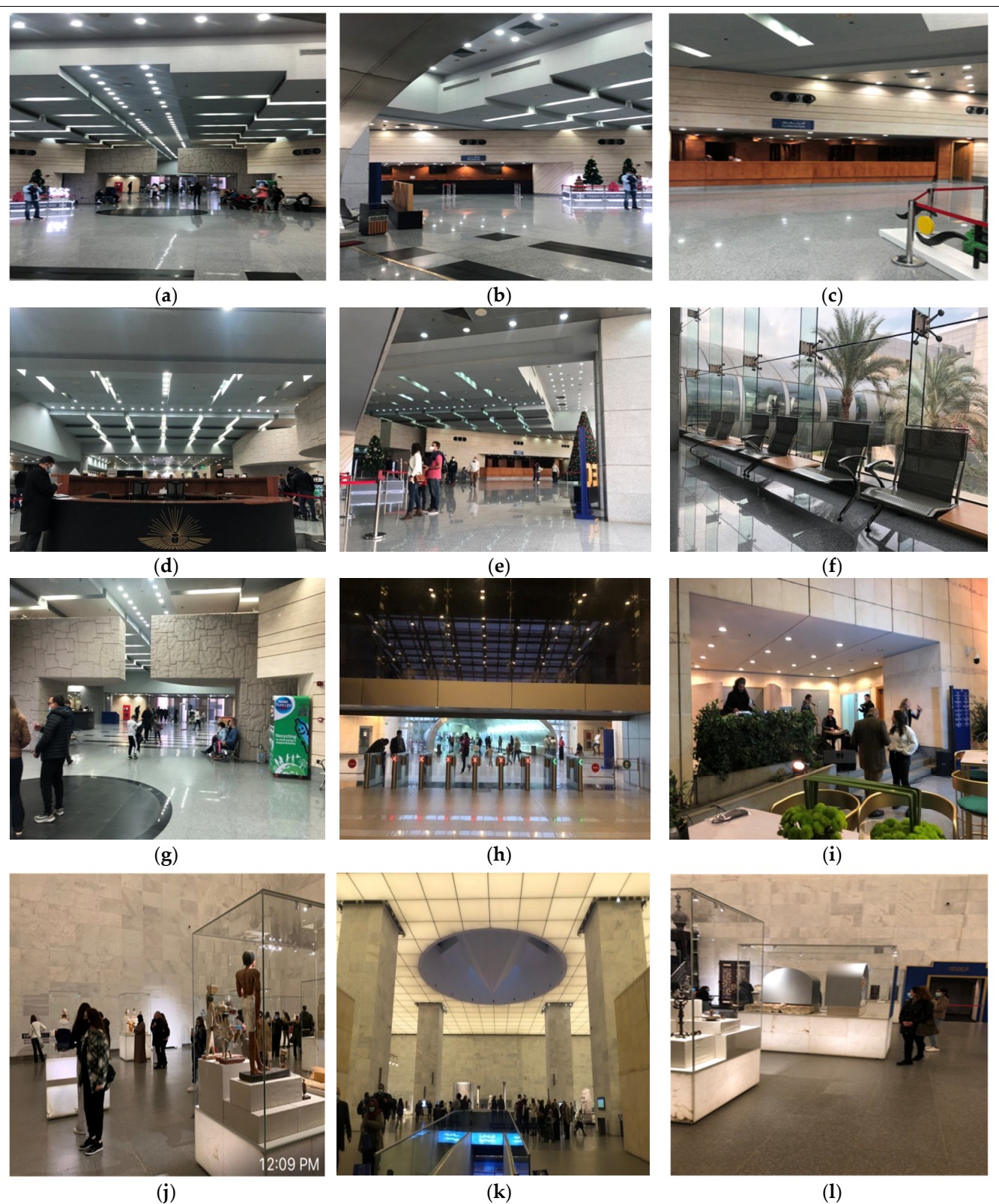

**Figure 12.** *Cont.*

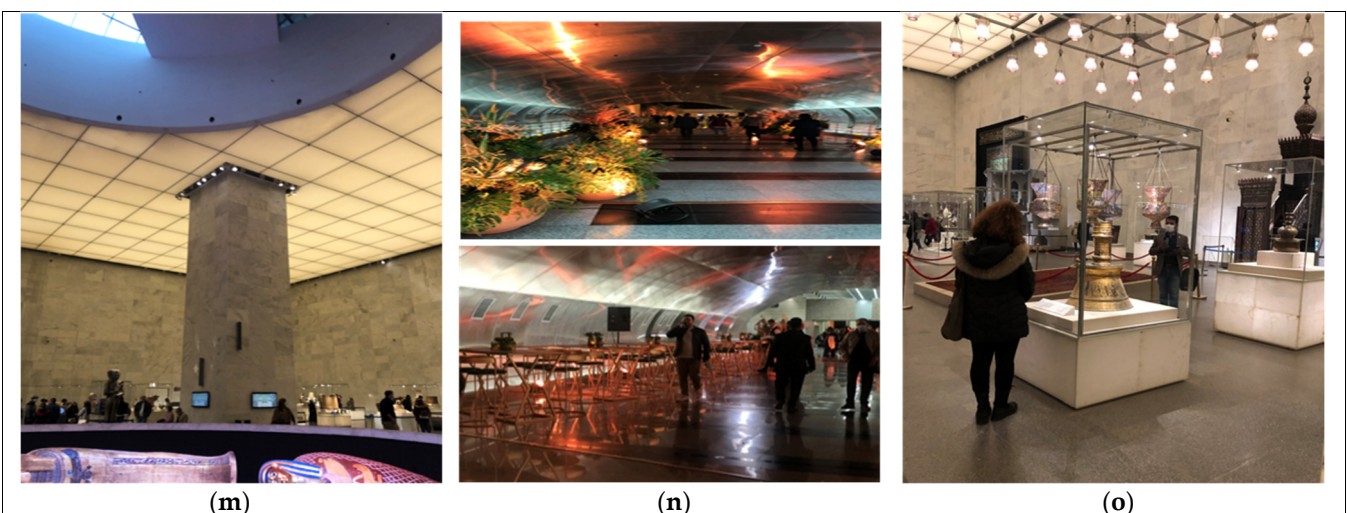

**Figure 12.** Lighting and illumination systems inside the NMEC building: (**a**) LED spotlights, main lobby; (**b**) LED lighting, tickets lobby (**c**) LED spotlights, cloakroom lobby; (**d**) LED spotlights, information desk; (**e**) LED spotlights, tickets desk; (**f**) seating area, tickets desk; (**g**) vending machine for bottle recycling; (**h**) LED spotlight, entry gates; (**i**) LED spotlight, music area; (**j**) spotlight, marble base of display boxes; (**k**) one-space lumen, the main hall, and a skylight; (**l**) spotlights, glass display boxes; (**m**) LED lights around a column; (**n**) lighting of the arcade tube at night; (**o**) hanging chandeliers, 25 in nos.

To measure the environmental dimension of the NMEC based on the 33 indicators developed by Izabela Luiza, seven sustainability dimensions were applied to evaluate its sustainability, as shown in Table A1 in Appendix A.

*5.2. Social Sustainability*

In this part, social aspects addressed by Yung et al., as well as Christer and Akram were studied, analyzed, and compared in order to help in developing an analytical framework for social aspects assessment [1,13]. Yung et al., categorized the heritage conservation and social benefits into five main categories: (a) social interaction and social networks; (b) community and cultural identity; (c) community development and accessibility; (d) cultural diversity; (e) local culture and history, in addition to three sub-categories improving the sense of community, (b) recalling collective memories for all citizens, and (c) providing affordable entrance fees, the latter will be addressed in the economic sustainability section.

Moreover, Christer and Akram have also created a social development framework, in which social development is classified into three main categories: (a) education (including awareness, values, research, and learning tools); (b) culture (including identity, culture exchange, and cultural diversity); (c) social capital and community welfare (including job opportunities, sense of place, social responsibility, public and community participation, social cohesion, and welfare). Figure 13 presents the categories developed by Yung et al. and by Christer and Akram. From the above, it is clear that the main concepts are somehow identical but are discussed only in different patterns. In this assessment, we will address two of the main concepts discussed above. The first and third categories (community welfare (job opportunities) and community and education), while the second category (culture); will be addressed later in the culture sustainability section.

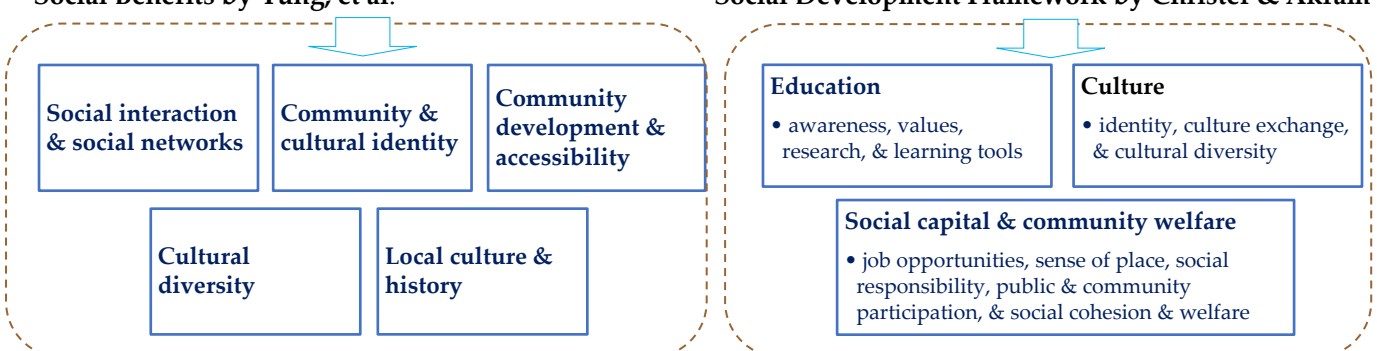

**Figure 13.** Categories of the social benefits and social development framework.[1] Developed by authors after Yung et al. [13] and Christer and Akram.

5.2.1. Occupancy Pattern (Visitors and Staff)

For social interaction and social networking, we measured the visitors' patterns during the time of COVID-19 through site visits and interviews. We addressed the social interaction by recording the visitors' types and activities (Egyptian citizens and non-Egyptian (tourists)) during the operation to identify how the NMEC, after the official opening on 3 April 2021, influences social and cultural sustainability. Table 4 lists records of counted visitors in one year after the official opening. During the observation of visitors (one day), the total recorded number of visitors on weekdays and on weekends is 1500–2000 and 3000–5300, respectively; and non-Egyptian (tourists) visitors account for 20 percent of these numbers. Regarding school visits, a maximum number of 10 school visits are allowed per day, with a maximum of 60 students per school (60 × 10 = 600 students per day) due to COVID-19 measures. Based on the opening hours of the NMEC from 09:00 to 17:00 on weekdays and from 09:00 to 17:00 plus 18:00 to 21:00 on Fridays, we recorded total visitors per year (613,210 persons) and the distribution of total visitors per month (average 51,101 persons). It is imperative to analyze the number of Egyptian visitors to the total number of visitors over one year to find out the social impact of the NMEC and the number of non-Egyptian (tourists) visitors to the total number of MENA-region visitors of the NMEC during one year. This has been assessed in terms of the percentage of visitor types.

**Table 4.** The number of visitors to NMEC over one year [1].

| Visitors' Types | Apr. | May | Jun. | Jul. | Aug. | Sept. | Oct. | Nov. | Dec. | Jan. | Feb. | Mar. |
|---|---|---|---|---|---|---|---|---|---|---|---|---|
| Egyptian | 22,306 | 22,109 | 26,369 | 45,019 | 45,976 | 27,600 | 32,311 | 41,042 | 46,768 | 21,199 | 64,671 | 15,855 |
| Non-Egyptian (MENA) | 995 | 2028 | 2918 | 6170 | 6816 | 5839 | 7846 | 9040 | 9893 | 8036 | 6704 | 10,043 |
| Non-Egyptian (Tourists) | 3316 | 5148 | 4574 | 5471 | 7104 | 8740 | 14,480 | 15,373 | 15,756 | 12,916 | 13,832 | 18,936 |
| Total Visitors | 26,617 | 29,285 | 33,861 | 56,660 | 59,896 | 42,179 | 54,637 | 65,455 | 72,417 | 42,151 | 85,207 | 44,845 |

[1] Monthly and total numbers of visitors from 4 April 2021 to 22 March 2022 are based on data collection and interviews with the NMEC officials (Source: The NMEC).

5.2.2. Community Welfare and Participation

Based on direct interviews with NMEC officials, the direct new jobs created by the NMEC development included almost 500 persons operating as public relations staff, museum administration, tickets, restoration staff, scientists and archaeologists, technical support including information technology staff, engineering department, human resources, and civil security, in addition to extra job creation for the cleaning, security, and gardening services provided by three private companies. Other jobs were created for the museum's cafeteria and the gift shop. Moreover, indirect jobs were created during the construction of the museum and its site; these employ around 2000 people. In terms of community participation, the NMEC has organized a number of workshops and shows aiming at fostering

the attention of the public, especially the youth, by creating an interactive relationship with them instead of being just a place to save and display history.

For community participation, a number of interactive art talks and thematic workshops were held in the museum on different educational topics and crafts, such as the ancient form of painting, "Tempera Painting," "String Art," "Queen Tiye's Secret," and the "Art of Tally", which is one of Egypt's most authentic craft and embroidery methods, as shown in (Figure 14). In addition, Cairo University launched an initiative, Friends of the City, on 19 March 2022 to visit the NMEC to learn and be involved in workshops about Egyptian civilization. Moreover, the NMEC organized a workshop in August 2022 for natural leather artwork and youth visitors, especially those above 18 years of age, as part of community participation. All of these activities and events, among many others, are dedicated to the purpose of community participation through involving and educating the youth [37].

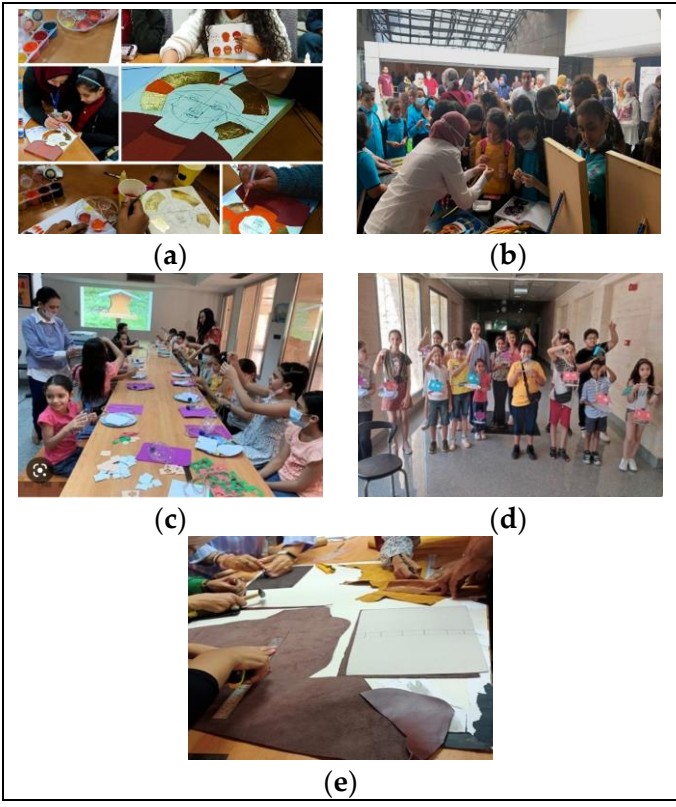

**Figure 14.** Community participation at the NMEC (**a**) tempera painting workshop; (**b**) "Queen Tiye's Secret" workshop; (**c**) engagement of children in reducing material consumption; (**d**) children after waste workshops; (**e**) participation of youth in natural leather workshops.

### 5.2.3. Education (Including Awareness, Values, Research, and Learning Tools)

As for education, awareness, and community cohesion, the museum is considered to be a hub for cultural and educational interaction. School trips and visits are continuously organized, with a maximum number of 10 school trips per day, at which each school trip has a maximum number of 60 students. This allows a maximum number of daily visits of 600 school pupils per day. Moreover, the NMEC organized a number of lectures, workshops, art fairs, and exhibitions that aimed at fostering the attention of the public, especially the youth, through creating an interactive relationship with them instead of being just a place to save and display history.

### 5.3. Economic Sustainability

According to Izabela Luiza et al., culture, economy, society, and environment are considered to be the four main sustainability pillars for museums [6]. These four pillars

together create an ecosystem for the sake of achieving sustainable development. At this point, economic sustainability is discussed in order to reveal its significance in the assessment of the NMEC, which is not considered to be like any other mega project, as it has a very significant cultural and historical, as well as economic, value that reflects the identity of Egypt as a country of great and mighty civilizations.

To define the economic sustainability of the NMEC, it is imperative to identify the assets' value of the urban renewal of the NMEC site and revenues by obtaining the cost of development, number of created jobs, number of employees, and operation costs. For the occupancy pattern, staff, the NMEC, as a new museum and socio-cultural hub, created direct new jobs for almost 520 people, including about 40 volunteers, as mentioned in Section 5.2.2 (community welfare (job opportunities)).

The visits and interviews indicate that the NMEC operating staff consists of 481 people: top management (three people), administrative and financial affairs (15 people), public relations (eight people), heritage staff (59 people), scientists (51 people), restorers (35 people) tickets counter (12 people), cloakroom (two people), Information Technology Centre (eight people), security (162), the NMEC café (six people), and the gift shop (four people), as well as gardeners and cleaning staff, and others (118 people). According to and Ministry of Tourism and Antiquities, the total cost of the development of the NMEC amounted to EGP 615 million—about USD 33.643 million [38].

The revenues were estimated by calculating the number of national and non-Egyptian (tourists) visitors. According to the NMEC, there is a two-ticket system for visiting the site: 200 EGP ($10.94 for non-Egyptian (tourists) visitors) and 60 EGP ($2.28 for Egyptian citizens); the tickets for Egyptian and non-Egyptian students are half of these adult prices [39]. Knowing the price of the ticket types (Table 5), the annual revenues could be calculated. For the economic sustainability assessment, the revenues based on the number of tickets sold are summarized in Table 6.

**Table 5.** Tickets and revenues by Egyptian and non-Egyptian visitors to the NMEC, April 2022.

| Types of Tickets and Service | Ticket Price (EGP) [1] |
|---|---|
| Egyptian citizens [2] | 60 ($3.28) |
| Egyptian students [2] | 30 ($1.64) |
| Non-Egyptian (tourists) visitors | 200 ($10.94) |
| Non-Egyptian students | 100 ($5.47) |
| Car park ticket per day | 30 ($1.64) |
| Photography (for personal use) | 50 ($2.74) |
| Visitors with camera holder | 20 ($1.09) |
| Audio tour | 30 ($1.64) |
| Average monthly revenues EGP | 4,256,265 ($232,837.30) |
| Total annual revenues EGP | 51,075,180 ($2,794,047) |

Source: NMEC website: https://egymonuments.com/en/nmec, accessed 31 Mach 2022) [39]. [1] EGP = $18.28 as per the exchange rate of the Central Bank of Egypt (CBE) on 1 April 2022. [2] Senior citizens (+60 years) and students from schools of architecture and antiquities are exempted.

**Table 6.** Revenues over one year from April 2021 to March 2022 in EGP millions [1].

| Tickets' Types | Apr. | May | Jun. | Jul. | Aug. | Sept. | Oct. | Nov. | Dec. | Jan. | Feb. | Mar. [2] |
|---|---|---|---|---|---|---|---|---|---|---|---|---|
| Egyptian visitors | 21,139 | 13,829 | 15,804 | 25,528 | 24,166 | 15,185 | 18,216 | 15,706 | 19,074 | 12,904 | 30,203 | 12,134 |
| Egyptian students | 1167 | 8280 | 10,565 | 19,491 | 21,810 | 12,415 | 14,095 | 25,336 | 27,694 | 8295 | 34,468 | 3721 |
| Non-Egyptian (tourists) visitors | 4044 | 6097 | 6185 | 9188 | 10,977 | 12,572 | 20,077 | 21,710 | 21,469 | 17,740 | 17,420 | 26,371 |
| Non-Egyptian students | 267 | 1079 | 1307 | 2453 | 2943 | 2007 | 2249 | 2703 | 4180 | 3212 | 3116 | 2608 |
| Total revenues ml. | 2139 | 2405 | 2633 | 4199 | 4594 | 3999 | 5756 | 6315 | 6687 | 4892 | 6642 | 6375 |

[1] Egyptian pounds = 18.45 US dollars at the exchange rate of April 2022 (Source: The NMEC). [2] Revenues are based on the total number of yearly visitors from 4 April 2021 to 22 March 2022 (Source: NMEC).

*5.4. Cultural Sustainability*

The National Museum of Egyptian Civilization (NMEC) cultural sustainability assessment is conducted by introducing an innovation consisting of the use of the UNESCO Thematic Indicators for Culture in the 2030 Agenda (2019, with respect to the model for measuring museums' sustainability according to Izabela Luiz Pop and Anca Broza (2016)). The UNESCO set is a framework of thematic indicators whose purpose is to measure and monitor the progress of culture's contribution to the national and local implementation of the Sustainable Development Goals (SDGs) and the targets of the 2030 Agenda for Sustainable Development. The framework assesses both the role of culture as a sector of activity, as well as the transversal contribution of culture across different SDGs and policy areas [40]. As the UNESCO indicators are a transversal set to all 17 SDGs, they are not a system for directly measuring the impact of culture only on SDG 11–Sustainable Cities and Communities. Moreover, it is a city-level set, and it is not at the scale of the building. To carry out the evaluation of the NMEC's cultural sustainability, it has been adapted to the case study.

5.4.1. Built Heritage, Cultural Heritage, and Adaptive Regeneration

In the 2030 Agenda, sustainable development is a dynamic process of continuous change in which culture provides the necessary transformative dimension that ensures the sustainability of urban development processes. As mentioned in chapter 4.1, Urban Renewal and Museums, in urban regeneration, the presence of cultural institutions, and especially museums, plays a decisive role in marking building interventions that are adaptive and proactive that and increase resilience. *Culture has received insufficient attention as an intrinsic component of sustainable development and must be translated and embedded in national and local development* [41]: For these reasons, the most advanced adaptive urban regeneration strategies are oriented to also include the increase in cultural resilience as a powerful driver of prevention and rehabilitation to promote the ability of communities to take care of their architectural habitat and to preserve the component symbols of the urban landscape and the intangible value of cultural identity.

Architecture and design shape people's lived experience through their built environment, and today's practitioners are considering sustainability in all its aspects as a central principle, emphasizing resilience, climate-friendly design, accessibility, and the identity and heritage of a city [41]: culture-based urban regeneration interventions also increase the cultural identity of the inhabitants, leading them to develop over time a strong sense of belonging to that precise place. Therefore the regenerative processes with cultural roots are among the most valid as they can count on the activation of concentric levels of the proximity of the various communities of reference, from those at international and national scales (as in the case of large museums to which the NMEC belongs) to the inhabitants of the city and the neighborhood. The sense of familiarity that binds the inhabitants of a place to each other and to their habitat develops over time and collects all the social dimensions of the life of a community and its culture. Every day a stratification of knowledge, traditions, and customs takes place in the city that defines it in a continuous, unique, and irreplaceable way to also form the physiognomy of that place. In turn, each change of the city also constitutes a temporal layer represented by a material reality in the physical structure of the city that is deposited daily in the experience and memory of its citizens, indissolubly binding genius loci and genius historiae [42].

In order to obtain innovative territorial development with added social value, the fundamental capital must be based on the connection between memory, constituted by the union connecting cultural heritage and local communities, and the graft of cultural innovation: if the permanence of memory and continuous change intrinsically linked to cultural activities are activated in synchrony, they are powerful catalysts of adaptive regeneration strategies that can bring urban sustainable transformations and resilience, increasing their social empowerment and resistance even to immaterial damages [43].

### 5.4.2. Built Heritage, Cultural Heritage, and Digital Cultural Heritage

By strongly enhancing the communities' and civil society's empowerment, cultural sustainability underlines the identity dimensions of places, that is, the heritage of collective and individual memories that are defined by the terms: *"genius loci, historical sense, context, townscape, place, character, critical regionalism. All of these are none other than different yet interconnected branches of cultural sustainability"* [44].

To work on these intangible dimensions of the urban context, it is necessary to use elaborations aimed at highlighting and communicating both the materiality of the contexts and the meaning of the built heritage (CH). In this scenario, a very important role is played by digital cultural heritage (DCH), which is the CH accessible by digital and media processes. The main currently adopted international lines of cultural policy have recently brought the inferences of the digital society into the heritage sciences, highlighting the importance of DCH, so much so that today this asset is divided into three integrated types: material heritage, intangible heritage, and digital heritage. The DCH inherent inbuilt heritage has recently gained prominence due to its ability to modernize cultural expressions in society, to support social innovation as an economic and inclusive resource, and to make a significant contribution to the cultural pillar of SDG 11 [45]. In this sense, the DCH is the most suitable instrument for the preventive monitoring of contexts for risk reduction, for the construction of the documentary framework, for heritage management in a framework of the sustainability of the transformations and promotion of the resilience of physical and social contexts that over the centuries have established unique identifying heritages.

In the last twenty years, in fact, a new approach to the use of data allowed the use of scientifically reliable metadata that can now be easily activated by the survey–representation–visualization–management–communication chain that data digitization has strongly integrated and speeded up [46–48]. This convergence of tools allows both the acquisition and the representation of multidimensional data to be used for innovative technical and dissemination communication strategies using a very wide range of languages going from 3D reconstructions and animations to augmented reality, from virtual reality to immersive and mixed environments. Since they help to overcome geographical and linguistic borders, digital media can therefore have a wide range of repercussions on culture, on social cohesion, and on the environment. The DCH also allows artistic creations and innovative ways of presenting and consuming cultural content, i.e., access with smart technologies to museums and 4.0 exhibitions or the presence of museums on social media.

The diffusion of electronic devices has allowed the rapid and easy diffusion of the DCH, expanding the audience and also accelerating an innovative approach. Therefore, the DCH is not only a specific key agent that implies a series of transversal effects of economic, educational and technological progress but also a broader cultural evolution that concerns social responsibility for the conservation of our digital cultural heritage, with theoretical aspects and regulatory links with a broad impact on society. Having said that, the NMEC recently launched a mobile application, "NMEC APP", in collaboration with IPMadiX and accessed by Google Play on Android and the App Store on iPhones, to provide an ultimate guide to the museum for visitors as well as in-depth information for those seeking more learning about the history of NMEC's collection and purchasing tickets [49].

### 5.4.3. Agenda 2030, SCH, and Cultural Sustainability

Cultural sustainability is closely related to the notion of sustainable cultural heritage (SCH), which gradually emerges starting from the 2030 Agenda, as a new domain that connects heritage and community and is therefore located at the intersection between economic sustainability (the cultural heritage involves heritage) and social heritage (cultural heritage involves individuals and communities).

All UNESCO conventions and resolutions on cultural heritage have, in fact, gradually updated their principles by implementing them in various SDGs of the 2030 Agenda and making more of UNESCO's efforts in four resolutions, adopted in 2010, 2011, 2013, and 2019, which recognize the role of culture as a strategic resource driving sustainable

development. For about a decade, therefore, a vision of the progressive strengthening of the role of cultural heritage has been affirmed at the international level due to the multiple benefits it brings to the economy, society, and the environment.

The most current notion sees it as a powerful driver of long-term oriented development and an enabling factor that can create employment and redevelop and offer spaces to the communities of various types that live and work around the CH, from professional operators working on heritage sciences to the so-called museum communities.

Only recently, the themes of cultural heritage have been re-interpreted, placing them in a new perspective linked to the general paradigm of sustainability in which CH is present not only for the consolidated need for protection and its intrinsic value as evidence of the memory of the past but also, for being a part of a system projected into the future, wherein cultural policies must be linked to the social cohesion, well-being, job policies, and the environment contributing to sustainable development in all its environmental, social, and economic dimensions.

In the 2030 Agenda, cultural sustainability is cited in the SDGs and targets:

- Goal/target 4.7 "By 2030, ensure that all learners acquire the knowledge and skills necessary to promote sustainable development through contribution of culture to sustainable development";
- Goal/target 8.9 "Conceive and implement by 2030 policies to promote sustainable tourism that creates jobs and promotes local culture and products";
- Goal/target 11.4 "Strengthen efforts to protect and safeguard the world's cultural and natural heritage"; and
- Goal/target 12.9 "Develop and implement tools to monitor the impacts of sustainable development for sustainable tourism, which creates jobs and promotes local culture and products".

The "Thematic Indicators for Culture in the 2030 Agenda" were developed and presented by UNESCO in 2019 to provide a theoretical and operational framework for measuring culture's contribution to the transversal implementation of SDGs and targets. Before conducting the NMEC assessment, a first general comparison has been conducted between the specific "museum" term and DCH domains in the *"Thematic Indicators for Culture in the 2030 Agenda"*. About the specific museums' term, it recurs in the indicators, and the specific DCH domain, which has been detected in the indicators as per Table 7.

**Table 7.** Assessment indicators for the specific museums' term and specific digital culture heritage domain [1].

| Indicator Number | Indicator Description |
|---|---|
| **Specific Museums' term** | |
| n. 2 | *Sustainable management of heritage* |
| n. 4 | *Cultural facilities* |
| n. 12 | *Governance of culture* |
| n. 14 | *Cultural knowledge* |
| n. 20 | *Access to culture* |
| n. 21 | *Cultural participation* |
| **Specific DCH domain** | |
| n. 4 | *Cultural facilities* to detect the distribution of cultural facilities through spatial mapping and spatial analysis |
| n. 5 | *Open space for culture* to detect through GIS the number and size of open spaces used for cultural purposes |
| n.19 | *Artistic freedom* to detect the evidence for activities implemented to promote digital creativity and competencies of artists working with new technologies |
| n. 21 | *Cultural participation* to identify three sub-indicators, including the measurement of individual cultural activities and the use of the internet for cultural purposes |

[1] Developed by authors after UNESCO.

### 5.4.4. The Cultural Sustainability of the NMEC

The assessment of the NMEC' cultural sustainability fits into this framework, which is carried out in close relation to the SCH discussed above. Despite the UNESCO *Culture 2030 Indicators* providing an aspirational tool rather than a normative assessment and being designed to analyze national, urban, and local scales, these indicators provide a useful starting point for the general concept; if critically selected and suitably adapted to the scale of the building complex, the *Culture 2030 indicators* can furnish a fruitful tentative guide to identifying the impact areas to be assessed and a suitable methodology for evaluating the outputs of the cultural institutions' policies. However, the framework of the Culture 2030 Indicators consists of four thematic dimensions, divided into 22 specific assessment parameters based on quantitative and qualitative data (Figure 15).

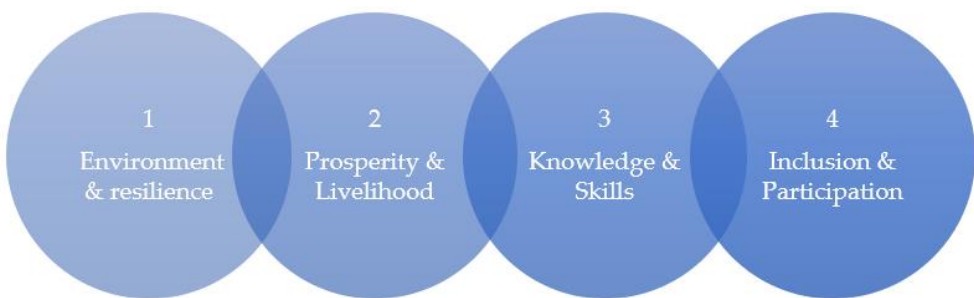

**Figure 15.** The Culture 2030 Indicators' Framework.

Since the NMEC's economic sustainability was assessed in Section 5.3 Economic Sustainability, pp. 17–18, the set was reduced from 22 to 15 indicators derived from dimensions A, C, and D in this study. To carry out the NMEC's cultural sustainability assessment, three NMEC cultural impact areas have been preliminarily identified and defined (the place, the people, and the relation between the place and the people) and have been segmented into 10 points/queries, then compared with the UNESCO indicators (Figure 16).

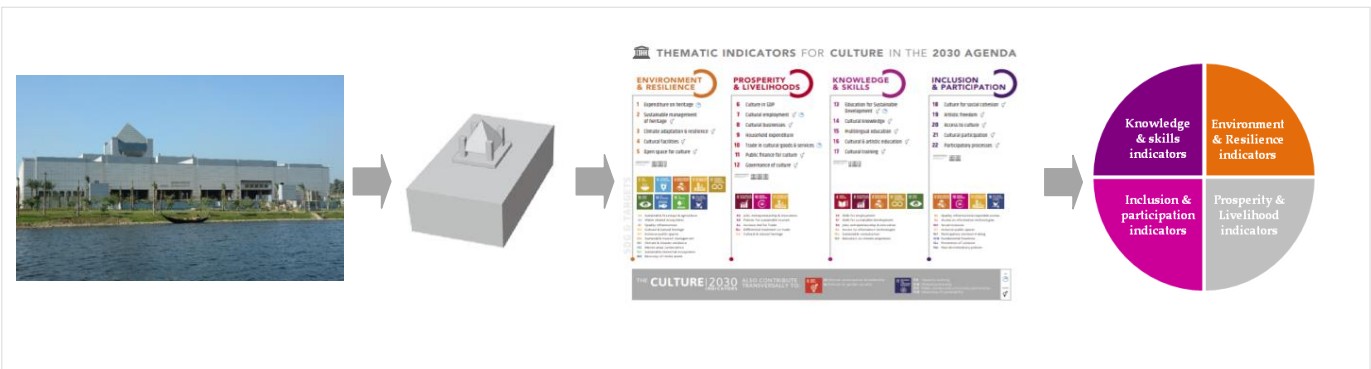

**Figure 16.** The concept of the NMEC's cultural sustainability assessment.

### 5.4.5. The PLACE Issue—About the NMEC Location and the Environment

The cultural sustainability of the NMEC has been analyzed, considering the place as a material trigger of cultural processes, and has been detected by comparing the characteristics of the place with the first dimension of *Environment and Resilience*. The location of the museum itself was deserted, but the surrounding area was inhabited for more than 100 years. In the mid-seventies of the last century, people began to flock to that area from Upper Egypt and the countryside for the purpose of housing and better living conditions in the neighboring areas of Cairo. When the low-income people could not find affordable housing in the city, they resided in the high plateau and built rooms of bricks in the area. The majority of the residents work in tanneries, local art crafts made from gypsum, and

pottery work. Nevertheless, the surrounding slums have been removed, and the inhabitants have been housed in Al Asmarat District (a newly furnished apartments' complex near Al Mokattam hills that provides houses to all these residents with educational, health, leisure, sports, and commercial facilities in the new neighborhood).

The main interest of the NMEC location is being in the historical area of old Cairo and surrounded by many touristic sites in Al Fustat, about 1.5 km from the museum, included in the UNESCO world heritage list. Then it matches with the indicator n. 2 *Sustainable management of heritage* containing the *Checklist* for the Sustainable Management of Heritage to detect the historical urban area recognition, mapping and protection, and the city registration of sites or buildings of historical importance (UNESCO indicators: Table 2 b). Moreover, the NMEC complex is characterized by being inserted in an environment strongly marked by the presence of the Ain Al-Sira Lake that is located south of the valuable area of Al-Fustat (the old capital of Cairo). The lake was formed throughout the history due to earthquakes that hit old Cairo and was abandoned since the Mamulki era due its location being far from inhabited areas. In the last century, the sulfur lake was famous for its abilities to help in treating skin diseases, but in the last 40 years, it was turned into a deserted area surrounded by some slums. Nevertheless, Ain Al-Sira Lake and the whole site underwent a comprehensive urban and landscape development from 2017 to 2021, and the area has been completely renovated, and the NMEC official opening was on 3 April 2021, as shown in Figure 17 and Appendix B—Figure A1.

The environmental renewal matches with the indicator n. 3 *Climate adaptation & resilience,* which measures actions "taken to foster climate change mitigation and adaptation and enhance resilience through sustainable safeguarding and management of tangible and intangible cultural heritage as well as natural heritage" (see Section 5.1. Environmental sustainability, pp. 11–14). However, it is to be reported that the NMEC building does not match this indicator in addressing the criterion: "When related to culture and historic districts of cities it is also important that construction materials, building techniques and architectural styles are aligned with those of historic buildings in the area in question.

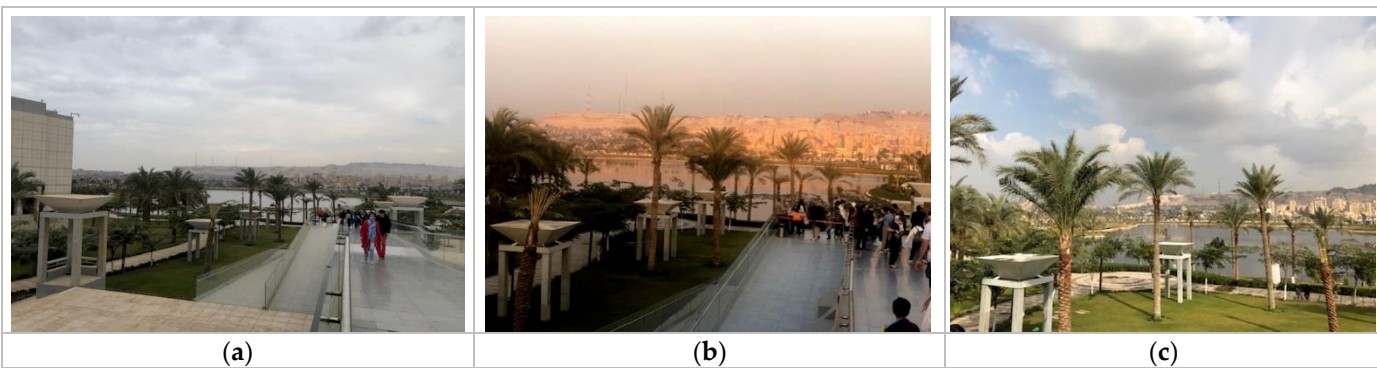

| (a) | (b) | (c) |

**Figure 17.** Ain Elsira Lake and NMEC landscape developed in 2021. (**a**) The lake seen from the NMEC building; (**b**) NMEC visitors walking by the view plaza to the lake; (**c**) close-up view of the lake and landscape. (Source: authors).

Historic buildings will also tend to use locally-sourced materials (though use of architectural material made in distant lands could also be a sign of status)". Having said that, the materials used in the NMEC building' façades are mainly stone and glass.

In addition, a number of interactive art talks and thematic workshops were organized and held at the museum on different cultural and educational topics and crafts such as the Symphony of Art and Sustainability "The first edition of the World Art Forum 2022, International Contemporary Art Forum for Sustainable Development and Exhibition" attended by 380 guests, including five ministers, 14 ambassadors, 24 professors, and many youth (Figure A2 in Appendix B), and a lecture on "Egypt's Royal Mummies: From Discovery to Display"attended by 320 persons. All of these events, among many others, were dedicated

to the purpose of involving and educating the youth and community [37]. Moreover, musical and singing events were performed including the international opera singer Farah Al-Deibani and pianist Amir Awad; another recent "Theatrical Singing Concert" depicting a novel and modern artistic play for monologue art in a musical acting form inspired by the rich legacy of the Egyptian cinema. Figure 15 illustrates one of the main cultural events for the public at the NMEC. Furthermore, a special public event took place on 17 May 2022 at the NMEC main theater and featured a flamenco show by the famous "Anabel Veloso". The event was organised in collaboration with the Spanish Embassy in Cairo and attended by almost 420 persons, including ministers, ambassadors, actors and famous figures, and the public. In total, the NMEC organized about 100 events from April 2021 till the end of September 2022, notably: (a) the renowned maestro Nader Abbassi, who articulated his insight on 'Reviving Musical Heritage"; (b) a lecture by renowned archaeologist and former Minister of Antiquities, Dr. Zahi Hawass, on the most recent archaeological discoveries in Egypt; (c) the Pharaonic Harp Goddess [50].

Pharaohs' Golden Parade

On 3 April 2021, 22 royal mummies were transported from the Egyptian museum in Tahrir to the National Museum of Egyptian Civilization (NMEC) in Fustat in a majestic ceremony that was broadcast in more than 400 international TV channels in 60 countries all over the world in a historic scene [51]. Figure A3 in Appendix B shows the Royal Parade. The Golden Pharaohs' Parade had a huge impact on the economic status, not only that of the museum but also of the country as a whole.

The marvelous Royal Parade was attended by the Head of State HE President of Egypt and high-level government officials, as well as the Secretary General of UNESCO, the head of the World Tourism Organization, ministers, ambassadors, and special guests from all over the world, including journalists, and reporters were invited to attend the ceremony [52].

5.4.6. The PEOPLE Issue—About the Communities in and around the Museum

The cultural sustainability of the NMEC has been analyzed considering the involvement of people and has been detected by comparing the characteristics of this policy with the third knowledge and skills and the fourth inclusion and participation dimensions. The NMEC has two types of Conservation Center Labs (restorations labs and scientific labs where 86 experts are employed. These capacity-building programs and mechanisms match with the n. 14 indicator–Cultural Knowledge (UNESCO indicators: Table 7 "Checklist for Cultural Knowledge: Evidence of capacity-building and training programs). About the informal education policy of the NMEC, school visits are arranged on weekdays and on weekends (see Section 5.2.1 *Occupancy pattern*, p. 15). In this rule, each school sends a teacher to meet the NMEC management to plan for the visit and to confirm the number of pupils per day (a maximum of 120 pupils per day by all school visits). For example, on Thursday, 30 December 2021, there was a visit by a preparatory school trips, including four groups; each is 20 pupils (80 in total plus teachers), and on Wednesday, 5 January 2022, there was a high school trip, each group consists of 14 girls (52 in total plus teachers). Moreover, the museum organizes workshops to train school students and others on maintaining old crafts and artworks that depict Egyptian heritage and culture, and it is managed by the NMEC Deputy Director of Antiquities. Concerning the professional education policy, the NMEC Training Center contributes to serving various sectors of the Ministry of Tourism and Antiquities through providing training courses that aim to enhance their environments and to raise the efficiency of the workers inside and outside the museum in terms of education and archaeological awareness. The same education department has a three-week training programs for tourist guides and gives them certificates for attendance. Regarding lifelong education, the museum organizes various artistic activities for the community and visitors, including exhibitions and life-painting by famous artists such as artist Farid Fadel (Figure 18).

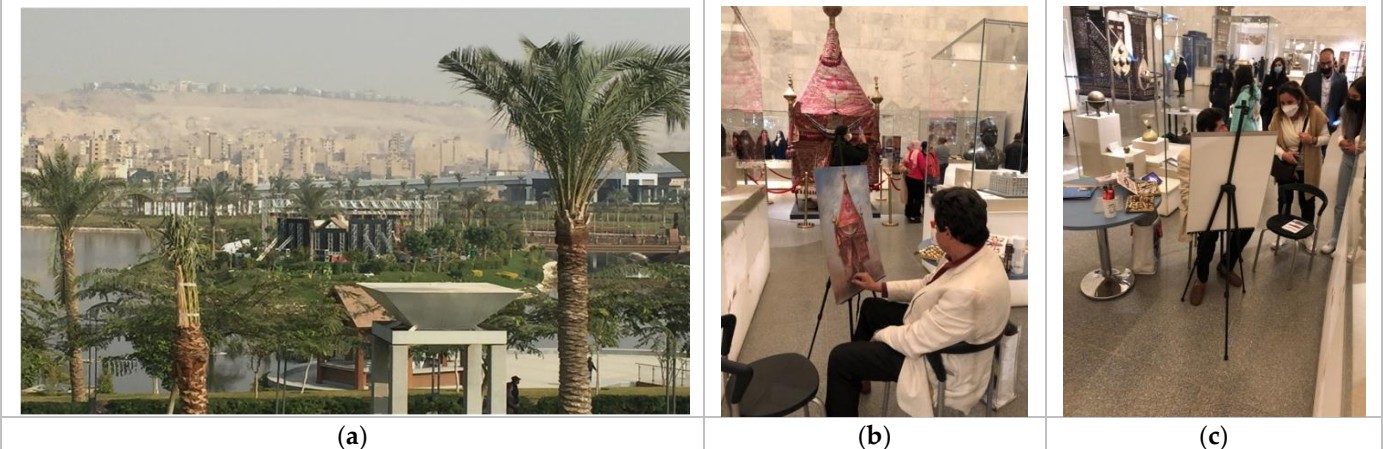

**Figure 18.** Cultural events preparation outside the museum and art sessions inside the NMEC main hall. (**a**) A light steel truss structure for a stage set up in preparation for events on the man-made island located on the edge of Ain Elsira Lake and it is linked by a bridge to the museum's surrounding landscape and gardens; (**b**) A live painting session by famous Egyptian and intentional artist Farid Fadel inside the main Hall of the NMEC, Wednesday, 5 January 2022 from 15:00 to 17:00; (**c**) Visitors are learning the art and painting technique during the live art session.

At the moment, the Museum does not provide multilingual information systems (Indicator n. 15–Multilingual education), but the museum's lockroom provides an English and Arabic amplifier system to tourists' guides and visitors (must be six people) to make their voice audible through the personal sound system and use it inside the museum halls to lower the noise level inside the main hall of the museum. Moreover, a self-service interactive freestanding LCD-touch screen displays in English and Arabic languages. There are five interactive LCD screens inside the main hall alone, as shown in Figure 19.

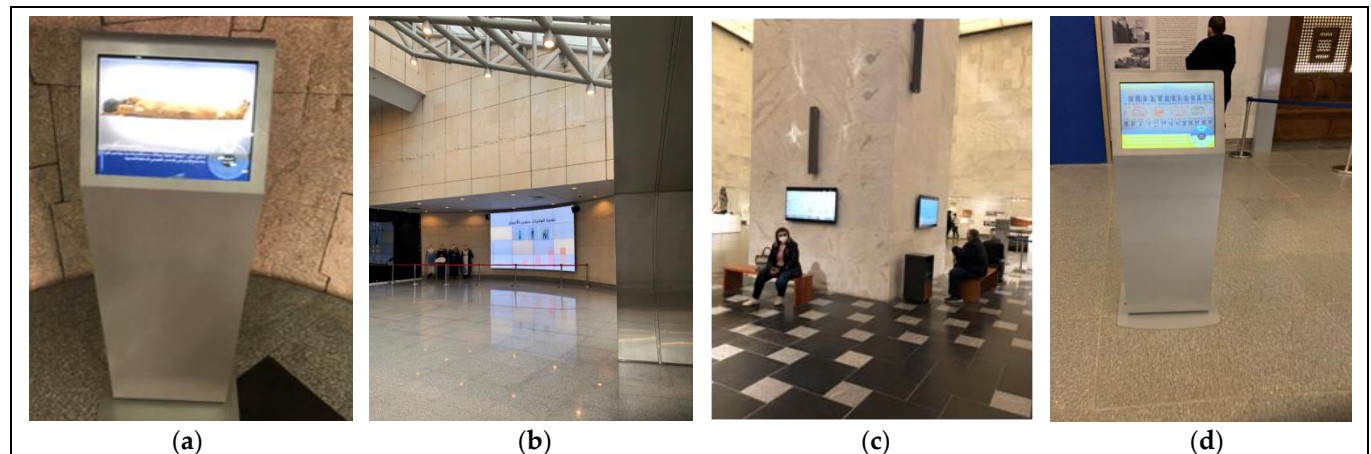

**Figure 19.** Wall and freestanding interactive LCD touch screen display at the NMEC main hall and outside the hall; (**a**) freestanding interactive display screen outside the main hall; (**b**) LCD screen outside the main hall; (**c**) wall LCD interactive display screen inside the main hall; (**d**) freestanding interactive display screen.

Furthermore, many additional events of the more general cultural and artistic education kind are in progress or in program (see Section 5.2.3 *Education* p.16) on a monthly basis throughout the year, and the outreach activities are as follows:

- The NMEC organizes monthly events in the theater and outside the museum building that take place in the large garden near the lake (public/private). Some events are

organized by the NMEC management and others by private companies. For public events, visitors can attend, but not for private ones.

- Among the surrounding landscape and gardens, the man-made island located around Ain Elsira Lake hosts many social and cultural events and act as an aesthetic addition to the NMEC landscape area (Figures 17 and A1 in Appendix B).
- The NMEC hosts the official opening of the World Art Forum (WAF) first edition '*A Symposium of Art and Sustainability*'. The event, held on 15 January 2022, was organized in collaboration with the United Nations Office in Egypt. The WAF art fair was opened to the public from 16–19 January 2022 (Figure A2 Appendix B)).

Such a program of activities matches the criteria of indicators n. 5–Open space for culture describing the extent of public open spaces used for cultural purposes, and n. 14–Cultural Knowledge containing the "*Checklist focusing on cultural education and capacity building*", n. 20–Access to culture describing the availability of cultural infrastructure, and n. 21–Cultural participation describing the Cultural site visits and the Cultural attendance. Since they involve three dimensions of the indicators set, these match and represent the largest area of the cultural sustainability of the NMEC.

5.4.7. The Relationship between the Place and the People Issue

The topic of the NMEC cultural sustainability has been analyzed considering the whole strategy and its current and future effects in/on the community and has been detected by comparing the characteristics of the general museum cultural trend with the dimensions – knowledge and skills, and inclusion and participation. In this context, there are few educational programs for sustainable development (Indicator n. 13–Education for Sustainable Development) at the moment; however, we can report that the NMEC has started to recycle and separate rubbish and waste. This was pursued through three types of waste baskets in every place of the museum. There is a plastic bottle recycling vending machine to encourage visitors, school pupils, and university students to learn about sustainability and recycle their water bottles, as per Figure 12g.

At the moment, the museum doesn't provide multilingual information systems (Indicator n. 15–Multilingual education), but the museum's lockroom provides an English and Arabic amplifier system to tourists' guides and visitors (must be six people) to make their voice audible through the personal sound system and use it inside the museum halls to lower the noise level inside the main hall of the museum. Moreover, a self-service interactive freestanding LCD touch screen displays in the English and Arabic languages. There are five interactive LCD screens inside the main hall alone, as shown in Figure 19.

Despite the fact that the NMEC does not have, at the moment, an official cultural and artistic education program for the general public (Indicator n. 16–Cultural and artistic education) that will be available when the theaters and cinematic experience, as well the amphitheater, are opened, as listed above, the museum is already organizing these events inside (in the lower floor or in the main hall) or outside the museum.

About the policy to access the museum as an "open space" for cultural activities, it generally refers to the fourth dimension: inclusion and participation, assessing the contribution of culture in building social cohesion as well as fostering inclusion and participation, focusing on the indicators to assess the capacity of culture to stimulate the effective engagement of local communities in public life.

In addition, the NMEC outdoor landscape has hosted many cultural/artistic events or events related to business/marketing; some of them were opened to all visitors who purchased NMEC tickets, and some were individual tickets. Many artistic events happen inside the museum and at the art galleries (lower floor), which shows that visitors can attend without any need for buying other tickets.

## 6. Assessment of the NMEC Social and Cultural Impacts

For mapping the NMEC cultural and social impacts, it is imperative to learn about the opening of the museum, as a cultural hub, on social media, specifically Google Trends,

Twitter, Facebook and Instagram, from May 2021 (after the NMEC opening) to May 2022 (one year). This was pursued with the data scraping technique [53] and exploiting GitHub to measure the word counts [54].

This mapping was based on many indicators, including: (a) number of posts per year, (b) number of followers per year, (c) number of likes per year, (d) number of people who check in on Facebook, (e) number of reviewers on Google and number of mentions on Google by object, and (f) Google ranking.

## 7. Results

Figures 20–34 and Tables 8 and 9 present the results of the assessment of the case study in terms of the environmental sustainability, social sustainability, economic sustainability, and cultural sustainability of the urban renewal of the NMEC, including the word counts on social media (Facebook, Instagram, Twitter, and Google) and the application of the UNESCO 15 Thematic indicators.

Figure 34 and Table 9 also illustrate the analysis of the UNESCO and Izabela Luiz Pop and Anca Broza's 33 indicators as well.

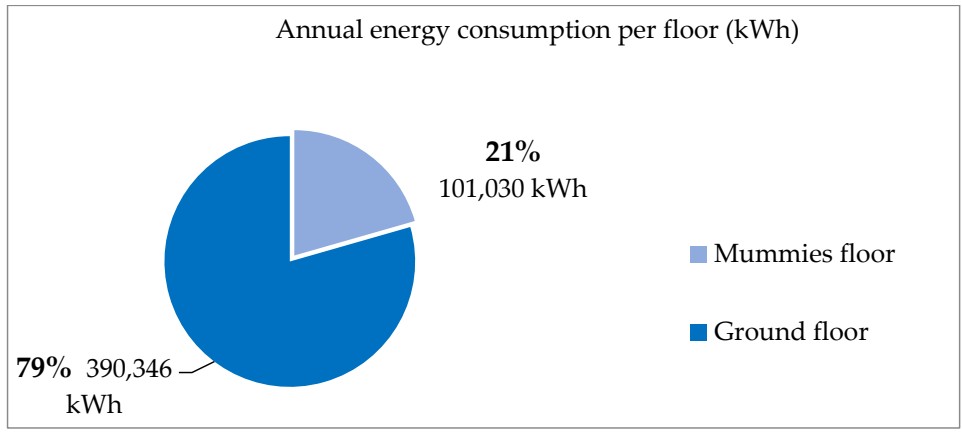

**Figure 20.** The NMEC energy consumption for the main museum building per floor.

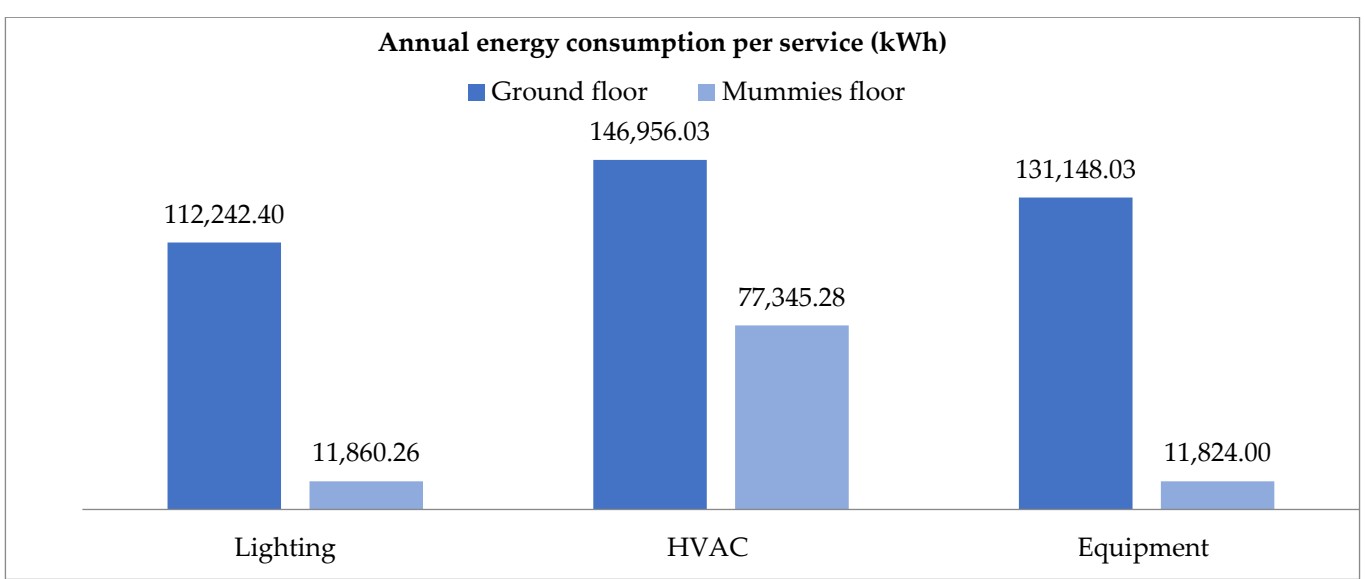

**Figure 21.** NMEC annual energy use by service for the ground floor and the mummies' floor.

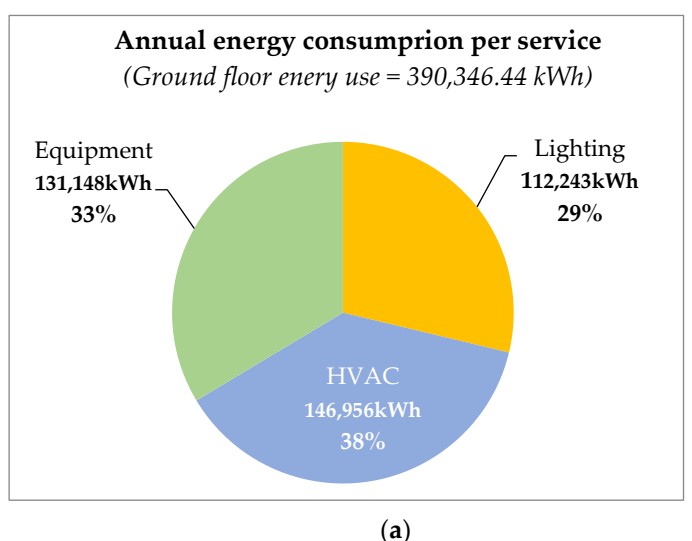

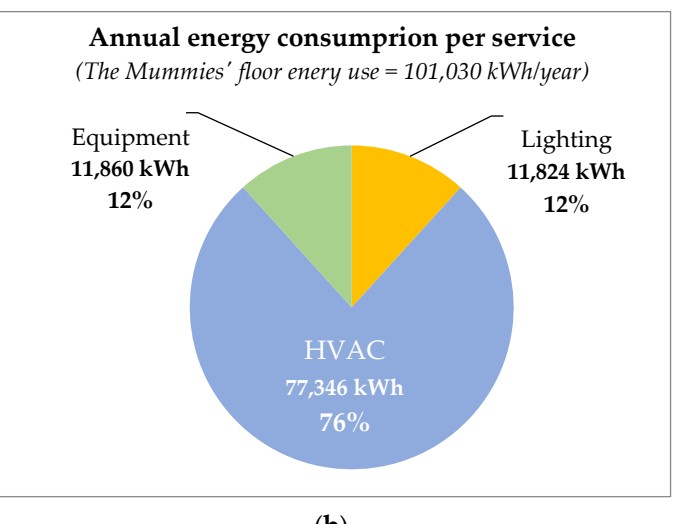

(**a**)

(**b**)

**Figure 22.** The NMEC annual energy consumption for the main museum building per type. (**a**) Annual energy consumption per service (kWh/yr)–ground floor; (**b**) annual energy consumption per service (kWh/yr)–mummies' floor.

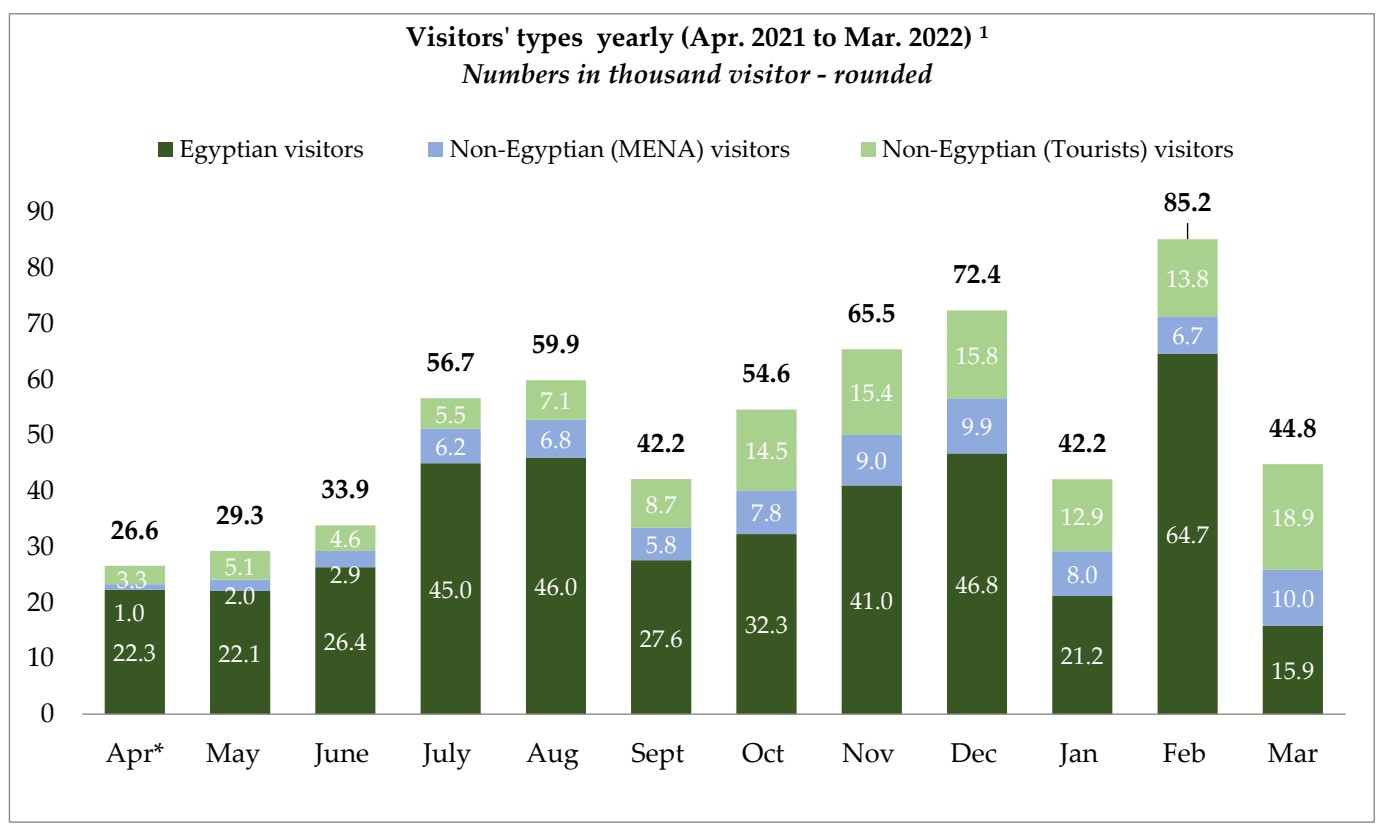

**Figure 23.** Distribution of visitors to the NMEC per month over one year. [1] Developed by authors based on data collection from the NMEC officials, * April 2021.

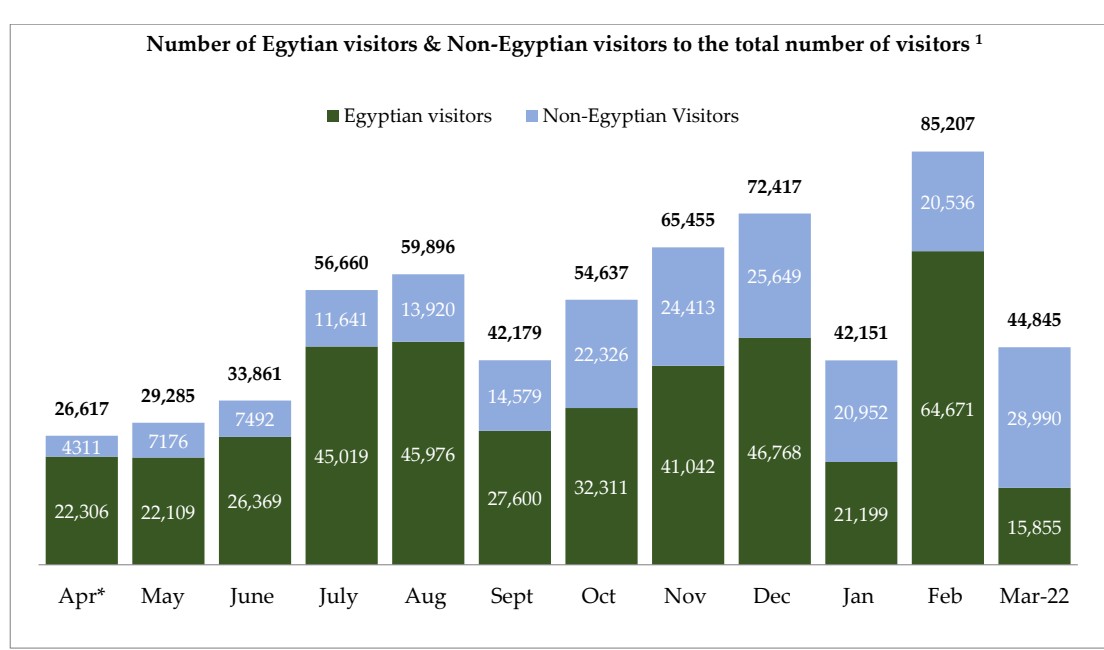

**Figure 24.** Number of Egyptian visitors to the total number of visitors over one year. [1] Developed by authors based on data collection from the NMEC officials, * April 2021.

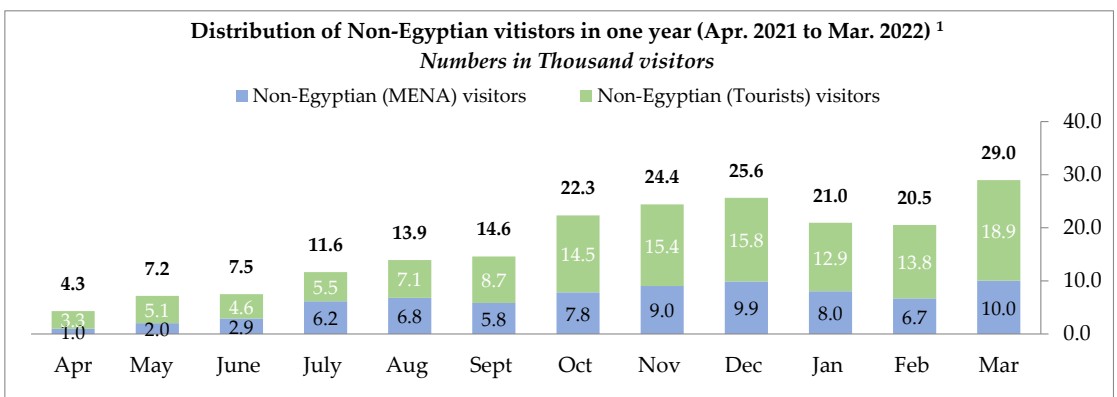

**Figure 25.** Number of non-Egyptian visitors of the NMEC during one year (2021–2022). [1] Developed by authors based on data collection from the NMEC officials.

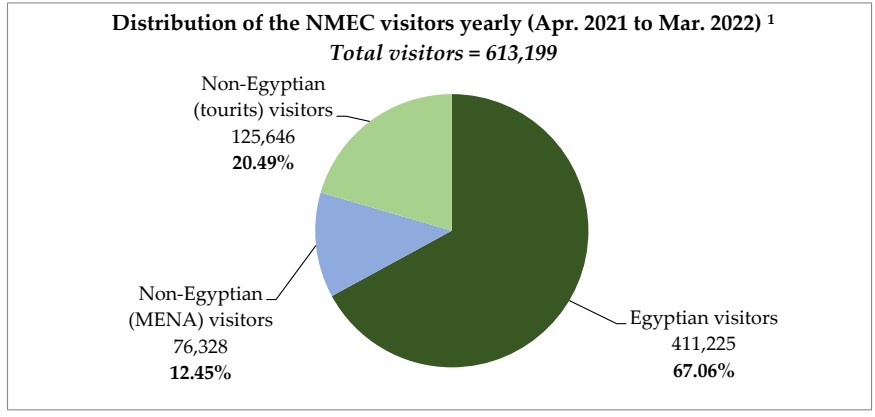

**Figure 26.** Visitor types to the NMEC during one year (2021–2022). [1] Developed by authors based on data collection from NMEC officials.

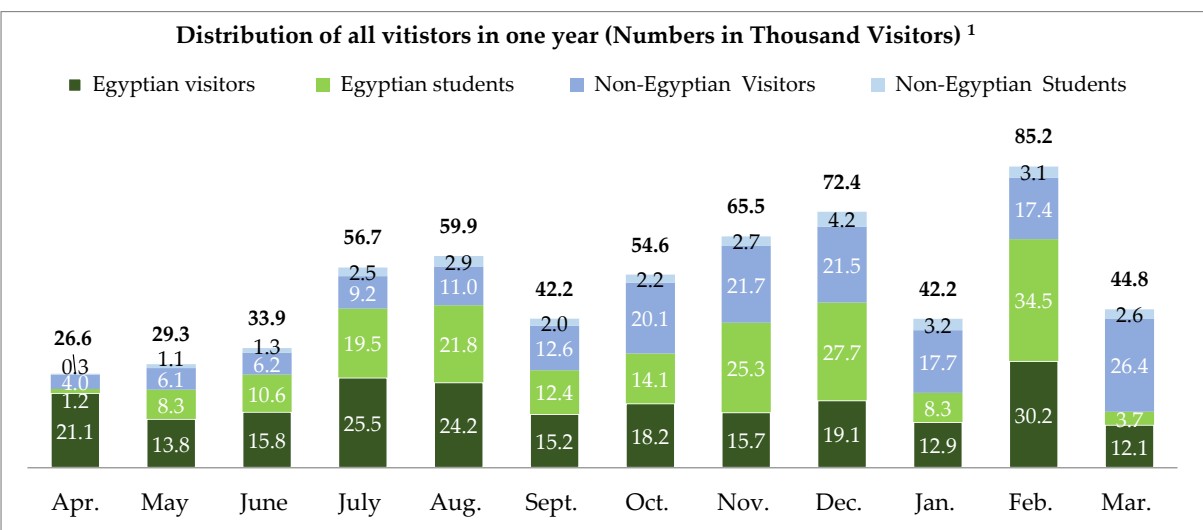

**Figure 27.** Distribution of all visitor types to the NMEC during one year (2021–2022). [1] Developed by authors based on data collection from the NMEC.

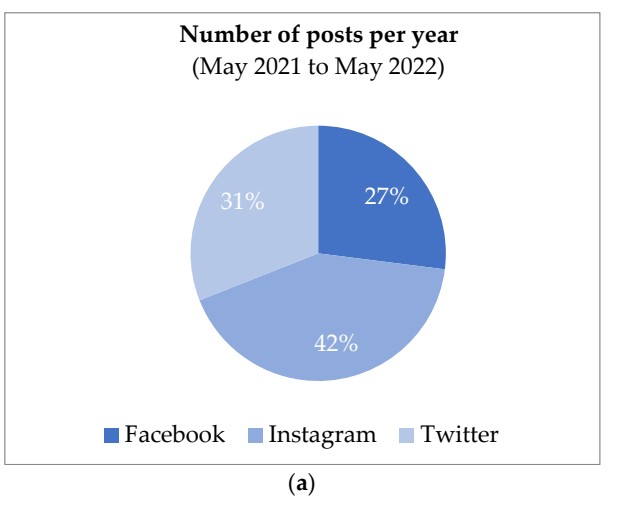

(**a**)

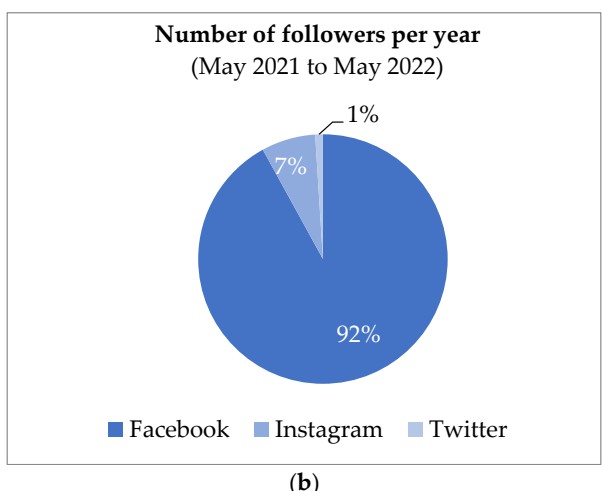

(**b**)

**Figure 28.** Word counts of the NMEC on social media engines–Facebook, Instagram, and Twitter. (**a**) Number of posts counted per year on social media; (**b**) number of followers per year on social media.

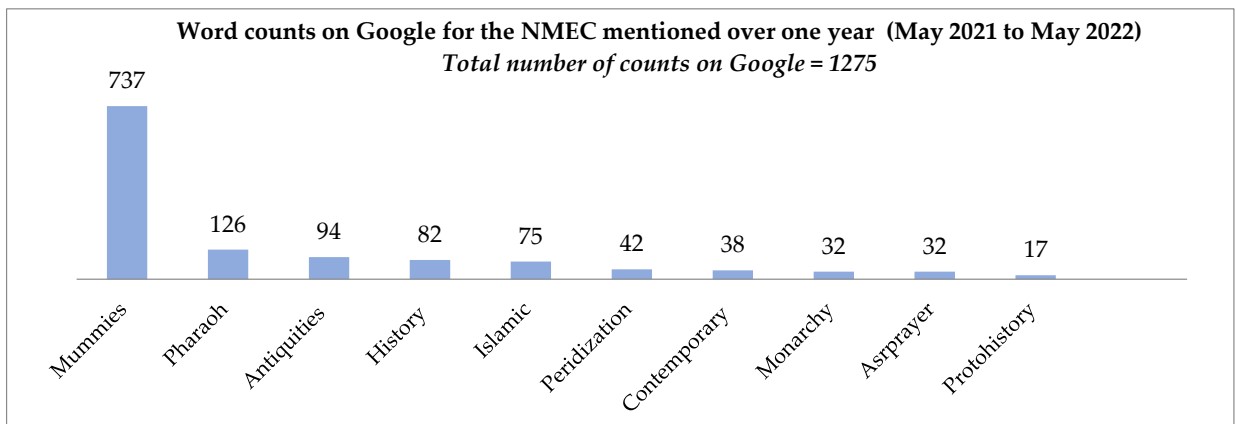

**Figure 29.** Word counts on Google by category of NMEC-exhibited objects.

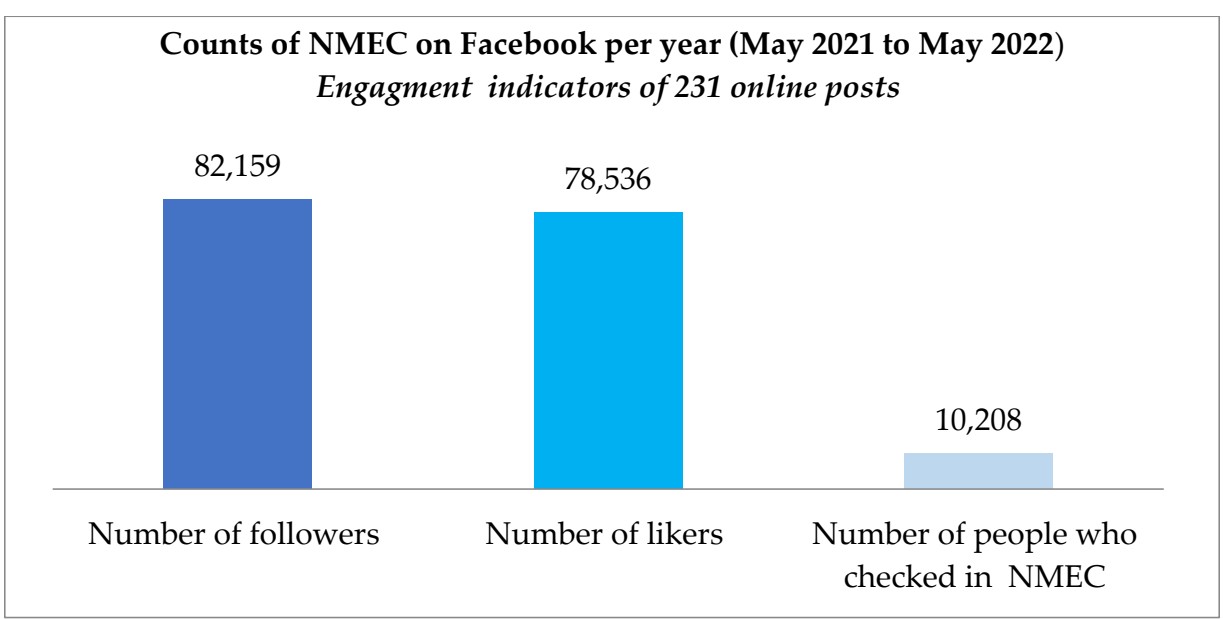

**Figure 30.** Comparison of the social media of the NMEC over one year.

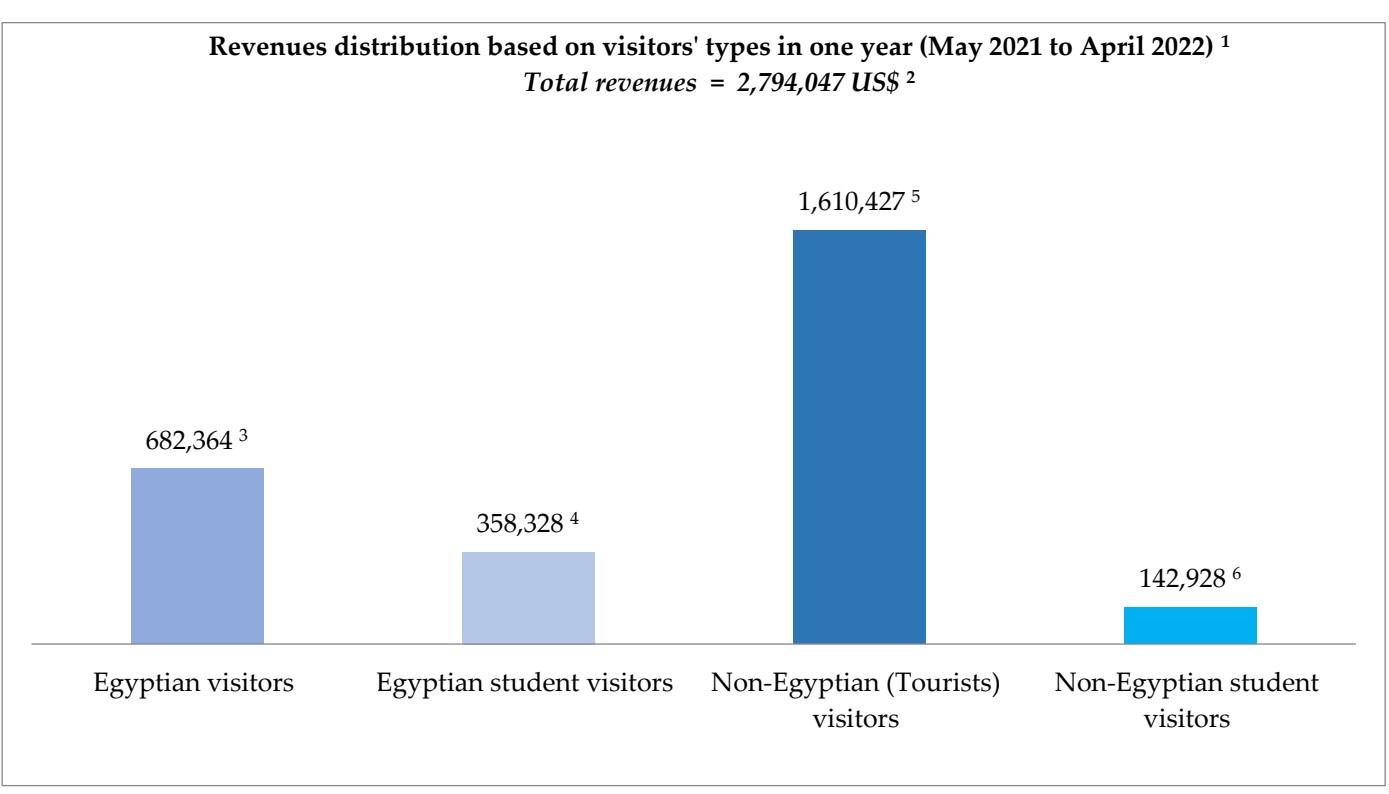

**Figure 31.** Total revenues and distribution by visitor types to NMEC during one year (2021–2022) [1,2,3,4,5,6]. [1] Developed by authors based on data collected from the NMEC. [2] Equivalent to EGP 12,473,610 as per the Central Bank of Egypt (CBE) exchange rate on 28 April 2022. [3] Equivalent to EGP 6,550,230 as per the CBE exchange rate on 28 April 2022. [4] Equivalent to EGP 29,438,610 as per the CBE exchange rate on 28 April 2022. [5] Equivalent to EGP 2,612,730 as per the CBE exchange rate on 28 April 2022. [6] Equivalent to EGP 51,075,180 as per the CBE exchange rate on 28 April 2022.

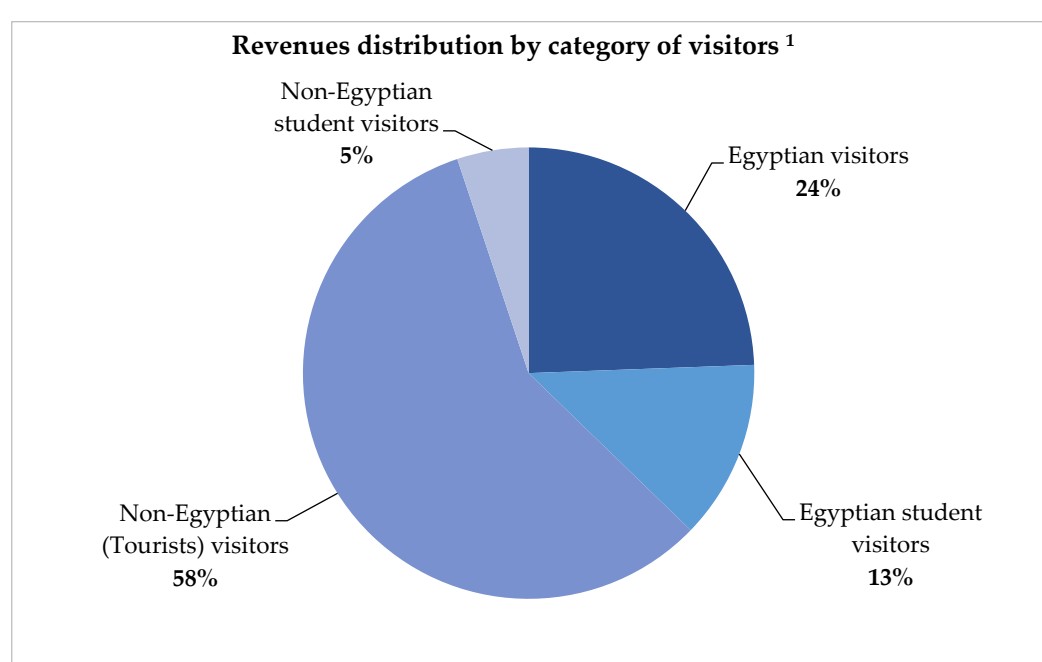

**Figure 32.** Percentage of revenue by visitor types to the NMEC during one year (2021–2022). [1] Developed by authors based on data collected from the NMEC officials.

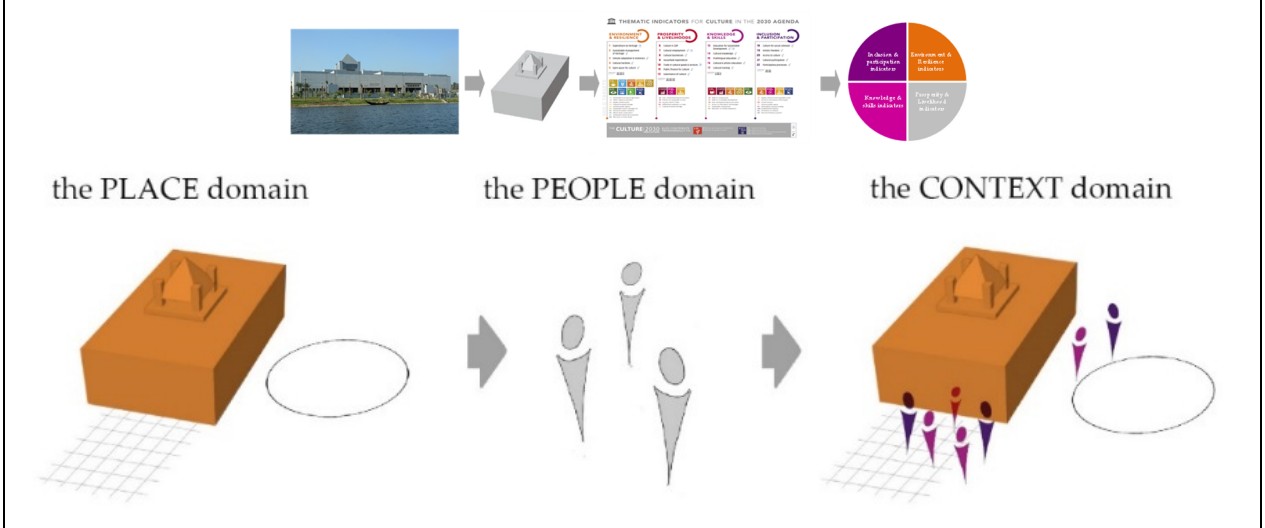

**Figure 33.** The relationship between the place and the people issue in the cultural sustainability assessment of the NMEC.

**Table 8.** Word counts and Google ranking of the NMEC social and cultural impacts on social media using data scraping.

| No. | Measured Parameter (Word Counts) [1] | Facebook | Instagram | Twitter | Google |
|-----|---------------------------------------|----------|-----------|---------|--------|
| 1 | Number of posts per year | 231 | 350 | 258 | - |
| 2 | Number of followers | 82,159 | 5848 | 1247 | NA |
| 3 | Number of likes | 78,536 | NA | NA | NA |
| 4 | Number of people who checked in social media | 10,208 | NA | NA | NA |
| 5 | Number of viewers–Google | NA | NA | NA | 9003 |
| 6 | Ranking | - | - | - | 4.7 |

[1] Word counts were conducted over one year from May 2021 to May 2022.

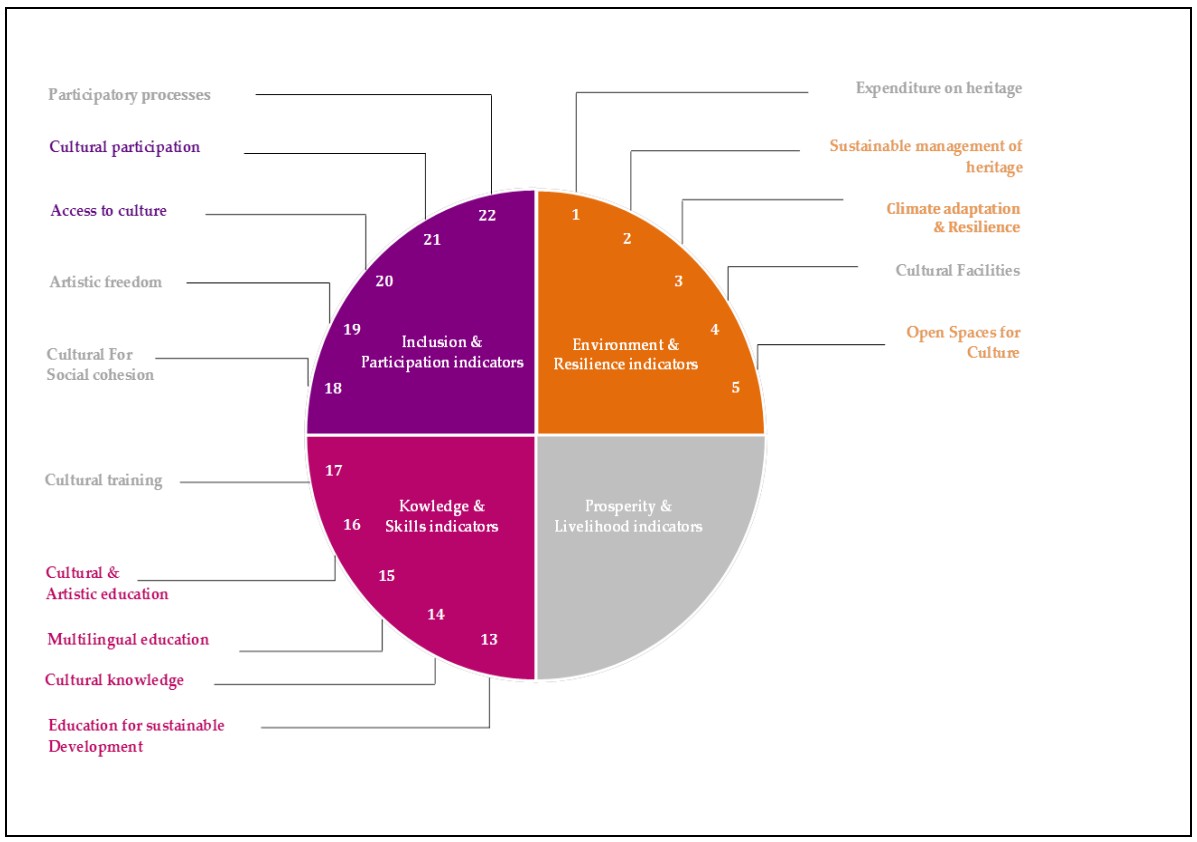

**Figure 34.** The match between the three NMEC domains and the UNESCO indicators in the cultural sustainability assessment.

**Table 9.** The indicators applied to evaluate the museum to measure the sustainability dimension (developed by authors).

| No. | Indicator | Optimum | Sustainability Dimension | Value [1] |
|---|---|---|---|---|
| 1 | State of conservation: the proportion of objects that are conserved perfectly (%) | Maximum | Cultural (collection storage, conservation, and research) | 99% |
| 2 | Storage conditions: the proportion of collections stored in appropriate conditions (%) | | | 95% |
| 3 | The degree of heritage research: the proportion of objects pertaining to which the documentation is complete (%) | | | 50% [2] 95% [3] |
| 4 | Meeting the microclimate conditions: the number of days in which there were no deviations [4] from the optimum microclimate parameter/365 (%) | | | 100% |
| 5 | The degree of exhibiting: (the number of objects exhibited (2000) per year × the number of days of exhibition/(the number of heritage objects (1952) × 365) (%) | Maximum | Social and cultural | 1.03% |
| 6 | Online visibility: number of mentions of the museum on Google (9003)/ Number of objects in the collection (2000) | | | 4.502 |
| 7 | Extension outside the main premises: (number of objects exhibited outside the main premises × number of days of exhibition)/(total number of objects not exhibited at the main premises × 365) (%) | | | 0.0% None |

**Table 9.** *Cont.*

| No. | Indicator | Optimum | Sustainability Dimension | Value [1] |
|---|---|---|---|---|
| 8 | Extension on the market: percentage of the number of objects lent to other institutions from the total of objects not exhibited (%) | Maximum | Social and Economic | 0.00% None |
| 9 | Events organized annually | | | 95% |
| 10 | Attractiveness of the museum's collections: Number of visitors/objects exhibited | | | 393.97 |
| 11 | Assessment of the museum by visitors: number of positive impressions (78,536)/total number of impressions written by visitors in the guestbook (2800), in a year (%) | | | 28.05% |
| 12 | Ratio of young and old staff: ratio of staff who are in the first 10 years of their career (%) | | | 60% |
| 13 | Online accessibility of the collection: number of objects (2000) expressed in as percentage from the whole collection published online (85) (%) | | | 4.5% |
| 14 | Labor productivity 1: number of events/number of staff | Maximum | Social and Economic | 0.198 |
| 15 | Labor productivity 2: number of temporary exhibitions/Staff | | | 0.010 |
| 16 | Labor productivity 3: number of visitors/number of staff | | | 1638.1 |
| 17 | Labor productivity 4: number of online unique visitors per year/staff (481) | | | 5.83% |
| 18 | Efficiency of the financial resource use: number of visitors/total expenses | | | 0.0163 |
| 19 | Efficiency of material resource use: number of visitors/exhibition area | | | 187.6 |
| 20 | Attractiveness of museum's electronic resources: number of online unique visitors yearly (89,254)/number of posts (839) | | | 106.38 |
| 21 | Voluntary involvement: number of hours worked by volunteers per year [23]/number of volunteers | Minimum | Natural environment (using the resources as efficiently as possible) | 252 |
| 22 | Capital productivity: earned income/total objects exhibited per year | | | 44,606.1 |
| 23 | Consumption of financial resources: objects exhibited per year/total expenses 1200/48,192,060 ($2.49 \times 10^{-5}$) | | | 0.000025 |
| 24 | Consumption of electricity: consumption of electricity/museum's area | | | 116.99 kWh/yr/m$^2$ |
| 25 | Consumption of thermal energy: consumption of thermal energy/museum's area | | | N/A |
| 26 | Consumption of water: consumption of water/(staff + volunteers + unique visitors at the main premises) 6727 L/month 2020 [5] | | | [6] |
| 27 | Consumption of consumables: expenses for consumable office materials (ink cartridges, stationery, files, etc.) and number of staff and volunteers (481 + 40) = 521 | | | 1898.19 |
| 28 | Consumption of fuel: consumption of fuel/number of external events | | | N/A |
| 29 | Labor productivity 5 (value): earned income/number of staff per year) | | | 5812.01 |
| 30 | The capacity of self-financing: earned income/total income (%) | | | 95.42% |

**Table 9.** *Cont.*

| No. | Indicator | Optimum | Sustainability Dimension | Value [1] |
|---|---|---|---|---|
| 31 | Quantitative liquidity of merchandise (calculated individually for each type of product): annual merchandise sales (pcs.)/quantities manufactured or purchased (pcs.) (%) | | | 29.67% |
| 32 | Value liquidity of merchandise: annual income from merchandise sales/value of stored merchandise | Maximum | Economic (efficiency, economic impact on the community) | 0.054 |
| 33 | Correlation between the number of tourists from the city or area and the number of visitors of the museum (this measures the museum's contribution to the economic development of the area) | | | 67.80% |

[1] Data collected from NMEC. [2] Documented. [3] Electronic documentation. [4] No deviation–NMEC is opened 365 days. [5] Water is estimated as per the reading record in 2020. [6] Water data was measured before the official opening (3 April 2021). x means multiply by, / means divided by.

*7.1. Environmental Sustainability Results*

Results of the annual energy consumption show that 79% is used in the ground floor (main museum hall), where 2000 objects are displayed, while the mummies' floor consumes 21% (Figure 20). The annual energy consumption of the NMEC per service for the ground floor is 112,242.40 kWh, 146,956.03 kWh, and 131,148.03 kWh, representing the lighting, air conditioning, and appliances, respectively. For the mummies' floor, these are about 11,860 kWh, 77,345 kWh, and 11,824 kWh, as depicted in Figure 21.

Based on the energy audit in Appendix A—Tables A2 and A3, the annual energy consumption of the NMEC opened parts–the ground floor (Figure 5a) and the mummies' floor (Figure 5b) are estimated for 491.38 MWh (491,375.98 kWh), and the total average energy consumption/m$^2$ is 20.24 kWh/m$^2$. The annual energy consumption for the main hall is 390,346.46 kWh (390.35 MW), while it is 101,029.54 kWh (101.030 MW) for the mummies' floor. The total annual energy consumption per square meter is 19.98 kWh/m$^2$ and 144 kWh/m$^2$ for the ground floor and the mummies' floor, respectively. In addition, Figure 22 shows the annual energy use for the museum's building per service type.

*7.2. Social Sustainability Results*

Table 4 and Figures 23–27 illustrate the number of visitors to the NMEC over one year from April 2021 to March 2022, the distribution of visitors' by type per month, the plotted number of Egyptian visitors, the total number of visitors of the NMEC over one year, as well as the number of non-Egyptian (tourists) visitors to the total number of visitors of the NMEC during one year, in addition to the percentage of visitors' types to the NMEC (2021–2022). In addition, the result of the word counts of the NMEC social and cultural impacts on social media (Facebook, Instagram, Twitter, and Google ranking) by virtue of the data scraping technique are shown in Figures 28–30 and Table 8.

*7.3. Economic Sustainability Results*

The results of the economic sustainability assessment of the NMEC are shown in Figure 31 to Figure 32. The total revenues generated by the NMEC during the study period of (April 2021 to March 2022) have reached USD 2,794,047 (equivalent to EGP 51,075.180). The number of Egyptian visitors represents nearly 67% of the total number of visitors, while the non-Egyptian visitors represent nearly 33% of the total visitors; it was found that the revenues generated by the non-Egyptian visitors represent nearly 62% of the total revenues generated, while the Egyptian visitors represent nearly 38% of the revenues generated.

By analyzing the fees per person, it was found that the fees for Egyptian visitors are highly subsidized by the administration to support and enhance the cultural aspect among local visitors. As shown in Figure 31, the revenues generated by Egyptian visitors are USD

682,364 and by Egyptian students are USD 358,328, resulting in the revenues generated by Egyptians being USD 1,040,692.

The revenues generated by non-Egyptian visitors reached USD 1,610,427, and the non-Egyptian students' revenues reached USD 142,928, totaling the revenues generated by the non-Egyptian, USD 1,753,355. In terms of economic sustainability and growth, it is highly recommended to compare the results shown in this research with the future revenue streams of 2022/2023 to cover the development in this aspect in future research.

### 7.4. Culture Sustainability Results

From the analysis of the 15 indicators selected to assess the case study, it emerges that those concerning the NMEC are distributed in various points of the three dimensions: A-Environment and resilience; C-Knowledge and Skills; D-Inclusion and Participation. For the assessment of the NMEC's cultural sustainability, Figure 33 presents the results of the culture assessment concerning the relationship between the place and the people issues, while Figure 34 illustrates the match between the three NMEC' domains and the UNESCO indicators in the cultural sustainability assessment.

### 7.5. Sustainability Dimensions and Indicators' Values

The assessment of the indicators in terms of the sustainability dimensions and optimum range (maximum and minimum) as well as the values, were calculated based on the 33 indicators (Table 9). For example, indicator no. 5–The degree of exhibiting: (the number of objects exhibited (2000) per year x the number of days of exhibition/(the number of heritage objects (1952) x365) (%) and so on for the other indicators, as described in the table under Indicator). Applying and calculating these indicators were based on data obtained through interviews at NMEC; only two indicators were not applied due to non-applicability; these are indicator 25–Consumption of thermal energy: Consumption of thermal energy/museum's area; and indicator 28–Consumption of fuel: Consumption of fuel/number of external events. Hence, the measured indicators' values are only 31, as shown in Table 9. The results indicate that the museum has a scored significantly high level of sustainability across different indicators. The NMEC successfully scored 99% in the first indicator that shows a high level of the state of conservation for the objects. Furthermore, it scored 95.67% in the 30th indicator that studies the capacity of self-financing, which indicates the good progress of NMEC towards economic sustainability.

## 8. Discussion

It is apparent from Figure 20 about the energy audit results that the annual energy consumption of the NMEC is near the electricity consumption rates obtained from the NMEC administration (estimated at 390,346.46 kWh (390.35 MW) for the main hall plan, while it is 101,029.54 kWh (101,029 MW) for the mummies' floor, which includes 17 mummies' rooms. It is also clear that the energy use for cooling is 76% and 38% in the mummies' floor and ground floor, respectively (almost double the ground floor (101,030 kWh/year) and of the mummies' floor (77,345.3 kWh/year)). The energy consumption for lighting in the ground floor (29% of the total) is almost 2.5 higher than in the mummies' floor (12% of the total). Nonetheless, the energy used for the equipment and apparatus in the ground floor (33% of the total) is almost 2.75 higher than that in the mummies' floor (12% of the total). This is related to the preservation conditions needed for the 17 mummies. The difference in the energy consumption per service on the ground floor (HVAC 76%, lighting 12%, and equipment and apparatus 12%) and the mummies' floor (Figure 22) is related to the entrance and area of the main exhibition hall and its ceiling height (more than 10 m), where objects are displayed with special lighting to illuminate the exhibition area and objects, as shown in Figures 7–9, 12 and 19.

In addition, it is clear that the annual energy consumption for each examined floor per square meter in the NMEC building is 19.98 kWh/m$^2$ and 144 kWh/m$^2$ for the ground floor and the mummies' floor, respectively (Figure 35). When compared to the RIBA benchmark

performance of public galleries and museums [55], such values reflect near energy-efficient floors that are well below 50 kWh/m² (good) and above 70 kWh/m² (typical), as seen in Figure 35. Only the mummies' floor consumes annual energy of 144 kWh/m², which is above the RIBA typical range, but it is justified.

In terms of social sustainability, it is clear from Figure 26 that over one year, from Apr 2021 to March 2022, the number of total visitors recorded is 613,199 persons. This number is quite significant for one year amid the COVID-19 pandemic and its related measures. Nonetheless, this number could be increased significantly post the age of COVID-19. By analyzing the number of visitors to the NMEC in the first year of operation, it is shown that the number of Egyptian visitors is higher, with 67% of the total, whereas the non-Egyptians constitute 33% of the total. The most important observation is that Egyptian students are higher compared to other types of visitors (Figures 23–25).

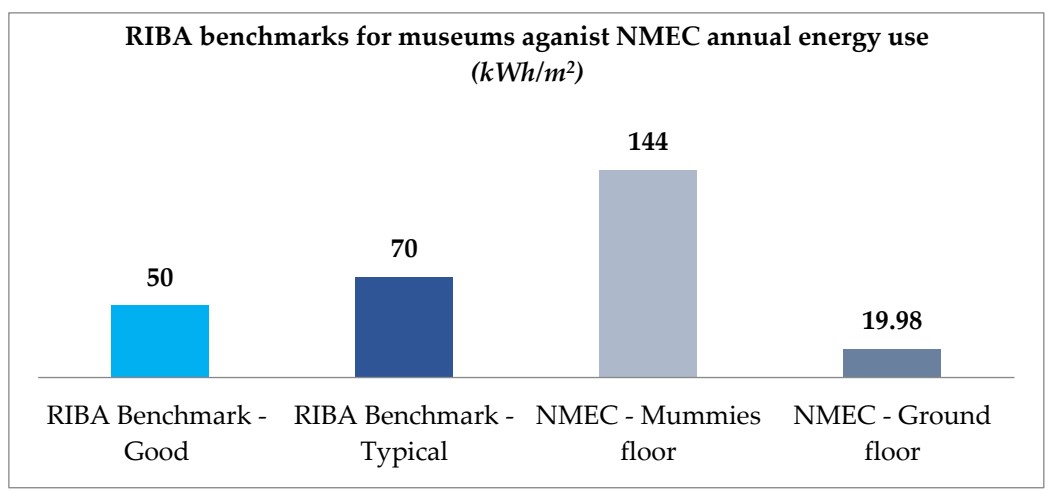

**Figure 35.** Mapping the NMEC energy consumption with RIBA benchmarks.

Results also indicate that the word counts of the NMEC on social media such as Facebook, Google, Instagram, and Twitter in the age of COVID-19 (May 2021 to May 2022) are relatively positive. It is clear from Figure 28 that the number of posts is 27%, 31%, and 42% on Facebook, Instagram, and Twitter, respectively. The highest is 42% on Instagram, followed by that on Twitter and Facebook (Figure 28a). It could be however ten times higher on normal days, i.e., the post-COVID age. However, the number of followers on social media due to the impact of the NMEC's bold opening after the Royal Parade, which was performed on 3 April 2021, was significant. Figure 28b illustrates the number of followers on Facebook, Instagram, and Twitter is 92%, 7%, and 1%, respectively. Figure 29 also presents the word counts on Google (1275 counts) by category of objects exhibited at NMEC. The record from the data scraping indicates that the number of posts from May 2021 to May 2022 is 231, 350, and 258 on Facebook, Instagram, and Twitter, respectively, while the highest number of posts is on Instagram at 350. It is also seen that the total counts on Facebook are 82,159, while it reached 5848 and 1247 on Instagram and Twitter, respectively. This shows that the highest number of records is on Facebook. The number of people who checked in through Facebook during their visit to the NMEC reached 10,802 over one year, whereas the number of reviewers on Google reached 9003 for the same period. It is clear from Figure 29 that the mummies' and pharaohs' objects recorded the multitude of 737 and 126 mentioned, followed by antiquities and artifacts corresponding to 94 and 82. As shown in Figure 30, the followers and likes are the highest counts on Facebook, reaching 82,159 and 78,536.

Community participation, an element of value beyond visits by school children, it is vital to highlight the term the creative economy (CE), also known as the orange economy (OE), a term coined by UNESCO and ICOM, and it refers to the sectors of the economy "whose main purpose is the production or reproduction, promotion, dissemination and/or

the marketing of goods, services, and activities that have cultural, artistic or patrimonial content". The CE refers to a wide range of sectors, and one of the main sectors that apply that is the tourism industry. Creative tourism, or orange tourism, is the type of tourism that allows visitors to have the opportunity to discover and further develop their creativity via participating in different experiences, which are shaped by the characteristics of the destination where they are taken [56]. The NMEC has demonstrated the application of creative tourism by providing different workshops and open sessions that provide cultural aspects and allows visitors and participants to engage in different activities that develop their creativity.

Comparing the social sustainability results with Yung et al.'s model of social benefits and Christer and Akram's social development framework, it is clear that the findings are almost aligned with both models, mainly: social interaction and network; local culture and history; community participation, accessibility, and affordable entrance fees throughout the year. As for the economic results, from the charts displayed in Figures 31 and 32, it is apparent that revenues from non-Egyptian (tourists) visitors are the highest; it is the source of 58% of the total amount of revenues in one year (2021–2022), followed by revenues from Egyptian visitors and Egyptian students, constituting 24% and 13%, respectively, while revenues from non-Egyptian students are considered to be the lowest, as it constitutes only 5% of the total amount of revenues. Moreover, the total revenues amount to USD 2,794,047 (about USD 2.8 million), which is considered to be a proper result for the first year of operations, especially during the COVD-19 pandemic. However, we highly recommend further actions to attract more non-Egyptian students through partnerships with different international cultural and educational organizations.

The actual findings of the NMEC's cultural sustainability assessment implicate tentative decision-making support to museum management to address future actions in the current weak directions. Although the analysis was carried out on the basis of the UNESCO set and adapting it to our case study, the NMEC cultural sustainability assessment showed a positive response to the evaluation criteria for 9 out of 15 indicators (Figure 34), distributed as follows:

DIMENSION A: indicators 2, 3, 5;
DIMENSION C: indicators 13, 14, 15, 16; and
DIMENSION D: indicators 20, 21.

The dimension on which NMEC ranks with the highest correspondence is the C-Knowledge and Skills dimension, followed by the A-Environment and Resilience dimension and the D-Inclusion and Participation dimension. The conceiving and management of the NMEC cultural project, therefore, highlights good cultural sustainability, which is based on many dimensions of reverberation of its presence and the way of relating its cultural heritage with society, communities, and the environment.

As stated above, in this article, the NMEC's cultural sustainability assessment has been examined, testing for the first time the use of the UNESCO set of indicators on the building, the surroundings, and the activities of the museum addressing the communities' involvement. From the perspective of previous studies, the results of the NMEC's cultural sustainability assessment conducted in this way represent the first attempt to overcome the previously specifically designed matrix for museums, but it is outdated.

This study also shows that the methodology needs to be better calibrated in the future by adapting to the scale of the building and of the urban district. There are 22 indicators conceived for the assessment of urban contexts. Future studies should be conducted in this direction and applied by focusing on the segmentation of the case study. This should address museums with similar sizes and characteristics. It is clear from Table 9 that it was possible to measure the indicators of sustainability dimensions, but only a few were not applicable, such as indicators n. 25 and n. 28. However, we attempted to measure the maximum and minimum values of the five sustainability dimensions listed in Table 9, but since the NMEC was opened to the public in April 2021, i.e., less than two years ago, there is no historical data to measure the progress of the museum towards the sustainability

level across multiple years, as recommended in the model used in this paper developed by Izabela L Pop and Anca Broza.

Based on the results of the sustainability assessment of the NMEC in terms of environmental, social, economic, and cultural analysis, it is clear that the NMEC can be aligned with museums' development and conditions (Table 2), and the ICOM and OCED guide for museums, especially the economic development and innovation in terms of cultural services inside and outside the museum that attracted 613,210 visitors, of which 201,974 visitors were tourists and 411,236 were local visitors for the first year after opening in April 2021. In addition, it can be said that the NMEC became facilitators of knowledge and creativity by creating opportunities for artists, entrepreneurs, designers, and craftsmen to display and access the collection, as shown in Figures 12, 14, and 18, as well as a recent concert by the singer Nisma Mahgoub, held on 10 September 2022. For the second point of the guide—urban regeneration and community development—it is clear from the sustainability assessment of the NMEC that the museum is considered a focal place in the urban design of the district and its cultural fabric.

It is clear that the NMEC developed activities that contribute to social capital. This is seen in the network created among the antiquities-restoration professionals and archaeologists through the recent exhibition organized by the NMEC in collaboration with the Embassy of Spain in Cairo, featuring antiquities-inspection tours to Upper Egypt and the archive of Eduardo Toda held in the period between 14th and 28th of September 2022. In addition, the Mexican experience features a glance at the intangible cultural heritage of Mexico in order to foster the cultural understanding of the social capital between the two nations (22 September 2022). As for becoming a center of a creative district, the NMEC stands as a creative hub for economic, social, and cultural activities. Finally, the NMEC supports eco-friendly initiatives, such as recycling and vending machines located inside the main lobby of the museum, as well as having recently conducted its initiative on carbon footprint.

In terms of the cultural development, education, and creativity aspects, the NMEC conducted many school visits (Section 5.2.2) and education activity development (Section 5.2.3) to educate and increase the knowledge of schools and university students about Egyptian civilization. This could be seen as a contribution to cultural and educational development as a source of inductive and reflective knowledge, not only for Egyptian youth but also for tourists. It can be stated that the NMEC presentation and interpretation of collections (50,000 objects, of which 2000 antiquities and artifacts are displayed in the main hall and the mummies hall, as well as the interactive screens (Figures 7–9, 12 and 19) support in greater extent creative skills to all visitors. By looking at the themes of museum sustainability listed in Figure 2, it is clear that the NMEC sustainability results are aligned with almost 19 out of 21 themes.

By comparing the NMEC with recently open museums globally in terms of area and objects, as shown in Table 10, it is clear that the NMEC stands unique, though its currently displayed 2000 objects out of 50,000 objects compared with other six museums [57]. The comparison of the NMEC with the other six global museums recently open is shown in Figure 36, while the comparison of the collection (objects) is presented in Figure 37. It is clear from the comparison that the NMEC land area and museum area are bigger than the global museums recently open but smaller than the Grand Egyptian Museum (GEM). In terms of objects, the NMEC (50,000) comes third, compared with the Art Gallery of New South Wales in Sydney, Australia; the National Museum of Norway in Oslo (400,000); and the Museum of Art & Photography in Bengalore, India (60,000), and it comes second locally after the GEM (100,000).

**Table 10.** Museums recently open globally.

| Item Museum | City, Country | Open Year | Land/Museum Area (m²) | Collection | Objects/Artifacts |
|---|---|---|---|---|---|
| Istanbul Museum of Modern Art | Istanbul, Turkey | January 2022 | 150,003,300 | Modern arts | 12,000 |
| Art Gallery of New South Wales | Sydney, Australia | October 2022 | 27,871 15,980 | Australian, European, and Asian fine arts | 36,000 |
| National Museum of Norway | Oslo, Norway | June 2022 | 54,600 14,000 | Norwegian Baldishol tapestry | 400,000 |
| Hong Kong Palace Museum | Beijing, China | July 2022 | 30,000 7800 | Rare books, traditional calligraphy & treasures | 914 out of 1.86 mL. works |
| Museum of Art & Photography | Bengalore, India | December 2022 | 5500 4087 | Arts, (physical and digital) photography, antiquities | 60,000 |
| Grand Egypt Museum | Giza, Egypt | November 2022 | 480,000 81,000 | Ancient Egypt Antiquities and artifacts | 100,000 |
| National Museum of Egyptian Civilization | Cairo, Egypt | April 2021 | 135,000 34,291 | Ancient Egypt Antiquities and artifacts | 500,002,000 |

(Source Developed by authors after [57]).

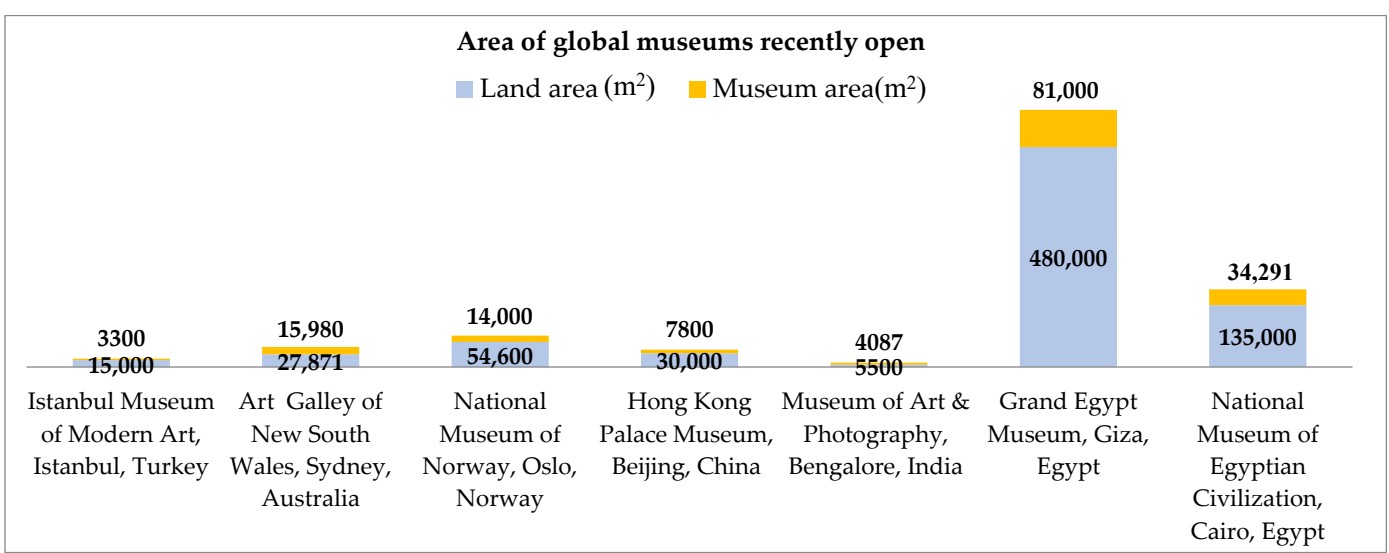

**Figure 36.** Comparison between the NMEC areas and the six global museums recently open in 2022.

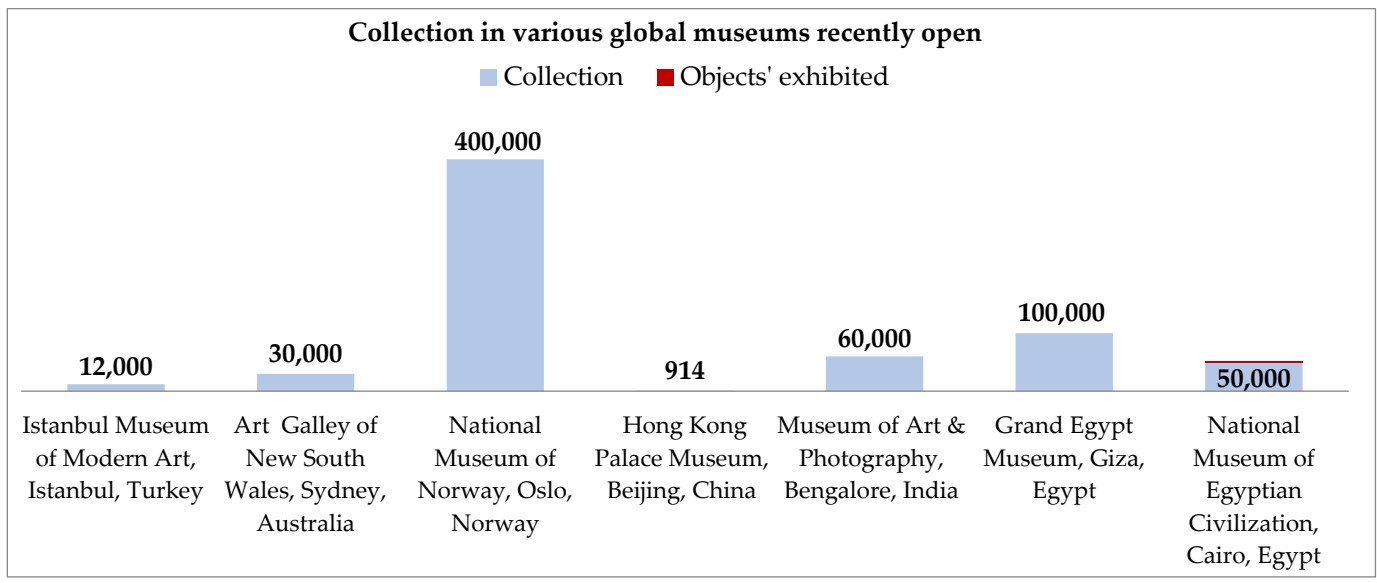

**Figure 37.** Comparison between the NMEC objects and the six global museums recently open in 2022.

## 9. Conclusions

The assessment of sustainability inducted after the opening of the National Museum of Egyptian Civilization (NMEC) yielded significant results. Environmental, social, economic, and cultural sustainability were studied. Both qualitative and quantitative approaches were successful in mapping the four sustainability pillars: environmental, social, economic, and cultural. The exploitation of 33 indicators to measure five sustainability dimensions in addition to the UNESCO 15 Thematic Indicators for Culture in the 2030 Agenda, wherein nine indicators only were met in the culture assessment of the NMEC. The sustainability assessment of the NMEC relates to the ICOM and OECD Guide for museums. For a better assessment of the environmental sustainability and economic sustainability, more yearly records are needed, since the NMEC has only one year of records since opening on 3 April 2021. It has been proven that sustainability assessment play a key role in assuring livable and regenerative cities. In addition, this research has significant importance in creating the first milestone in measurement by providing clear values for each indicator. The values defined in this paper will pave the road for further research in the future to create comparison studies across different years and building similar research covering different museums in Egypt to produce an analysis of the average values of the indicators in museums nationwide.

**Author Contributions:** Conceptualization, M.A., P.P. and D.E.; data curation, M.A., D.E. and M.B.; formal analysis, M.A. and P.P.; investigation, M.A., D.E. and M.B.; methodology, M.A., P.P. and D.E.; project administration, M.A.; resources, M.A.; software, M.A.; supervision, M.A.; validation, M.A., P.P., D.E. and M.F.; visualization, M.A. and P.P.; writing–review and editing, M.A. and M.F. All authors have read and agreed to the published version of the manuscript.

**Funding:** This research received no external funding.

**Institutional Review Board Statement:** Not applicable.

**Informed Consent Statement:** Informed consent was obtained from all subjects involved in the study.

**Data Availability Statement:** All of the data shown in this study have been submitted in the form of data tables embedded in the research article. All photos are the property of the authors. Also, any further data regarding this study are available upon request from the corresponding author.

**Acknowledgments:** Authors would like to thank all who provided assistance and support to complete this research work. In this context, we expressed our sincere gratitude and heartfelt appreciation to Ahmed Ghoniem, Head of Authority and Executive Director of the National Museum of Egyptian Civilization (NMEC), for all his benevolent support and facilitating of the visits, interviews with NMEC administration, and data collection; without it, this research would not be completed. Special thanks to NMEC staff, namely Sayed Abou-Elfadl and the Public Relations Department–NMEC, for their interview held at the end of December 2021. Sincere thanks go to Ahmed Elbayoumi, Director of Finance & Administration–NMEC, for the meeting and support in providing the data to calculate the sustainability indicators and facilitating several meetings with other NMEC staff. We also thank Aayat Soliman, Manager of Heritage Objects Storage–NMEC, and Iman Saad, Manager of Wood Heritage objects–NMEC. In addition, we would like to thank Mohamed Azzam, Manager of Information Center–NMEC, and Amr Fouad, Software Developer–NMEC, for their support in providing the word count on social media about NMEC. Moreover, we appreciate the effort and support of architect Dina Yasser, Department of Architecture, Faculty of Engineering, Cairo University, for her assistance in conducting the assessment of the urban heat island effect (UHIE) for the NMEC urban area depicted in Figures 10b,c and 11. Last but not least, we express our thanks to Yasmina Ragab and Mohamed Shahwan–teaching assistant at the Department of Architecture, Faculty of Engineering at Cairo University; and Sherifa Elhaggan for their great effort in proofreading this article. Finally, we convey our appreciation to Farid Fadel, artist, for his insight during the temporary exhibition and for providing the opportunity to tour his great artifacts at the NMEC event in December 2021.

**Conflicts of Interest:** The authors declare no conflict of interest. The authors also declare that there are no personal circumstances or interest that may be perceived as inappropriately influencing the representation or interpretation of reported research results. In addition, this research work received no funds from any funding authorities in Egypt or abroad. Hence, there was no role for the funders

in the design of the study, in the collection, analysis, or interpretation of data, in the writing of the manuscript, or in the decision to publish the results.

## Appendix A

**Table A1.** Indicators for measuring museums' sustainability according to Izabela Luiza Pop and Anca Borza [1].

| No. | Indicator | Optimum | Sustainability Dimension |
|---|---|---|---|
| 1 | State of conservation: the proportion of objects conserved perfectly (%) | Maximum | Cultural (collection storage, conservation, and research) |
| 2 | Storage conditions: the proportion of collections stored in appropriate conditions (%) | | |
| 3 | The degree of heritage research: the proportion of objects pertaining to which the documentation is complete (%) | | |
| 4 | Meeting the microclimate conditions: the number of days in which there were no deviations from the optimum microclimate parameter/365 (%) | | |
| 5 | The degree of exhibiting: (the number of objects exhibited per year x the number of days of exhibition/(the number of heritage objects x365) (%) | Maximum | Social and cultural |
| 6 | Online visibility: number of mentions of the museum on Google/number of objects in the collection | | |
| 7 | Extension outside the main premises: (number of objects exhibited outside the main premises x number of days of exhibition)/(total number of objects not exhibited at the main premises x365) (%) | | |
| 8 | Extension on the market: the percentage of the number of objects lent to other institutions from total of objects not exhibited (%) | Maximum | Social and economic |
| 9 | Events organized annually | | |
| 10 | Attractiveness of the museum's collections: number of visitors/objects exhibited | | |
| 11 | Assessment of the museum by visitors: No. of positive impressions/total number of impressions written by visitors in the guestbook, in a year (%) | | |
| 12 | Ratio of young and old staff: ratio of staff who are in the first 10 years of their career (%) | | |
| 13 | Online accessibility of the collection: number of objects expressed as a percentage from the whole collection published online (%) | | |
| 14 | Labor productivity 1: number of events/number of staff | Maximum | Social and economic |
| 15 | Labor productivity 2: number of temporary exhibitions/staff | | |
| 16 | Labor productivity 3: number of visitors/number of staff | | |
| 17 | Labor productivity 4: number of online unique visitors per year/staff | | |
| 18 | Efficiency of the financial resource use: number of visitors/total expenses | | |
| 19 | Efficiency of material resource use: number of visitors/exhibition area | | |
| 20 | Attractiveness of museum's electronic resources: number of online unique visitors yearly/number of posts | | |
| 21 | Voluntary involvement: number of hours worked by volunteers per year/number of volunteers | Minimum | Natural environment (using the resources as efficiently as possible) |
| 22 | Capital productivity: earned income/total objects exhibited per year | | |
| 23 | Consumption of financial resources: objects exhibited per year./total expenses | | |
| 24 | Consumption of electricity: consumption of electricity/museum's area | | |
| 25 | Consumption of thermal energy: consumption of thermal energy/museum's area | | |
| 26 | Consumption of water: consumption of water/(staff + volunteers + unique visitors on the main premises) | | |
| 27 | Consumption of consumables: expenses for consumable office materials (ink cartridges, stationery, files, etc.)/number of staff and volunteers | | |
| 28 | Consumption of fuel: consumption of fuel/number of external events | | |

**Table A1.** *Cont.*

| No. | Indicator | Optimum | Sustainability Dimension |
|---|---|---|---|
| 29 | Labor productivity 5 (value): earned income/number of staff | Maximum | Economic (efficiency, economic impact on the community) |
| 30 | The capacity of self-financing: earned income/total income (%) | | |
| 31 | Quantitative liquidity of merchandise (calculated individually for each type of products): annual merchandise sales (pcs.)/quantities manufactured or purchased (pcs.) (%) | | |
| 32 | Value liquidity of merchandise: annual income from merchandise sales/value of stored merchandise | | |
| 33 | Correlation between the number of tourists from the city or area and the number of visitors of the museum (this measures the museum's contribution to the economic development of the area) | | |

[1] Developed by authors after Izabela Luiza Pop and Anca Borza [11], x means multiply by, / means divided by.

**Table A2.** Energy audit for the NMEC main building–ground floor.

| Energy Consumption | | | | | | | | | | | | | | | | | | | | | | | | |
|---|---|---|---|---|---|---|---|---|---|---|---|---|---|---|---|---|---|---|---|---|---|---|---|---|
| | | **Working Hours** | | | | | | | | | **Lighting** | | | | | **HVAC** | | | | **Equipment & Apparatus** | | | | |
| | Zone | No. | Floor | Hrs/day (Sun-Thur) | Hrs/Day (Fri-Sat) | Evening Events 1 Day | Days/wk | Hrs/wk | Hrs/month | Hrs/yr. | Light Type | No. | Power/lamp (W) | Power (W) | Energy use (kWh) | Cooling type | No. | Power (W) | Energy use (kWh) | Security Devices | No. | Power/unit (w) | Power (W) | Energy use (kWh) |
| Ground floor | Entrance Hall & main entrance | 1 | FL01 | 10 | 14 | 7 | 78 | 312 | 3744 | | 90 LED compact Lamp spotlights / 40 LED compact Lamp spotlights | 90 / 40 | 26 / 26 | 2.340 / 1.040 | 8760.96 / 3893.76 | | | | | 3 Security cameras / 2 X-ray gates / 10 Sound speakers / 2 Wi-Fi devices | 3 / 2 / 10 / 2 | 2 / 1000 / 100 / 20 | 0.006 / 2.0 / 1.0 / 0.04 | 22.464 / 7488.00 / 3744.00 / 149.76 |
| | Side Hall 1 | 1 | FL01 | 9 | 12 | 7 | 73 | 292 | 3504 | | 40 LED Lamp spotlights | 40 | 26 | 1.04 | 3644.16 | | | | | 1 Smart display device / 12 Sound speakers / 1 Wi-Fi device / 4 Security cameras | 1 / 12 / 1 / 4 | 80 / 100 / 20 / 2 | 0.08 / 0.120 / 0.02 / 0.008 | 280.32 / 420.48 / 70.08 / 28.032 |
| | Side Hall 2 | 1 | FL01 | 9 | 12 | 7 | 73 | 292 | 3504 | | 40 LED Lamp spotlights | 40 | 26 | 1.04 | 3644.16 | Carier Central AC (chilled water) | 2 | 11,190 | 78,419.52 | 1 Smart display device / 12 Sound speakers / 1 Wi-Fi device / 4 Security cameras | 1 / 12 / 1 / 4 | 80 / 100 / 20 / 2 | 0.08 / 0.120 / 0.02 / 0.008 | 280.32 / 420.48 / 70.08 / 28.032 |
| | Staff corridor | 2 | FL01 | 9 | | | 7 | 63 | 252 | 3024 | 6 LED Lamp spotlights | 6 | 26 | 0.156 | 967.68 | | | | | 1 Security camera | 1 | 2 | 0.002 | 12.10 |
| | Admin offices | 6 | FL01 | 9 | | | 7 | 63 | 252 | 3024 | 12 LED Lamp spotlights | 12 | 26 | 0.312 | 5660.93 | | | | | 2 Desktop computers / 1 Printer | 2 / 1 | 100 / 135 | 0.2 / 0.135 | 3628.80 / 2449.44 |
| | Gift shop | 1 | FL01 | 9 | 9 | | 7 | 63 | 252 | 3024 | 128 LED Lamp spotlights / 56 LED display spotlights / 8 Track-light directed spotlights | 128 / 56 / 8 | 26 / 4 / 12 | 3.328 / 0.224 / 0.096 | 10,063.87 / 677.38 / 290.30 | | | | | 1 Cashier / 1 Desktop computer / 4 Security cameras / 32 Sound system speakers | 1 / 1 / 4 / 32 | 100 / 100 / 2 / 100 | 0.1 / 0.1 / 0.008 / 3.2 | 302.40 / 302.40 / 24.19 / 9676.80 |
| | Toilets | 6 | FL01 | 9 | 12 | | 7 | 73 | 292 | 3504 | 10 LED Lamps / 6 Small LED spotlights | 10 / 6 | 26 / 9 | 0.260 / 0.054 | 5466.24 / 1135.30 | | | | | 2 Hand dryers | 2 | 1240 | 2.48 | 52,139.52 |

**Table A2.** *Cont.*

**Energy Consupmtion**

| Zone | No. | Floor | Hrs/day (Sun-Thur) | Hrs/Day (Fri-Sat) | Evening Events 1 Day | Days/wk | Hrs/wk | Hrs/month | Hrs/yr. | Light Type | No. | Power/lamp (W) | Power (W) | Energy use (kWh) | Cooling type | No. | Power (W) | Energy use (kWh) | Security Devices | No. | Power/unit (w) | Power (W) | Energy use (kWh) |
|---|---|---|---|---|---|---|---|---|---|---|---|---|---|---|---|---|---|---|---|---|---|---|---|
| Reception hall | 1 | FL01 | 9 | 12 | 7 | 73 | 292 | 3504 | | 80 LED Lamp spotlights | 80 | 26 | 2.08 | 7288.30 | | | | | 2 Escalators<br>2 Elevators<br>16 Sound speakers<br>2 Smart display devices<br>2 LCD screens | 2<br>2<br>16<br>2<br>2 | 250<br>350<br>100<br>80<br>120 | 0.5<br>0.7<br>1.6<br>0.160<br>0.240 | 1752.00<br>2452.80<br>5606.40<br>560.64<br>840.96 |
| Cafeteria | 1 | FL01 | 9 | 9 | 7 | 63 | 252 | 3024 | | 97 LED Lamp spotlights | 97 | 26 | 2.522 | 7626.53 | | | | | 1 Display fridge<br>1 Fridge<br>1 Electric boiler<br>1 Microwave<br>1 Coffee machine<br>1 Horizontal deep freezer<br>29 Sound speakers<br>1 Cashier<br>1 Visa payment machine<br>1 Router | 1<br>1<br>1<br>1<br>1<br>1<br>29<br>1<br>1<br>1 | 350<br>350<br>1500<br>1200<br>1300<br>145<br>100<br>100<br>4<br>20 | 0.350<br>0.350<br>1.5<br>1.2<br>1.3<br>0.145<br>2.9<br>0.1<br>0.004<br>0.02 | 1058.40<br>1058.40<br>4536.00<br>3628.80<br>3931.20<br>438.48<br>8769.60<br>302.40<br>12.096<br>60.48 |
| Tube Corridor | 1 | FL01 | 9 | 9 | 7 | 63 | 252 | 3024 | | 16 LED Lamps (each 1.1m long) | 16 | 26 | 0.416 | 1257.98 | | | | | 2 Security camera | 2 | 2 | 0.004 | 12.096 |
| Tickets' Hall | 1 | FL01 | 9 | 9 | 7 | 63 | 252 | 3024 | | 75 LED Lamp spotlights | 75 | 26 | 1.95 | 5896.80 | Carier Central AC (chilled water) | 2 | 2238 | 14,824.51 | 2 Security cameras<br>8 Sound speakers<br>3 LCD screens<br>1 Cashier<br>2 Visa payment machines | 2<br>8<br>3<br>1<br>2 | 2<br>100<br>120<br>100<br>4 | 0.004<br>0.8<br>0.360<br>0.1<br>0.008 | 12.096<br>2419.20<br>1088.64<br>302.4<br>24.19 |
| Lockers' Hall | 1 | FL01 | 9 | 12 | 7 | 69 | 276 | 3312 | | 70 LED Lamp spotlights | 70 | 26 | 1.82 | 6027.84 | | | | | 8 Sound speakers<br>3 Security cameras | 8<br>3 | 100<br>2 | 0.8<br>0.006 | 2649.60<br>19.872 |
| Services' room | 1 | FL01 | 9 | | 7 | 63 | 252 | 3024 | | 12 LED Lamp spotlights | 12 | 26 | 0.312 | 943.49 | | | | | | | | | |

**Table A2.** *Cont.*

**Energy Consupmtion**

| Zone | No. | Floor | Hrs/day (Sun-Thur) | Hrs/Day (Fri-Sat) | Evening Events 1 Day | Days/wk | Hrs/wk | Hrs/month | Hrs/yr. | Light Type | No. | Power/lamp (W) | Power (W) | Energy use (kWh) | Cooling type | No. | Power (W) | Energy use (kWh) | Security Devices | No. | Power/unit (w) | Power (W) | Energy use (kWh) |
|---|---|---|---|---|---|---|---|---|---|---|---|---|---|---|---|---|---|---|---|---|---|---|---|
| Pre-main Exhibition Hall | 1 | FL01 | 9 | 12 | 7 | 69 | 276 | 3312 | 158 LED lamps spotlights | 158 | 26 | 4.108 | 13,605.59 | | | | | 2 Smart display screens<br>1 LED screen (4.4 m × 3.0 m)<br>4 Security cameras<br>7 Smart tickets scanners (gates)<br>1 Recycling vending machine | 2<br>1<br>4<br>7<br>1 | 120<br>120<br>2<br>3<br>20 | 0.24<br>0.12<br>0.008<br>0.21<br>0.2 | 794.88<br>397.44<br>26.49<br>695.52<br>662.40 |
| Main Exhibition Hall | 1 | FL01 | 9 | 12 | 7 | 69 | 276 | 3312 | 187 LED Lamp spotlights<br>26 Track-light directed spotlights<br>35 Ancient light with Florescent bulbs<br>240 LED display spotlights<br>30 LED Bar/m | 187<br>26<br>35<br>240<br>30 | 26<br>30<br>12<br>4<br>20 | 4.862<br>0.780<br>0.420<br>0.960<br>0.600 | 16,102.94<br>2583.36<br>1391.04<br>3179.52<br>1987.20 | Carier Central AC (chilled water) | 1 | 14,920 | 49,415.04 | 4 Smart display devices<br>8 LCD screens<br>6 Security cameras<br>8 Large sound speakers<br>3 Wi-Fi devices | 4<br>8<br>6<br>8<br>3 | 80<br>120<br>2<br>20<br>20 | 0.32<br>0.96<br>0.012<br>0.16<br>0.06 | 1059.84<br>3179.52<br>39.75<br>529.92<br>198.72 |

**Table A2.** *Cont.*

| Energy Consupmtion | | | | | | | | | | | | | | | | | | | | | | | | |
|---|---|---|---|---|---|---|---|---|---|---|---|---|---|---|---|---|---|---|---|---|---|---|---|---|
| Zone | **Working Hours** | | | | | | | | | **Lighting** | | | | | | **HVAC** | | | | | **Equipment & Apparatus** | | | | |
| | No. | Floor | Hrs/day (Sun-Thur) | Hrs/Day (Fri-Sat) | Evening Events 1 Day | Days/wk | Hrs/wk | Hrs/month | Hrs/yr. | Light Type | No. | Power/lamp (W) | Power (W) | Energy use (kWh) | | Cooling type | No. | Power (W) | Energy use (kWh) | Security Devices | No. | Power/unit (w) | Power (W) | Energy use (kWh) |
| Theatre | 1 | FL01 | | 4 | 1 | 4 | 16 | 192 | | 12 LED Lamp - side spotlights 72 LED Spotlights 24 Backside spotlights 5 Back spotlights 5 Corridor spotlights 3 Large directed spotlights 3 Large spotlights - Stage 16 Small corner Spotlights | 12 72 24 5 5 3 3 16 | 26 26 26 20 20 30 30 12 | 0.312 1.872 0.624 1.00 1.00 0.09 0.09 0.192 | 59.904 359.424 119.808 192 192 17.28 17.28 36.864 | | Carier Central AC (chilled water) | 2 | 11,190 | 4296.96 | 2 Sound systems 3 Microphones 8 Amplifiers 45 Sound speakers | 2 3 8 45 | 10 60 20 10 | 0.02 0.18 0.16 0.45 | 3.84 34.56 30.72 86.40 |
| Sub-total | | | | | | | | | | | | | | 112,242.40 | | Sub-total | | | 146,956.03 | Sub-total | | | | 131,148.03 |
| Total (kWh) | 390,346.46 | | | | | | | | | | | | | | | | | | | | | | | |

Table A3. Energy audit for the NMEC main building–the mummies' floor.

| | | | | | | | | | | | | | | | | | | | | |
|---|---|---|---|---|---|---|---|---|---|---|---|---|---|---|---|---|---|---|---|---|
| **Energy Consupmtion** | | | | | | | | | | | | | | | | | | | | |
| | | | **Working Hours** | | | | | **Lighting** | | | | **HVAC** | | | | **Equipment & Apparatus** | | | | |
| **Zone** | **No.** | **Floor** | **Hrs/Day** | **Days/Wk** | **Hrs/Wk** | **Hrs/Month** | **Hrs/Y** | **Light Type** | **Power/Lamp (W)** | **Power (W)** | **Energy Use (kWh)** | **Cooling Type** | **No.** | **Power (W)** | **Energy Use (kWh)** | **Security Devices** | **No.** | **Power/Unit (W)** | **Power (W)** | **Energy Use (kWh)** |
| Mummies Hall 01 | 1 | FL00 | 9 | 6 | 54 | 216 | 2592 | 7 LED spotlights<br>3 Floor Spotlight<br>3 Display LED light | 26<br>4<br>4 | 0.182<br>0.012<br>0.012 | 471.7<br>31.104<br>31.104 | | | | | 2 Security cameras<br>3 Sound speakers | 2<br>3 | 2<br>100 | 0.004<br>0.3 | 10.368<br>777.6 |
| Mummies Hall 02 | 1 | FL00 | 9 | 6 | 54 | 216 | 2592 | 22 LED spotlights<br>12 Floor Spotlights<br>10 Display LED lights | 26<br>4<br>4 | 0.572<br>0.048<br>0.040 | 1482.6<br>124.4<br>103.68 | | | | | 2 Security cameras<br>3 Sound speakers | 2<br>3 | 2<br>100 | 0.004<br>0.3 | 10.368<br>777.6 |
| Mummies Hall 03 | 1 | FL00 | 9 | 6 | 54 | 216 | 2592 | 2 LED spotlights<br>4 Display LED lights | 26<br>4 | 0.052<br>0.108 | 134.7<br>279.9 | | | | | 1 Security camera<br>1 Sound speaker | 1<br>1 | 2<br>100 | 0.002<br>0.1 | 5.184<br>259.2 |
| Mummies Hall 04 | 1 | FL00 | 9 | 6 | 54 | 216 | 2592 | 6 LED spotlights<br>3 Floor spotlights<br>4 Display LED lights | 26<br>4<br>4 | 0.156<br>0.012<br>0.016 | 404.35<br>31.104<br>41.472 | | | | | 2 Security cameras<br>3 Sound speakers | 2<br>3 | 2<br>100 | 0.004<br>0.3 | 10.368<br>777.6 |
| Mummies Hall 05 | 1 | FL00 | 9 | 6 | 54 | 216 | 2592 | 5 LED spotlight<br>4 Display LED light | 26<br>4 | 0.130<br>0.016 | 336.96<br>41.472 | | | | | 2 Security cameras<br>3 Sound speakers | 2<br>3 | 2<br>100 | 0.004<br>0.3 | 10.368<br>777.6 |
| Mummies Hall 06 | 1 | FL00 | 9 | 6 | 54 | 216 | 2592 | 8 LED spotlights<br>5 Floor spotlights<br>14 Display LED lights | 26<br>4<br>4 | 0.208<br>0.020<br>0.056 | 539.136<br>51.84<br>145.152 | | | | | 2 Security cameras<br>3 Sound speakers | 2<br>3 | 2<br>100 | 0.004<br>0.3 | 10.368<br>777.6 |
| Mummies Hall 07 | 1 | FL00 | 9 | 6 | 54 | 216 | 2592 | 9 LED spotlight<br>6 Floor Spotlight<br>4 Display LED light | 26<br>4<br>4 | 0.234<br>0.024<br>0.016 | 606.528<br>62.208<br>41.472 | Carier Central AC (Chilled water) | 2 | 14,920 | 77,345.28 | 2 Security cameras<br>3 Sound speakers | 2<br>3 | 2<br>100 | 0.004<br>0.3 | 10.368<br>777.6 |
| Mummies Hall 08 | 1 | FL00 | 9 | 6 | 54 | 216 | 2592 | 8 LED spotlight<br>6 Floor Spotlight<br>4 Display LED light | 26<br>4<br>4 | 0.208<br>0.024<br>0.016 | 539.136<br>62.208<br>41.472 | | | | | 2 Security cameras<br>3 Sound speakers | 2<br>3 | 2<br>100 | 0.004<br>0.3 | 10.368<br>777.6 |
| Mummies Hall 09 | 1 | FL00 | 9 | 6 | 54 | 216 | 2592 | 4 LED spotlights<br>5 Display LED lights | 26<br>4 | 0.104<br>0.020 | 269.568<br>51.84 | | | | | 2 Security cameras<br>2 Sound speakers | 2<br>2 | 2<br>100 | 0.004<br>0.3 | 10.368<br>777.6 |
| Mummies Hall 10 | 1 | FL00 | 9 | 6 | 54 | 216 | 2592 | 2 LED spotlights<br>4 Display LED lights | 26<br>4 | 0.052<br>0.016 | 134.784<br>41.472 | | | | | 2 Security cameras<br>3 Sound speakers | 2<br>3 | 2<br>100 | 0.004<br>0.3 | 10.368<br>777.6 |
| Mummies Hall 11 | 1 | FL00 | 9 | 6 | 54 | 216 | 2592 | 6 LED spotlights<br>5 Floor Lamps<br>8 Display LED spotlights | 26<br>4<br>4 | 0.156<br>0.020<br>0.032 | 404.352<br>51.84<br>82.944 | | | | | 2 Security cameras<br>3 Sound speakers | 2<br>3 | 2<br>100 | 0.004<br>0.3 | 10.368<br>777.6 |
| Mummies Hall 12 | 1 | FL00 | 9 | 6 | 54 | 216 | 2592 | 7 LED spotlights<br>4 Floor Lamps | 26<br>4 | 0.182<br>0.016 | 471.744<br>41.472 | | | | | 2 Security cameras<br>3 Sound speakers | 2<br>3 | 2<br>100 | 0.004<br>0.3 | 10.368<br>777.6 |
| Mummies Hall 13 | 1 | FL00 | 9 | 6 | 54 | 216 | 2592 | 6 LED spotlights<br>4 Floor Lamp | 26<br>4 | 0.156<br>0.016 | 404.352<br>41.472 | | | | | 1 Security camera<br>1 Sound speakers | 1<br>1 | 2<br>100 | 0.002<br>0.1 | 5.184<br>259.2 |

(Zone column, spanning all rows: Mummies floor)

**Table A3.** *Cont.*

| **Energy Consumption** | | | | | | | | | | | | | | | | | | | | |
|---|---|---|---|---|---|---|---|---|---|---|---|---|---|---|---|---|---|---|---|---|
| | | **Working Hours** | | | | | | **Lighting** | | | | | **HVAC** | | | | **Equipment & Apparatus** | | | |
| Zone | No. | Floor | Hrs/Day | Days/Wk | Hrs/Wk | Hrs/Month | Hrs/Y | Light Type | Power/Lamp (W) | Power (W) | Energy Use (kWh) | Cooling Type | No. | Power (W) | Energy Use (kWh) | Security Devices | No. | Power/Unit (W) | Power (W) | Energy Use (kWh) |
| Mummies Hall 14 | 1 | FL00 | 9 | 6 | 54 | 216 | 2592 | 26 LED spotlight<br>6 Floor Lamp | 26<br>4 | 0.676<br>0.024 | 1752.192<br>62.208 | | | | | 1 Security cameras<br>1 Sound speakers | 1<br>1 | 2<br>100 | 0.002<br>0.1 | 5.184<br>259.2 |
| Mummies Hall 15 | 1 | FL00 | 9 | 6 | 54 | 216 | 2592 | 10 LED spotlights<br>4 Floor Lamps<br>9 Display LED spotlights | 26<br>4<br>4 | 0.260<br>0.016<br>0.036 | 673.4<br>41.472<br>93.312 | | | | | 2 Security cameras<br>3 Sound speakers | 2<br>3 | 2<br>100 | 0.004<br>0.3 | 10.368<br>777.6 |
| Mummies Hall 16 | 1 | FL00 | 9 | 6 | 54 | 216 | 2592 | 17 LED spotlights<br>9 Floor Lamps | 26<br>4 | 0.442<br>0.036 | 1145.66<br>93.312 | | | | | 2 Security cameras<br>3 Sound speakers | 2<br>3 | 2<br>100 | 0.004<br>0.3 | 10.368<br>777.6 |
| Mummies Hall 17 | 1 | FL00 | 9 | 6 | 54 | 216 | 2592 | 5 LED spotlights<br>6 Floor Lamps | 26<br>4 | 0.130<br>0.024 | 336.96<br>62.208 | | | | | 2 Security cameras<br>3 Sound speakers | 2<br>3 | 2<br>100 | 0.004<br>0.3 | 10.368<br>777.6 |
| **Sub-total** | | | | | | | | | | | **11,860.26** | **Sub-total** | | | **77,345.28** | **Sub-total** | | | | **11,824** |
| **Total (kWh)** | | **101,029.54** | | | | | | | | | | | | | | | | | | |

## Appendix B

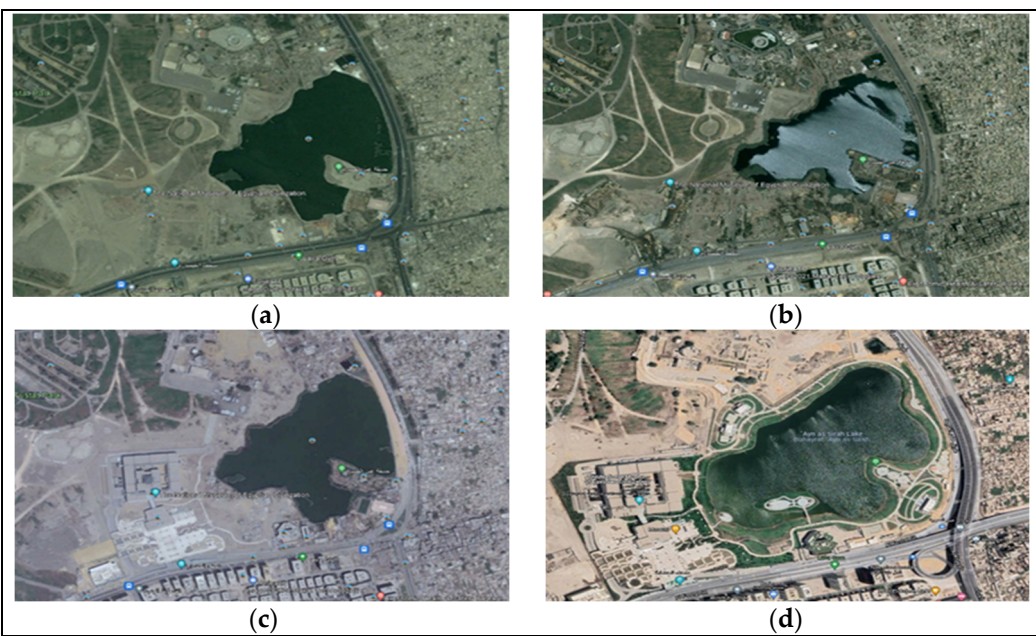

**Figure A1.** Timeline of Ain Elsira district development from the 70s to date (1980, 1990, 2004, 2021). (**a**) the lake in 1980; (**b**) the lake condition in 1990; (**c**) the lake in 2004; (**d**) the lake after development in 2021. (Source: Google Maps 1980, 1990, 2004, 2022).

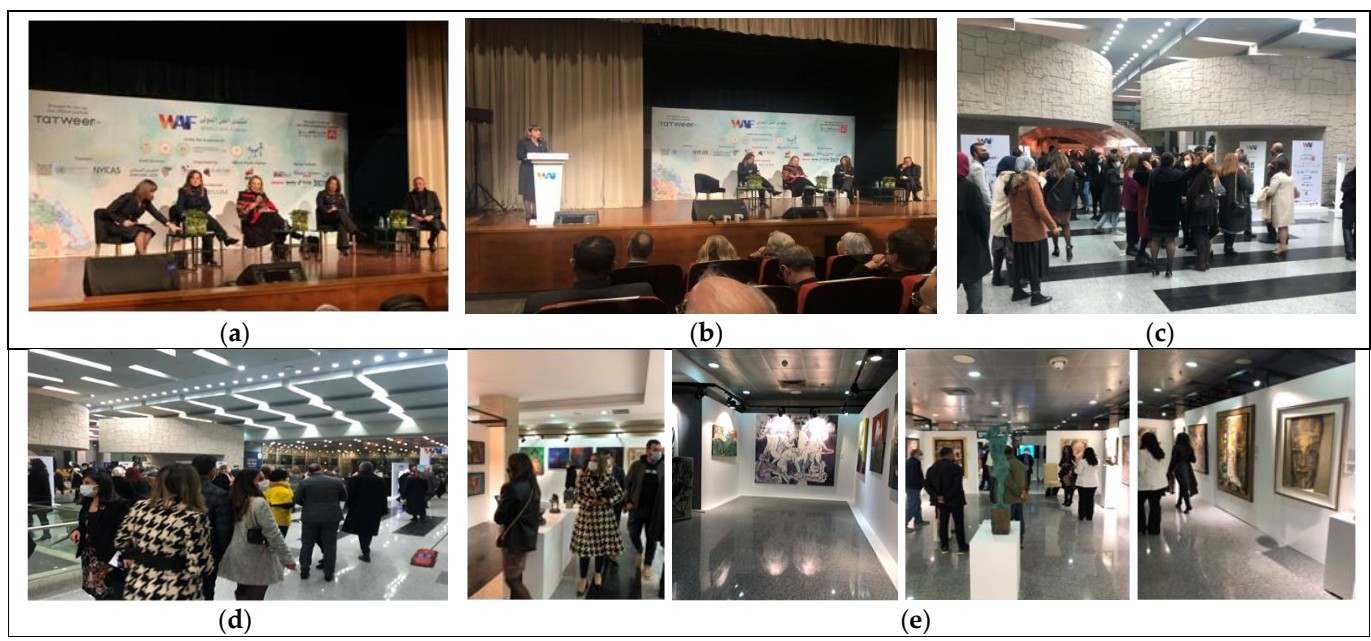

**Figure A2.** Symphony of Art and Sustainability–The first edition of the World Art Forum on International Contemporary Art Forum for Sustainable Development: (**a**) high-level speakers (Ahmed Ghoneim, Head of Authority of the NMEC, Ghada Waly, Executive Director of the United Nations Office on Drugs & Crime, General Director of United Nations Office at Vienna, Hala El-Said, Minister of Planning and Economic Development, and Elena Panova, UN Egypt Resident Coordinator); (**b**) Elena Panova addressing the floor; (**c**,**d**) invited guests visiting the exhibition; (**e**) part of the artwork.

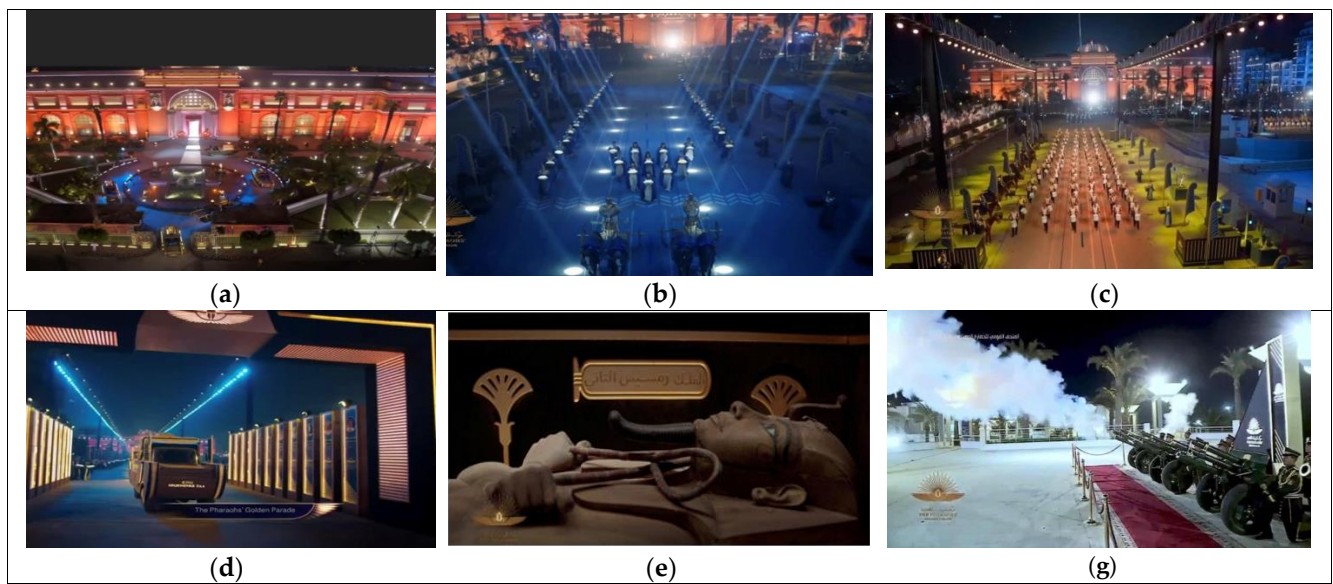

**Figure A3.** The Golden Mummies Parade to the NMEC: (**a**) the mummies parade leaving the Egyptian museum in Tahrir; (**b**) ancient ladies march; (**c**) military music parade; (**d**) king and queen mummies arriving at the NEMC; (**e**) King Ramses II inside a special car during the Golden Mummies Parade; (**f**) 21 rounds of cannon greeting the arrival of the mummies parade in the NMEC main plaza (Source: EgyptToday [58]).

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
