# Peer review of "Sustainability Assessment of the National Museum of Egyptian Civilization (NMEC): Environmental, Social, Economic, and Cultural Analysis"

_sustainability, doi:10.3390/su142013080_

Round 1

Reviewer 1 Report

Environmental, Social and Cultural  sustainability development  in museums practice is an important issue discussed by many  authors, researchers and museum managers during last decade. What I find lacking in this presented article is absence  in the references of  important models and guides published by OECD and ICOM (Museums and Local Development - OECD;   OECD-ICOM-GUIDE-MUSEUMS.pdf) . These  publications (link above) might create much larger context for NMEC to assess its possision, and provide additional sources to compare NMEC with other museums recently open.

Author Response

Response to Reviewer 1 Comments

Point 1: Environmental, Social and Cultural sustainability development in museums practice is an important issue discussed by many authors, researchers and museum managers during last decade. What I find lacking in this presented article is absence  in the references of  important models and guides published by OECD and ICOM (Museums and Local Development – OECD;   OECD-ICOM-GUIDE-MUSEUMS.pdf). These publications (link above) might create much larger context for NMEC to assess its possision, and provide additional sources to compare NMEC with other museums recently open.

Response 1: Please provide your response for Point 1. (in red)

  • Thank you very much for taking the time to review the paper. We also appreciate sharing these important resources of OECD - ICOM Guide. We have studied the recommended resources and added a major part of the introduction related to the ICOM report. This part in the introduction has been marked in yellow to highlight the added sections. We also presented and discussed further the recent definition of ‘museums’ based on the ICOM approved framework of the 26th ICOM General Conference held in Prague, Czech Republic on 24 August 2022 as well as several other statistics related to OECD – ICOM showing the cultural and economic impact of different museums as well. In addition, the authors have included Figure 1. We have also added the themes of ICOM Guide for museums as well as Table 1 highlighting the ICOM and OECD guide for Museums in terms of policy options for the local government along with the actions options for museums.

  • Moreover, we linked the results of the NMEC sustainability assessment with the ICOM and OECD Guide, mainly Action Options for Museums as well as relating the results to Table 2 and Figure 2 as well. This part is written as a new section the end of Discussions (highlighted in blue).

  • Furthermore, we compared the NMEC with other global museums recently open. This part is highlighted at the end of the Discussions section. We also added Table 10 (new) to depict the data for these museums along with that of NMEC, and plotted the comparison results in new figures (Fig. 36 and Fig. 37).

  • Finally, we sincerely thank you again for your valued comments, which helped us in enhancing the paper manuscript

Reviewer 2 Report

The authors carry out a study on the area of influence of the NMEC in the Old Cairo District, however, the article seems to be mainly focused on the analysis of the museum in relation to sustainability in its four dimensions: cultural, social, economic and environmental, without incorporating the territory and society and how it has been influenced by the presence of the museum.

The study conducted may be interesting, but the title does not correspond to the content of the article, so I propose a change in the title. On the other hand, although the term "sustainable development" is defined, the authors do not define the term "museum", which is mainly important, more so now after the new definition proposed and discussed by ICOM and approved on 24 August this year 2022, in the framework of the 26th General Conference of ICOM held in Prague. This definition includes community participation as an element of value, beyond visits by schoolchildren. This approach includes orange tourism or creative tourism, elements that could be incorporated into the study as proposals. 

It is recommended that the entire text be revised, as it contains some typos. The analysis of employment is limited only to the staff employed in the museum. If the analysis is about the impact of the museum on the territory or urban renewal derived from the museum, perhaps it should go deeper into the associated indirect employment, such as transport or transfer services between the main points of the district and the museum or associated shops, outside the museum. 

I would not put entertainment under 5.2.2 Education, this would fit better into the culture dimension. 

The content of heading 5.3.1 does not correspond to heading 5.3 of the economic analysis. I would either remove it or move it to the analysis in the social or cultural dimensions. In any case, the article is too long, with a length of more than 40 pages, and I would recommend removing content that does not contribute much to the results obtained.

In line 389, there is a table 5 where the values evaluated in relation to the indicators for evaluating the environmental dimension are mentioned, but this table does not appear in the text. 

It is not clear how they have calculated the indicators in table 9. The assessment is very subjective, especially in indicators related to the cultural dimension. 

In general, the study offers interesting results in relation to the sustainability of the museum, but does not provide sufficient knowledge on the influence of the museum on the territory of the district. It is accepted with important modifications. 

Author Response

Response to Reviewer 2 Comments

Point 1: The authors carry out a study on the area of influence of the NMEC in the Old Cairo District, however, the article seems to be mainly focused on the analysis of the museum in relation to sustainability in its four dimensions: cultural, social, economic and environmental, without incorporating the territory and society and how it has been influenced by the presence of the museum.

Response 1: Please provide your response for Point 1. (in red)

  • Thank you very much for taking the time to thoroughly review the paper. We deeply appreciate all the valued comments and notes received, that truly enhanced the coherence of the work. For this point, we have conducted a major update on the text, and we have reduced the parts discussing about the urban renewal of the old Cairo district, or the impact of the museum on the community, in order to keep the paper focused on the sustainability analysis of the NMEC. We have also changed the title of the paper to reflect this note to read:

‘’ Sustainability Assessment of the National Museum of Egyptian Civilization (NMEC): Environmental, Social, Economic, and Cultural Analysis “.

Point 2: The study conducted may be interesting, but the title does not correspond to the content of the article, so I propose a change in the title. On the other hand, although the term "sustainable development" is defined, the authors do not define the term "museum", which is mainly important, more so now after the new definition proposed and discussed by ICOM and approved on 24 August this year 2022, in the framework of the 26th General Conference of ICOM held in Prague. This definition includes community participation as an element of value, beyond visits by schoolchildren. This approach includes orange tourism or creative tourism, elements that could be incorporated into the study as proposals.

Response 2: Please provide your response for Point 2. (in red)

  • Thank you for your comment. We have changed the title of the paper to be more relevant to the work. The updated title is "Sustainability Assessment of the National Museum of Egyptian Civilization (NMEC): Environmental, Social, Economic, and Cultural Analysis“. Also, we thank you very much for sharing these important resources of OECD – ICOM Guide. We have studied the recommended resources and added a major part of the introduction related to the ICOM report. This part in the introduction has been marked in yellow to highlight the added part. The part discusses further the definition of ‘museums’ based on the ICOM as well as several other statistics related to OECD – ICOM showing the cultural and economic impact of different museums as well. In addition, the authors have included Figure 1 as a new one (since Figure 1 in the old version was deleted in the revised paper). The new Figure 1 addresses the themes of ICOM Guide for museums as well as Table 1. This new table is about ICOM and OECD Guide for Museums in terms of the policy options for the local government along with the actions options for museums. We have also added some text about community participation and a new figure (Figure 14 new numbering). We also linked the results of the social sustainability to the content in Figures 2 and 13.
  • Regarding orange tourism or creative tourism, we have highlighted this part by defining both terms, and we wrote a new section on the NMEC addressing orange tourism in the Discussions section. This can be found in the cited text [57].

Point 3: The study conducted may be interesting, but the title does not correspond to the content of the article, so I propose a change in the title. On the other hand, although the term "sustainable development" is defined, the authors do not define the term "museum", which is mainly important, more so now after the new definition proposed and discussed by ICOM and approved on 24 August this year 2022, in the framework of the 26th General Conference of ICOM held in Prague. This definition includes community participation as an element of value, beyond visits by schoolchildren. This approach includes orange tourism or creative tourism, elements that could be incorporated into the study as proposals.

Response 3: Please provide your response for Point 3. (in red)

Both Points 3 and 2 (above) are indentical, howerver below is our reposnse.

  • Thank you for your comment. We have changed the title of the paper to be more relevant to the work. The updated title is "Sustainability Assessment of the National Museum of Egyptian Civilization (NMEC): Environmental, Social, Economic, and Cultural Analysis“. Also, we thank you very much for sharing these important resources of OECD – ICOM Guide. We have studied the recommended resources and added a major part of the introduction related to the ICOM report. This part is depicted in the introduction has been marked in yellow to highlight the added part. The part discusses further the definition of ‘museums’ based on the ICOM as well as several other statistics related to OECD – ICOM showing the cultural and economic impact of different museums as well.
  • In addition, the authors have included Figure 1 as a new one (since Figure 1 in the old version was deleted in the revised paper). The new Figure 1 addresses the themes of ICOM Guide for museums as well as Table 1. This new table is about ICOM and OECD Guide for Museums in terms of the policy options for the local government along with the actions options for museums. We have also added some text about community participation and a new figure (Figure 14 new numbering). We also linked the results of the social sustainability to the content in Figures 2 and 13.
  • Regarding orange tourism or creative tourism, we have highlighted this part by defining both terms, and we wrote a new section on the NMEC addressing orange tourism in the Discussions section. This can be found in the cited text [57].

Point 4: It is recommended that the entire text be revised, as it contains some typos. The analysis of employment is limited only to the staff employed in the museum. If the analysis is about the impact of the museum on the territory or urban renewal derived from the museum, perhaps it should go deeper into the associated indirect employment, such as transport or transfer services between the main points of the district and the museum or associated shops, outside the museum.

Response 4: Please provide your response for Point 4. (in red)

  • Thank you for sharing this comment. The authors have conducted vigorous and extensive revision for the entire text and fixed all the English language spelling and grammar upon the comment. Regarding the other part, the study has focused mainly on the employed staff in the museum to maintain the integrity and data accuracy of the quantitative study conducted. We believe that the museum has positive impact in elevating hundreds of supportive services jobs associated indirectly with the museum such as transporters, tour guides, souvenir sellers, and others; and we will focus on this matter in our next paper once we have more clear numbers or estimations about the supportive services.

Point 5: I would not put entertainment under 5.2.2 Education, this would fit better into the culture dimension.

Response 5: Please provide your response for Point 5. (in red)

  • Thank you for sharing this comment. We have moved the entertainment part from section 5.2.2. Education and included it in the cultural sustainability dimension, under section heading 5.4.5. It is placed after Figure 18 (new numbering). This moved section is highlighted in yellow and blue.

Point 6: The content of heading 5.3.1 does not correspond to heading 5.3 of the economic analysis. I would either remove it or move it to the analysis in the social or cultural dimensions. In any case, the article is too long, with a length of more than 40 pages, and I would recommend removing content that does not contribute much to the results obtained.

Response 6: Please provide your response for Point 6. (in red)

  • Thank you for pointing out this part. We have changed the location of section 5.3.1, to be a new section 5.4.5.1 in the cultural sustainability after section 5.4.5.
  • Regarding the length of the paper, we completely agree with the comment in this regard, and the reason behind the length that we have included materials (tables, figures, photographs) as supplementary material to support the text. Based on the comment, we have deleted some figures and changed the location of the some supporting figures and tables to be placed in the Appendices (Appendix A & Appendix B) instead of being in the main text. However, the paper after revision is 18,918 words, which is less than most of journals text (less than 20,000 words).

Point 7: In line 389, there is a table 5 where the values evaluated in relation to the indicators for evaluating the environmental dimension are mentioned, but this table does not appear in the text. 

Response 7: Please provide your response for Point 7. (in red)

  • Thank you for pointing this out. All the tables and figures have been rigorously revised again for any misplacement or wrong numbering. The updated version of the paper shows the correct location of the Table 5 in association with related text.

Point 8: It is not clear how they have calculated the indicators in table 9. The assessment is very subjective, especially in indicators related to the cultural dimension. 

Response 8: Please provide your response for Point 8. (in red)

  • Thank you for the comment and question. The methodology part has been completely revised where we added more information about the qualitative and quantitative methods adopted for this study. All the new text in this section has been marked in yellow colour.
  • In terms of the quantitative approach, we mainly utilized the model published in the MDPI Sustainability published in 2016 under the title “Factors Influencing Museum Sustainability and Indicators for Museum Sustainability Measurement” and authored by Izabela Luiza Pop and Anca Borza. The 33-indicator model provides a tool to measure the sustainability of museums based on quantitative measures covering different aspects such as social, economic, environmental and cultural.
  • As for the method behind the calculation of each factor, we would like to present indicator number 6 which is part of the Social & Cultural Dimension as an example. The indicator discusses the Online Visibility of the NMEC, which is calculated – based on the 33 indicators published model – by dividing the number of number of mentions of the museum on Google (9003)/ Number of objects in the collection (2000), resulting in 4.502. The number of mentions of the museum on Google has been obtained data search techniques by the authors, while the number of the objects in the collection has been obtained by communicating with the Museum management. The same method has been used across the 33 indicators, where the data were obtained through authenticated and direct communication with the museum management. We have also added the sign * means multiply by and sign / means divided by to Table A1 in Appendix in order to make it easier and clearer to the reader.

Point 9: In general, the study offers interesting results in relation to the sustainability of the museum, but does not provide sufficient knowledge on the influence of the museum on the territory of the district. It is accepted with important modifications

Response 9: Please provide your response for Point 9. (in red)

  • Thank you very much for your comment. While the paper might not provide extensive details about the impact of the urban renewal of the district, the paper tends more to shed the light on the sustainability measures of the NMEC. We believe that the paper can be considered to be the pioneer to conduct this quantitative study on a new museum in Egypt measuring the 33 indicators, which can create a stepping stone and open the way for further researches to be conducted and compare the sustainability of different locations, as well as comparing the sustainability of specific location over years to measure the positive progress towards sustainability. To this end, the authors would like to thank you very much for all the valued comments and notes that truly enhanced the content and level of the coherence of the paper, and we have adopted all the changes required.
  • Finally, we added new references and deleted all referneces related to urban renewal.

Thank you again for your time and review.

Authors

Round 2

Reviewer 2 Report

Although the conclusions can be improved (I believe that more conclusions could have been drawn from the results obtained), the authors' effort to apply the suggested changes is favourably valued, one of them being greater clarity and exposition of the discussion of the research, which makes up for the scant content in the conclusions section. The clarifications in the methodology and presentation of the results are also appreciated. The article is accepted in its current state for publication.